

# Learning tensor networks with tensor cross interpolation: New algorithms and libraries

Yuriel Núñez Fernández[1,2*], Marc K. Ritter[3,4], Matthieu Jeannin[2], Jheng-Wei Li[2],
Thomas Kloss[1], Thibaud Louvet[2], Satoshi Terasaki[5], Olivier Parcollet[4,6],
Jan von Delft[3], Hiroshi Shinaoka[7] and Xavier Waintal[2†]

**1** Université Grenoble Alpes, Neel Institute CNRS, F-38000 Grenoble, France
**2** Université Grenoble Alpes, CEA, Grenoble INP, IRIG, Pheliqs, F-38000 Grenoble, France
**3** Arnold Sommerfeld Center for Theoretical Physics, Center for NanoScience,
and Munich Center for Quantum Science and Technology,
Ludwig-Maximilians-Universität München, 80333 Munich, Germany
**4** Center for Computational Quantum Physics, Flatiron Institute,
162 5th Avenue, New York, NY 10010, USA
**5** AtelierArith, 980-0004, Miyagi, Japan
**6** Université Paris-Saclay, CNRS, CEA, Institut de physique théorique,
91191, Gif-sur-Yvette, France
**7** Department of Physics, Saitama University, Saitama 338-8570, Japan

⋆ yurielnf@gmail.com , † xavier.waintal@cea.fr

## Abstract

The tensor cross interpolation (TCI) algorithm is a rank-revealing algorithm for decomposing low-rank, high-dimensional tensors into tensor trains/matrix product states (MPS). TCI learns a compact MPS representation of the entire object from a tiny training data set. Once obtained, the large existing MPS toolbox provides exponentially fast algorithms for performing a large set of operations. We discuss several improvements and variants of TCI. In particular, we show that replacing the cross interpolation by the partially rank-revealing LU decomposition yields a more stable and more flexible algorithm than the original algorithm. We also present two open source libraries, `xfac` in Python/C++ and `TensorCrossInterpolation.jl` in Julia, that implement these improved algorithms, and illustrate them on several applications. These include sign-problem-free integration in large dimension, the "superhigh-resolution" quantics representation of functions, the solution of partial differential equations, the superfast Fourier transform, the computation of partition functions, and the construction of matrix product operators.

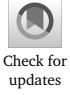

# 1 Introduction

Tensor networks, widely used in quantum physics, are increasingly being used also in other areas of science. They offer compressed representations of functions of one or more variables. *A priori*, a tensor of degree $\mathcal{L}$, $F_{\sigma_1 \cdots \sigma_{\mathcal{L}}}$, with indices $\sigma_\ell = 1, \ldots, d$, requires exponential resources in memory and computation time to be stored and manipulated, since it contains $d^{\mathcal{L}}$ elements—a manifestation of the well-known *curse of dimensionality*. However, just as a matrix (a tensor of degree 2) can be compressed if it has low rank, a tensor of higher degree can be strongly compressed if it has a low-rank structure. Then, exponential reductions in computational costs for performing standard linear algebra operations are possible, allowing the curse of dimensionality to be evaded.

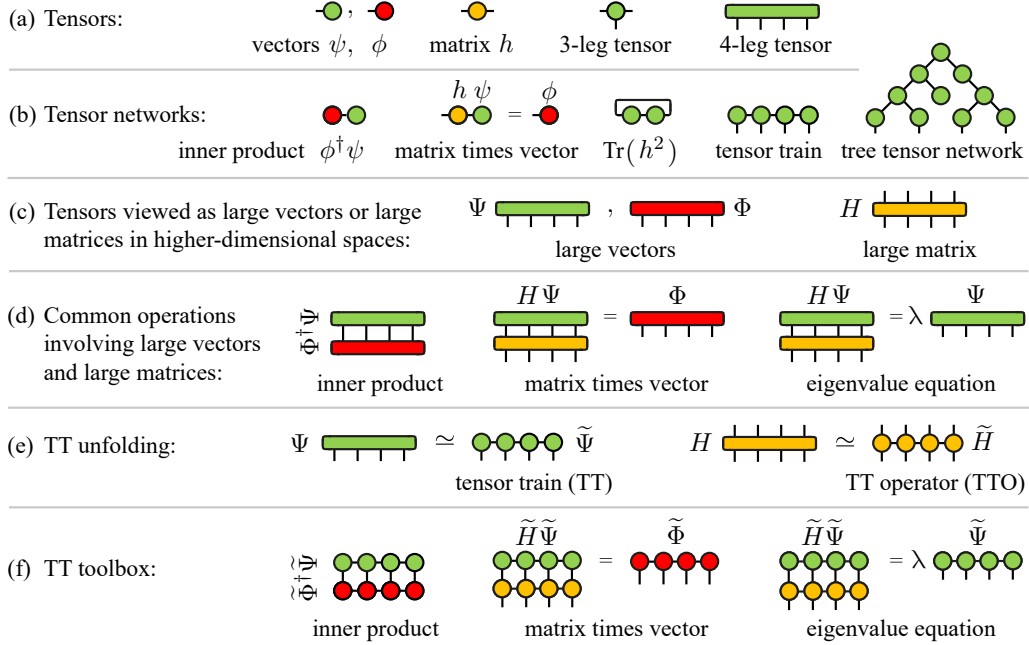

Figure 1: Schematic depiction of key ingredients of the standard MPS toolbox. (a) Colored shapes with legs represent tensors with indices. (b) Tensors connected by bonds, representing sums over shared indices, form tensor networks. (c) Tensors and linear operators acting on them represent large vectors (green, red) and large matrices (yellow) in higher-dimensional vector spaces. (d) Common calculations in these spaces include computing inner products $\Phi^\dagger\Psi$, solving linear problems $H\Psi = \Phi$, computing a few eigenvalues $H\Psi = \lambda\Psi$, and more. (e) Tensors representing large vectors or linear operators can be *unfolded* into MPS or tensor train operators (MPO), respectively. (f) The standard MPS toolbox includes algorithms for performing calculations with MPS and MPO. If these have have low rank, such calculations can be performed in polynomial time, even for exponentially large vector spaces. The `xfac` and `TCI.jl` libraries expand the MPS toolbox by providing tools for unfolding tensors into MPS using exponentially fast tensor cross interpolation (TCI) algorithms, for expressing functions as MPO, and for manipulating the latter.

In physics, functions describing physical quantities and the tensors representing them indeed often do have a hidden structure. A prominent example is the density matrix renormalization group (DMRG), the method of choice for treating one-dimensional quantum lattice models [1]. There, quantum wavefunctions and operators are expressed as tensor networks that in the physics community are called *matrix product states* (MPSs) and *matrix product operators* (MPOs), respectively, or *tensor trains* in the applied mathematics community. (In this work, "MPS" and "tensor train" will be used interchangeably.) Many algorithms for manipulating such objects have been developed in the quantum information and many-body communities [2–5]. We collectively refer to them as the "standard MPS toolbox" [6,7]; Figure 1 depicts some of its ingredients using tensor network diagrams. These algorithms achieve exponential speedup for linear algebra operations (computing scalar products, solving linear systems, diagonalization, ...) with large but compressible vectors and matrices. Although initially developed for many-body physics, the MPS toolbox is increasingly being used in other, seemingly unrelated, domains of application. It appears, indeed, that many common mathematical objects are in fact of low rank.

A crucial recent development is the emergence of a new category of algorithms that allow one to detect low-rank properties and automatically construct the associated low rank tensor representations. They are collectively called tensor cross interpolation (TCI) algorithms [8–12], the subject of this article. Based on the cross interpolation (CI) decomposition of matrices instead of the singular value decomposition (SVD) widely used in standard tensor network techniques, TCI algorithms construct low-rank decompositions of a given tensor. Their main characteristic is that they do not take the entire tensor as input (in contrast to SVD-based decompositions) but request only a small number of tensor elements (the "pivots"). Their costs thus scale linearly with $\mathcal{L}$, even though the tensor has exponentially many elements. In this sense, TCI algorithms are akin to machine learning: they seek compact representations of a large dataset (the tensor) based on a small subset (the pivots). Moreover, they are *rank-revealing*: for low-rank tensors they rapidly find accurate low-rank decompositions (in most cases, see discussions below); for high-rank tensors they exhibit slow convergence rather than giving bad decompositions. TCI has been used recently, e.g., as an efficient (sign-problem-free) alternative to Monte Carlo sampling for calculating high-dimensional integrals arising in Feynman diagrams for the quantum many-body problem [13]; to find minima of functions [14]; to calculate topological invariants [15]; to calculate overlaps between atomic orbitals [16]; to solve the Schrödinger equation of the $H_2^+$ ion [16]; and, in mathematical finance, to speed up Fourier-transform-based option pricing [17].

Among the many applications of tensor networks, the so-called *quantics* [18–20] representation of functions of one or more variables has recently gained interest in various fields, including many-body field theory [21–25], turbulence [26–28], plasma physics [29], quantum chemistry [16], and denoising in quantum simulation [30]. Quantics tensor representations yield exponentially high resolution, and often have low-rank, even for functions exhibiting scale separation between large- and small-scale features. Such representation can be efficiently revealed using TCI [21]. Moreover, it can be exploited to perform many standard operation on functions (e.g. integration, multiplication, convolution, Fourier transform, ...) exponentially faster than when using naive brute-force discretizations. For example, quantics yields a compact basis for solving partial differential equations, similar to a basis of orthogonal (e.g. Chebyshev) polynomials.

This article has three main goals:

- We present new variants of TCI algorithms that are more robust and/or faster than previous ones. They are based on rank-revealing partial LU (prrLU) decomposition, which is equivalent to but more flexible and stable than traditional CI. The new variants offer useful new functionality beyond proposing new pivots, such as the ability to remove bad pivots, to add global pivots, to compress an existing MPS.

- We showcase various TCI applications (both with and without quantics), such as integrating multivariate functions, computing partition functions, integrating partial differential equations, constructing complex MPOs for many-body physics.

- We present the API of two open source libraries that implement TCI and quantics algorithms as well as related tools: `xfac`, written in C++ with python bindings; and `TensorCrossInterpolation.jl` (or `TCI.jl` for short), written in Julia.

Below, Sec. 2 very briefly describes and illustrates the capabilities of TCI, serving as a minimal primer for starting to use the libraries. Readers interested mainly in trying out TCI (or learning what it can do) may subsequently proceed directly to Secs. 5–7, which present several illustrative applications. Sec. 3 describes the formal relation between CI and prrLU at the matrix level, Sec. 4 presents our prrLU-based algorithms for tensors of higher degree. Finally Sec. 8 discusses the API of the `xfac` and `TCI.jl` libraries. Several appendices are devoted to technical details.

## 2 An introduction to tensor cross interpolation (TCI)

In this section, we present a quick primer on TCI algorithms without details, to set the scene for exploring our libraries and studying the examples in Section 5 and beyond.

### 2.1 The input and output of TCI

Consider a tensor $F$ of degree $\mathcal{L}$, with elements $F_{\boldsymbol{\sigma}}$ labeled by indices $\boldsymbol{\sigma} = (\sigma_1, \ldots, \sigma_{\mathcal{L}})$, with $1 \leq \sigma_{\ell} \leq d_{\ell}$. For simplicity, we will denote the dimension $d = d_{\ell}$ if all the dimensions $d_{\ell}$ are equal. Our goal is to obtain an approximate factorization of $F$ as a matrix product state (MPS), that we denote $\widetilde{F}_{\boldsymbol{\sigma}}$. An MPS has the following form and graphical representation:

$$F_{\boldsymbol{\sigma}} \approx \widetilde{F}_{\boldsymbol{\sigma}} = \prod_{\ell=1}^{\mathcal{L}} M_{\ell}^{\sigma_{\ell}} = [M_1]_{1a_1}^{\sigma_1} [M_2]_{a_1 a_2}^{\sigma_2} \cdots [M_{\mathcal{L}}]_{a_{\mathcal{L}-1} 1}^{\sigma_{\mathcal{L}}}, \tag{1}$$

Implicit summation over repeated indices (Einstein convention) is understood and depicted graphically by connecting tensors by bonds. Each three-leg tensor $M_{\ell}$ has elements $[M_{\ell}]_{a_{\ell-1} a_{\ell}}^{\sigma_{\ell}}$, and can also be viewed as a matrix $M_{\ell}^{\sigma_{\ell}}$ with indices $a_{\ell-1}, a_{\ell}$. The *external indices* $\sigma_{\ell}$ have dimensions $d_{\ell}$. The *internal (or bond) indices* $a_{\ell}$ have dimensions $\chi_{\ell}$, called the bond dimensions of the tensor. By convention, we use $\chi_0 = \chi_{\mathcal{L}} = 1$ to preserve a matrix product structure. We define $\chi \equiv \max_{\ell} \chi_{\ell}$ as the *rank* of the tensor.

The approximation (1) can be made arbitrarily accurate by increasing $\chi_{\ell}$, potentially exponentially with $\mathcal{L}$ like $\chi_{\ell} \sim \min\{d^{\ell}, d^{\mathcal{L}-\ell}\}$. A tensor is said to be *compressible* or *low-rank* if it can be approximated by a MPS form with a small rank $\chi$.

TCI algorithms aim to construct low-rank MPS approximations (actually interpolations) for a given tensor $F$ using a minimal number of its elements. They are high-dimensional generalizations of matrix decomposition methods, like the cross interpolation (CI) decomposition or the partially rank-revealing LU decomposition (prrLU) [31]. Indeed, they progressively refine the $\widetilde{F}$ approximation, increasing the ranks, by searching for *pivots* (high-dimensional generalizations of Gaussian elimination pivots), using CI or prrLU on two-dimensional slices of the tensor. TCI algorithms come with an error estimate $\epsilon(\chi_{\ell})$, which can be reduced below a specified tolerance $\tau$ by suitably increasing $\chi_{\ell}$. Moreover, they are *rank-revealing*: if a given tensor $F$ admits a low-rank MPS approximation, the algorithms will almost always find it; if the tensor is not of low rank (e.g. a tensor with random entries), the algorithms fail to converge and the computed error remains large.

Concretely, TCI algorithms take as input a tensor $F$ in the form of a function returning the value $F_{\boldsymbol{\sigma}}$ for any $\boldsymbol{\sigma}$; they explore its structure by sampling (in a deterministic way) some of its elements; and they return as output a list of tensors $M_1, \ldots, M_{\mathcal{L}}$ for the MPS approximation $\widetilde{F}$. Importantly, TCI algorithms do not require *all* $d^{\mathcal{L}}$ tensor elements of $F$ but can construct $\widetilde{F}$ by calling $F_{\boldsymbol{\sigma}}$ only $\mathcal{O}(\mathcal{L} d \chi^2)$ times. The TCI algorithms have a time complexity $\mathcal{O}(\mathcal{L} d \chi^3)$ [12], that is exponentially smaller than the total number of elements. The TCI form is fully specified by $\mathcal{O}(\mathcal{L} \chi^2)$ pivot indices, which are sufficient to reconstruct the whole tensor at the specified tolerance. Furthermore, the TCI form allows an efficient evaluation of any tensor element.

Since TCI algorithms sample a given tensor $F$ in a deterministic manner to construct a compressed representation $\widetilde{F}$, they can be viewed as machine learning algorithms. We will discuss the analogy with neural networks learning techniques in Section 4.8.

## 2.2 An illustrative application: Integration in large dimension

TCI algorithms allow new usages of the MPS tensor representation not contained in other tensor toolkits, for example integration or summation in large dimensions [8,12]. Consider a function $f(\mathbf{x})$, with $\mathbf{x} = (x_1, \ldots, x_{\mathcal{L}})$. We wish to calculate the $\mathcal{L}$-dimensional integral $\int d^{\mathcal{L}}\mathbf{x}\, f(\mathbf{x})$. We map $f$ onto a tensor $F$ by discretizing each variable $x_\ell$ onto a grid of $d$ distinct points $\{p_1, p_2, \ldots, p_d\}$, e.g. the points of a Gauss quadrature or the Chebyshev points. Then, the *natural tensor representation $F$* of $f$ on this grid is defined as

$$F_{\boldsymbol{\sigma}} = f(p_{\sigma_1}, p_{\sigma_2}, \ldots, p_{\sigma_{\mathcal{L}}}) = \underset{\sigma_1 \quad \sigma_2 \quad \cdots \quad \sigma_{\mathcal{L}}}{\boxed{\phantom{xxxxxxxxxxxxxxx}}} \quad , \tag{2}$$

with $\sigma_\ell = 1, \ldots, d$. This can be given as input to TCI. The resulting $\widetilde{F}$ yields a factorized approximation for $f$ when all its arguments lie on the grid,

$$f(x_1, \ldots, x_{\mathcal{L}}) \approx M_1(x_1) M_2(x_2) \cdots M_{\mathcal{L}}(x_{\mathcal{L}}) = \overset{M_1 \ M_2 \qquad\qquad M_{\mathcal{L}}}{\underset{x_1 \ \ x_2 \quad \cdots \quad x_{\mathcal{L}}}{\times\!-\!\circ\!-\!\circ\!-\!\circ\!-\!\circ\!-\!\circ\!-\!\times}}, \tag{3}$$

for $x_\ell \in \{p_1, p_2, \ldots, p_d\}$, with $M_\ell(p_{\sigma_\ell}) \equiv M_\ell^{\sigma_\ell}$. The notation $M_\ell(x)$ reflects the fact that the approximation can be extended to the continuum, i.e. for all $x$ (see the discussion in App. A.4, as well as Eqs. (7–9) of Ref. [13]). When $\widetilde{F}$ is low rank, $f$ is *almost separable* (it would be separable if the rank $\chi = 1$). The integral of the factorized $f$ is straightforward to compute as [8, 12, 13]

$$\int d^{\mathcal{L}}\mathbf{x} f(\mathbf{x}) \approx \int dx_1\, M_1(x_1) \int dx_2\, M_2(x_2) \cdots \int dx_{\mathcal{L}}\, M_{\mathcal{L}}(x_{\mathcal{L}}), \tag{4}$$

i.e. *one*-dimensional integrals followed by a sequence of matrix-vector multiplications. Since TCI algorithms can compute the compressed MPS form with a "small" number of evaluations of $f$ (one for each requested tensor element), the integral computation is performed in $\mathcal{O}(\mathcal{L}d\chi^2) \ll \mathcal{O}(d^{\mathcal{L}})$ calls to the function $f(\mathbf{x})$. In practice, this method has been shown to be very successful, even when the function $f$ is highly oscillatory. For example, it was recently shown to outperform traditional approaches for computing high-order perturbative expansions in the quantum many-body problem [13, 32]. Quite generally, TCI can be considered as a possible alternative to Monte Carlo sampling, particularly attractive if a sign problem (rapid oscillations of the integrand) makes Monte Carlo fail.

As an illustration, we compute a 10-dimensional integral with an oscillatory argument,

$$I = 10^3 \int\limits_{[-1,+1]^{10}} d^{10}\mathbf{x} \cos\left(10 \sum_{\ell=1}^{10} x_\ell^2\right) \exp\left[-10^{-3}\left(\sum_{\ell=1}^{10} x_\ell\right)^4\right], \tag{5}$$

using TCI with Gauss–Kronrod quadrature rules. As shown in Fig. 2, TCI converges approximately as $1/N_{\mathrm{eval}}^4$, where $N_{\mathrm{eval}}$ is the number of evaluations of the integrand. For comparison, Monte Carlo integration would converge as $\mathcal{O}(1/\sqrt{N_{\mathrm{eval}}})$ and encounter a sign problem due to the cosine term in the integrand.

In practice, our `xfac`/`TCI.jl` libraries take a user-defined, real- or complex-valued function $f(\mathbf{x})$ as input and construct a tensor train representation $\widetilde{F}_{\boldsymbol{\sigma}}$ with a user-specified tolerance $\tau$ or rank $\chi$. Our TCI toolbox contains algorithms to decompose a tensor $F$ or to recompress a given MPS decomposition. After a MPS form of $F$ has been obtained, it can be used directly or transformed into one of several canonical forms (cf. Sec. 4.5) and used with other standard tensor toolkits such as ITensor [33]. In Sections 5 and beyond, we present various examples of applications. Readers interested mainly in these may prefer to the upcoming two Sections 3 and 4, which are devoted to the details of the algorithms.

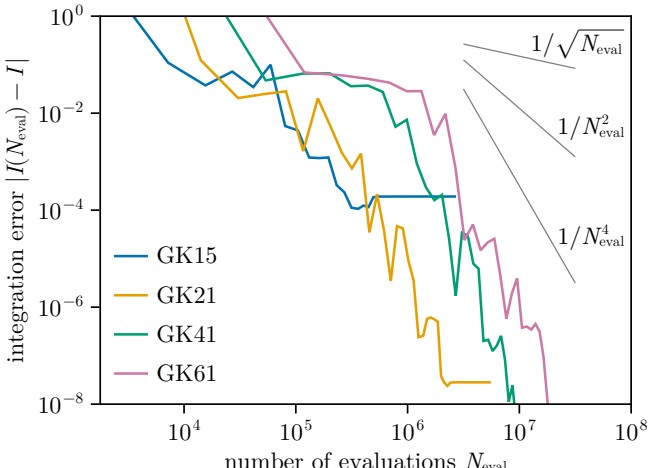

Figure 2: Convergence of the 10-dimensional integral $I$ of Eq. (5). $I(N_{\text{eval}})$ is computed using TCI with 15, 21, 41 and 61-point Gauss–Kronrod quadrature in each dimension, and $N_{\text{eval}}$ is the number of evaluations of the integrand. With 41- and 61-point quadrature, the value converges to $I = -5.4960415218049$. Convergence of the lower-order quadrature rules is limited by the number of discretization points.

## 3 Mathematical preliminaries: Low-rank decomposition of matrices from a few rows and columns

The original TCI algorithm [8–10] is based on the matrix cross interpolation (CI) formula, which constructs low rank approximations of matrices from crosses formed by subsets of their rows and columns. In this paper, we focus on a different but mathematically equivalent strategy for constructing cross interpolations, based on partial rank-revealing LU (prrLU) decompositions. This offers several advantages, in particular in term of stability.

A low-rank matrix is strongly *compressible*. Indeed, if $A = (\mathbf{a}_1, \dots, \mathbf{a}_n)$ is an $m{\times}n$ matrix with column vectors $\mathbf{a}_j$ and (low) rank $\chi$, each column can be expressed as a linear combination of a subset of $\chi$ of them ($\mathbf{a}_j = \sum_{i=1}^{\chi} \mathbf{b}_i C_{ij}$). Denoting the $m \times \chi$ submatrix $B = (\mathbf{b}_1, \dots, \mathbf{b}_\chi)$, we have $A = BC$. It is sufficient to store $B$ and $C$, i.e. $\chi(m+n)$ elements instead of $mn$, which is a large reduction when the rank is small ($\chi \ll \min(m,n)$).

The compressibility extends to matrices which are *approximately* of low rank. Using the SVD decomposition, a matrix $A$ is rewritten as $A = UDV^\dagger$ with $D$ a diagonal matrix of singular values, which can be truncated at some tolerance to yield a low-rank approximation $\widetilde{A}$ of $A$. While SVD is optimal (it minimizes the error $\|A - \widetilde{A}\|_F$ in the Frobenius norm), this comes at a cost: the *entire* matrix $A$ is required for the decomposition. Here, we are interested in CI and prrLU, two low-rank approximations techniques which require only a subset of rows and columns of the matrix. Both are well-known and in fact intimately related [34].

This section is organized as follows: after recalling CI in Section 3.1, we review some standard material on Schur complements, prrLU and its relationship with CI. This section focuses exclusively on matrices; we generalize to tensors in the next section.

### 3.1 Matrix cross interpolation (CI)

Let us first recall the matrix cross interpolation (CI) formula [11, 35–42], cf. section III of Ref. [13] for an introduction.

Let $A$ be a $m \times n$ matrix of rank $\chi$. We write $\mathbb{I} = \{1, \dots, m\}$ and $\mathbb{J} = \{1, \dots, n\}$ for the ordered sets of all row or column indices, respectively, and $\mathcal{I} = \{i_1, \dots, i_{\widetilde{\chi}}\} \subset \mathbb{I}$ and $\mathcal{J} = \{j_1, \dots, j_{\widetilde{\chi}}\} \subset \mathbb{J}$ for subsets of $\widetilde{\chi}$ row and column indices. Following a standard MATLAB convention, we write $A(\mathcal{I}, \mathcal{J})$ for the submatrix or *slice* containing all intersections of $\mathcal{I}$-rows and $\mathcal{J}$-columns (i.e. rows and columns labeled by indices in $\mathcal{I}$ and $\mathcal{J}$, respectively), with elements

$$[A(\mathcal{I}, \mathcal{J})]_{\alpha\beta} \equiv A_{i_\alpha, j_\beta}, \tag{6}$$

$\forall \alpha, \beta \in \{1, \dots, \widetilde{\chi}\}$. In particular, $A(\mathbb{I}, \mathbb{J}) = A$. In the following, we assume $\widetilde{\chi} \leq \chi$, with $\mathcal{I}$ and $\mathcal{J}$ chosen such that the matrix $A(\mathcal{I}, \mathcal{J})$ is non-singular. We define the following slices of $A$:

$$P = A(\mathcal{I}, \mathcal{J}), \qquad C = A(\mathbb{I}, \mathcal{J}), \qquad R = A(\mathcal{I}, \mathbb{J}). \tag{7}$$

$P = A(\mathcal{I}, \mathcal{J})$ is the *pivot matrix*. Its elements are called *pivots*, labeled by index pairs $(i, j) \in \mathcal{I} \times \mathcal{J}$. These index pairs are called pivots, too (a common abuse of terminology), and the index sets $\mathcal{I}, \mathcal{J}$ specifying them are called *pivot lists*. In other words, the slice $C = A(\mathbb{I}, \mathcal{J})$ gathers all columns containing pivots, the slice $R = A(\mathcal{I}, \mathbb{J})$ gathers all rows containing pivots, and $P$ contains their intersections (thus it is a subslice of both).

The *CI formula* gives a rank-$\widetilde{\chi}$ approximation $\widetilde{A}$ of $A$ [38] that can be expressed in the following equivalent forms:

$$A \approx C P^{-1} R = \widetilde{A}, \tag{8}$$

$$A(\mathbb{I}, \mathbb{J}) \approx A(\mathbb{I}, \mathcal{J}) P^{-1} A(\mathcal{I}, \mathbb{J}) = \widetilde{A}(\mathbb{I}, \mathbb{J}), \tag{9}$$

$$A_{i'j'} = \underset{\mathbb{I}}{\overset{i'}{\rule{0pt}{0pt}}} \fbox{} \underset{\mathbb{J}}{\overset{j'}{\rule{0pt}{0pt}}} \approx \underset{\mathbb{I}}{\overset{i'}{\rule{0pt}{0pt}}} \fbox{} \underset{\mathcal{J}}{\overset{j}{\rule{0pt}{0pt}}} \blacklozenge \underset{\mathcal{I}}{\overset{i}{\rule{0pt}{0pt}}} \fbox{} \underset{\mathbb{J}}{\overset{j'}{\rule{0pt}{0pt}}},$$

The third line depicts this factorization diagrammatically through the *insertion of two pivot bonds*. There, the external indices $i' \in \mathbb{I}$ and $j' \in \mathbb{J}$ are fixed, $\blacklozenge$ represents $P^{-1}$, and the two internal bonds represent sums $\sum_{j \in \mathcal{J}} \sum_{i \in \mathcal{I}}$ over the pivot lists $\mathcal{I}, \mathcal{J}$. The fourth line visualizes this for $\widetilde{\chi} = 3$, with $\mathcal{J}$-columns colored red, $\mathcal{I}$-rows blue, and pivots purple.

The CI formula (9) has two important properties: (i) For $\widetilde{\chi} = \chi$, Eq. (9) exactly reproduces the entire matrix, $\widetilde{A} = A$ (as explained below). (ii) For any $\widetilde{\chi} \leq \chi$ it yields an interpolation, i.e. it exactly reproduces all $\mathcal{I}$-rows and $\mathcal{J}$-columns of $A$. Indeed, when considering only the $\mathcal{I}$-rows or $\mathcal{J}$-columns of $\widetilde{A}(\mathbb{I}, \mathbb{J})$ in Eq. (9), we obtain

$$\quad : \quad \widetilde{A}(\mathcal{I}, \mathbb{J}) = A(\mathcal{I}, \mathbb{J}), \quad \text{since} \quad A(\mathcal{I}, \mathcal{J}) P^{-1} = \mathbb{1}, \tag{10a}$$

$$\quad : \quad \widetilde{A}(\mathbb{I}, \mathcal{J}) = A(\mathbb{I}, \mathcal{J}), \quad \text{since} \quad P^{-1} A(\mathcal{I}, \mathcal{J}) = \mathbb{1}, \tag{10b}$$

where $\mathbb{1}$ denotes a $\widetilde{\chi} \times \widetilde{\chi}$ unit matrix.

The accuracy of a CI interpolation depends on the choice of pivots. Efficient heuristic strategies for finding good pivots are thus of key importance. They will be discussed in Sec. 3.3.2.

## 3.2 A few properties of Schur complements

This section discusses an important object of linear algebra, the Schur complement. Of primary importance to us are two facts that allow us to make the connection between CI and prrLU.

First, the Schur complement is essentially the *error* of the CI approximation. Second, the Schur complement can be obtained *iteratively* by eliminating (in the sense of Gaussian elimination) rows and columns of the initial matrix one after the other and in any order. With these two properties, we will be able to prove that the prrLU algorithm discussed in the next section actually yields a CI approximation.

### 3.2.1 Definitions and basic properties

Let us consider a matrix $A$ made of 4 blocks

$$A = \begin{pmatrix} A_{11} & A_{12} \\ A_{21} & A_{22} \end{pmatrix}, \tag{11}$$

with $A_{11}$ assumed square and invertible. The Schur complement $[A/A_{11}]$ is defined by

$$[A/A_{11}] \equiv A_{22} - A_{21}(A_{11})^{-1}A_{12}. \tag{12}$$

The matrix $A$ can be factorized as

$$\begin{pmatrix} A_{11} & A_{12} \\ A_{21} & A_{22} \end{pmatrix} = \begin{pmatrix} \mathbb{1}_{11} & 0 \\ A_{21}A_{11}^{-1} & \mathbb{1}_{22} \end{pmatrix} \begin{pmatrix} A_{11} & 0 \\ 0 & [A/A_{11}] \end{pmatrix} \begin{pmatrix} \mathbb{1}_{11} & A_{11}^{-1}A_{12} \\ 0 & \mathbb{1}_{22} \end{pmatrix}. \tag{13}$$

This leads to the Schur determinant identity

$$\det A = \det A_{11} \det[A/A_{11}], \tag{14}$$

and (by inverting (13), see also Appendix A.1) to the relation

$$\left(A^{-1}\right)_{22} = [A/A_{11}]^{-1}. \tag{15}$$

### 3.2.2 The quotient property

When used for successively eliminating blocks, the Schur complement does not depend on the order in which the different blocks are eliminated. This is expressed by *the quotient property of the Schur complement* [43]. We illustrate this property on a $3 \times 3$ block matrix,

$$A = \begin{pmatrix} A_{11} & A_{12} & A_{13} \\ A_{21} & A_{22} & A_{23} \\ A_{31} & A_{32} & A_{33} \end{pmatrix}, \qquad B \equiv \begin{pmatrix} A_{11} & A_{12} \\ A_{21} & A_{22} \end{pmatrix}, \tag{16}$$

where $B$ is a submatrix of $A$. We assume that $A_{11}$ and $A_{22}$ are square and invertible. Then the quotient formula reads

$$\left[[A/A_{11}]/[B/A_{11}]\right] = \left[A/B\right] = \left[[A/A_{22}]/[B/A_{22}]\right]. \tag{17}$$

A simple explicit proof of this property is provided in Appendix A.1, see also [44].

As the order of block elimination does not matter, we will use a simpler notation

$$\left[[A/1]/2\right] = \left[[A/2]/1\right] = \left[A/(1,2)\right], \tag{18}$$

where /1 or /2 denotes the elimination of the 11- or 22 block, and /(1, 2) the elimination of the square matrix containing both. Let us also note that permutations of rows and columns in the 11- and 22-blocks can be taken before or after taking the Schur complement $[A/(1,2)]$ without affecting the result [44]. For matrices involving a larger number of blocks, iterative application of the Schur quotient rule to successively eliminate blocks 11 to $xx$ reads

$$\left[[[A/1]/2]\ldots]/x\right] = \left[A/(1,2,\ldots,x)\right]. \tag{19}$$

### 3.2.3 Relation with CI

The error in the matrix cross interpolation formula is directly given by the Schur complement to the pivot matrix.

To see this, let us permute the rows and columns of $A$ such that all pivots lie in the first $\widetilde{\chi}$ rows and columns, labeled $\mathcal{I}_1 = \mathcal{J}_1 = \{1, \ldots, \widetilde{\chi}\}$, with $\mathcal{I}_2 = \mathbb{I} \setminus \mathcal{I}_1$ and $\mathcal{J}_2 = \mathbb{J} \setminus \mathcal{J}_1$ labeling the remaining rows and columns, respectively. Then, the permuted matrix (again denoted $A$ for simplicity) has the block form

$$A(\mathbb{I}, \mathbb{J}) = \begin{pmatrix} A(\mathcal{I}_1, \mathcal{J}_1) & A(\mathcal{I}_1, \mathcal{J}_2) \\ A(\mathcal{I}_2, \mathcal{J}_1) & A(\mathcal{I}_2, \mathcal{J}_2) \end{pmatrix} = \begin{pmatrix} A_{11} & A_{12} \\ A_{21} & A_{22} \end{pmatrix}, \tag{20}$$

and the pivot matrix is $P = A_{11} = A(\mathcal{I}_1, \mathcal{J}_1)$. The CI formula (9) now takes the form

$$\widetilde{A} = \begin{pmatrix} A_{11} \\ A_{21} \end{pmatrix} (A_{11})^{-1} \begin{pmatrix} A_{11} & A_{12} \end{pmatrix} = \begin{pmatrix} A_{11} & A_{12} \\ A_{21} & A_{21}(A_{11})^{-1}A_{12} \end{pmatrix}, \tag{21}$$

$$A - \widetilde{A} = \begin{pmatrix} 0 & 0 \\ 0 & [A/A_{11}] \end{pmatrix}. \tag{22}$$

The interpolation is exact for the 11-, 21- and 12-blocks, but not for the 22-block where the error is the Schur complement $[A/A_{11}]$. Since the latter depends on the inverse of the pivot matrix, a strategy for reducing the error is to choose the pivots such that $|\det A_{11}|$ is maximal —a criterion known as the *maximum volume principle* [35, 41]. Finding the pivots that satisfy the maximum volume principle is in general exponentially difficult but, as we shall see, there exist good heuristics that get close to this optimum in practice.

### 3.2.4 Relation with self-energy

In physics context, the Schur complement is closely related to the notion of self-energy, which appears in a non-interacting model by integrating out some degrees of freedom. Consider a Hamiltonian matrix

$$H = H_0 + V = \begin{pmatrix} H_{11} & 0 \\ 0 & H_{22} \end{pmatrix} + \begin{pmatrix} 0 & H_{12} \\ H_{21} & 0 \end{pmatrix}. \tag{23}$$

The Green's function at energy $E$ is defined as $G(E) = (E - H)^{-1}$. Its restriction to the 22-block is given by the Dyson equation,

$$[G(E)]_{22} = (E - H_{22} - \Sigma)^{-1}, \tag{24}$$

where $\Sigma = H_{21}(E - H_{11})^{-1}H_{12}$ is the so-called self-energy. The Dyson equation can be proven by applying Eq. (15) to $[G(E)]_{22} = [(E - H)^{-1}]_{22}$ and inserting the definition of the Schur complement, Eq. (12):

$$[G(E)]_{22} = [(E - H)^{-1}]_{22} = [(E - H)/(E - H)_{11}]^{-1}$$
$$= \Big[(E - H)_{22} - \underbrace{H_{21}[(E - H)_{11}]^{-1}H_{12}}_{\Sigma}\Big]^{-1}. \tag{25}$$

### 3.2.5 Restriction of the Schur complement

A trivial, yet important, property of the Schur complement is that the restriction of the Schur complement to a limited numbers of rows and columns is equal to the Schur complement of the full matrix restricted to those rows and columns (plus the pivots). More precisely, if $\mathcal{I}_1$

and $\mathcal{J}_1$ are the lists of pivots specifying the Schur complement and $\mathcal{I}_2$ and $\mathcal{J}_2$ are lists of rows and columns of interest, one has

$$[A(\mathcal{I},\mathcal{J})/A(\mathcal{I}_1,\mathcal{J}_1)](\mathcal{I}_2,\mathcal{J}_2) = [A(\mathcal{I}_1 \cup \mathcal{I}_2, \mathcal{J}_1 \cup \mathcal{J}_2)/A(\mathcal{I}_1,\mathcal{J}_1)], \qquad (26)$$

where $\mathcal{I}_1, \mathcal{I}_2 \subseteq \mathcal{I}$ and $\mathcal{J}_1, \mathcal{J}_2 \subseteq \mathcal{J}$. This property follows directly from the definition of the Schur complement.

## 3.3 Partial rank-revealing LU decomposition

In this section, we discuss partial rank-revealing LU (prrLU) decomposition. While mathematically equivalent to the CI decomposition, it is numerically more stable as the pivot matrices are never constructed nor inverted explicitly.

A matrix decomposition is *rank-revealing* when it allows the determination of the rank of the matrix: the decomposition $A = XDY$ is rank-revealing if both $X$ and $Y$ are well-conditioned and $D$ is diagonal. The rank is given by the number of non-zero entries on the diagonal of $D$. A well-known rank-revealing decomposition is SVD.

### 3.3.1 Default full search prrLU algorithm

The standard LU decomposition factorizes a matrix as $A = LDU$, where $L$ is lower-triangular, $D$ diagonal and $U$ upper-triangular [31]. It implements the Gaussian elimination algorithm for inverting matrices or solving linear systems of equations. The prrLU decomposition is an LU variant with two particular features: (i) It is *rank-revealing*: the largest remaining element, found by pivoting on both rows and columns, is used for the next pivot. (ii) It is *partial*: Gaussian elimination is stopped after constructing the first $\widetilde{\chi}$ columns of $L$ and rows of $U$, such that $LDU$ is a rank-$\widetilde{\chi}$ factorization of $A$.

The prrLU decomposition is computed using a fully-pivoted Gaussian elimination scheme, based on Eq. (13), which we reproduce here for convenience.

$$\begin{pmatrix} A_{11} & A_{12} \\ A_{21} & A_{22} \end{pmatrix} = \begin{pmatrix} \mathbb{1}_{11} & 0 \\ A_{21}A_{11}^{-1} & \mathbb{1}_{22} \end{pmatrix} \begin{pmatrix} A_{11} & 0 \\ 0 & [A/A_{11}] \end{pmatrix} \begin{pmatrix} \mathbb{1}_{11} & A_{11}^{-1}A_{12} \\ 0 & \mathbb{1}_{22} \end{pmatrix}. \qquad (27)$$

Note that the right side has a block $LDU$ structure. The algorithm utilizes this as follows. First, we permute the rows and columns of $A$ such that its largest element (in modulus) is positioned into the top left 11-position, then apply the above identity with a 11-block of size $1 \times 1$. Next, we repeat this procedure on the lower-right block of the second matrix on the right of Eq. (27) (hereafter, the "central" matrix), i.e. on $[A/1]$. We continue iteratively, yielding $[A/(1,2)]$, $[A/(1,2,3)]$, etc., thereby progressively diagonalizing the central matrix while maintaining the lower- and upper-triangular form of $L$ and $U$. Before each application of Eq. (27) we choose the largest element of the previous Schur complement as new pivot and permute it to the top left position of that submatrix. This strategy of maximizing the pivot improves the algorithm's stability, since it minimizes the inverse of the new pivot, which enters the left and right matrices [35,41] and corresponds to the maximum volume strategy over the new pivot, see Appendix B2 of [13]. After $\widetilde{\chi}$ steps we obtain a prrLU decomposition of the form

$$A = \begin{pmatrix} L_{11} & 0 \\ L_{21} & \mathbb{1}_{22} \end{pmatrix} \begin{pmatrix} D & 0 \\ 0 & [A/(1,\ldots,\widetilde{\chi})] \end{pmatrix} \begin{pmatrix} U_{11} & U_{12} \\ 0 & \mathbb{1}_{22} \end{pmatrix}. \qquad (28)$$

Here, $L_{11}$ and $U_{11}$ have diagonal entries equal to 1 and are lower- or upper-triangular, respectively, and $D$ (shorthand for $D_{11}$) is diagonal [31,42]. The block subscripts 11, 12, 21, 22 label blocks with row and column indices given by $\mathcal{I}_1 = \mathcal{J}_1 = \{1,\ldots,\widetilde{\chi}\}$, $\mathcal{I}_2 = \mathbb{I} \setminus \mathcal{I}_1$, and $\mathcal{J}_2 = \mathbb{J} \setminus \mathcal{J}_1$, where these indices refer to the *pivoted* version of the original $A$. When the Schur

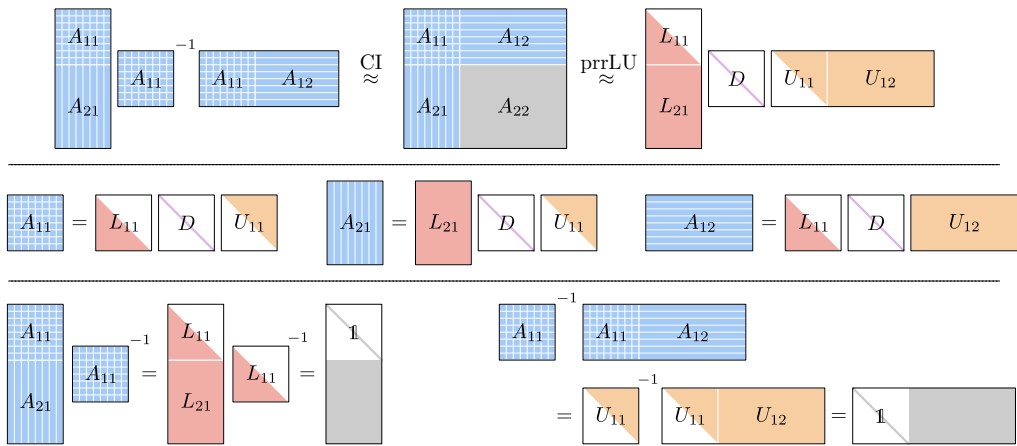

Figure 3: Equivalence between CI and prrLU. The prrLU decomposition provides all the matrices of the CI. Top: Eqs. (22) and (30); middle: Eqs. (32a-32c); bottom: Eqs. (32d-32e). White portions of matrices are equal to 0.

complement becomes zero, after $\chi$ steps, the scheme terminates, identifying $\chi$ as the rank of $A$.

Now, note that (for any $\widetilde{\chi} \leq \chi$) Eq. (28) can be recast into the form

$$A = LDU + \begin{pmatrix} 0 & 0 \\ 0 & [A/(1,\ldots,\widetilde{\chi})] \end{pmatrix}, \qquad L = \begin{pmatrix} L_{11} \\ L_{21} \end{pmatrix}, \qquad U = \begin{pmatrix} U_{11} & U_{12} \end{pmatrix}. \tag{29}$$

This precisely matches the CI formula (22). Again the Schur complement $[A/(1,\ldots,\widetilde{\chi})]$ is the error in the factorization. Thus, prrLU actually yields an CI [34, 42], given by

$$\widetilde{A} = LDU = \begin{pmatrix} L_{11} \\ L_{21} \end{pmatrix} D \begin{pmatrix} U_{11} & U_{12} \end{pmatrix}. \tag{30}$$

Explicit relations between the CI and prrLU representations are obtained from Eq. (30):

$$\begin{pmatrix} A_{11} \\ A_{21} \end{pmatrix} (A_{11})^{-1} \begin{pmatrix} A_{11} & A_{12} \end{pmatrix} = \begin{pmatrix} L_{11}DU_{11} \\ L_{21}DU_{11} \end{pmatrix} (L_{11}DU_{11})^{-1} \begin{pmatrix} L_{11}DU_{11} & L_{11}DU_{12} \end{pmatrix}, \tag{31}$$

where, abusing notation, $A_{xy} = A(\mathcal{I}_x, \mathcal{J}_y)$ now denote blocks of the pivoted version of the original $A$. This yields the following identifications, depicted schematically in Fig. 3:

$$A_{11} = P = L_{11}DU_{11}, \tag{32a}$$

$$A_{21} = L_{21}DU_{11}, \tag{32b}$$

$$A_{12} = L_{11}DU_{12}, \tag{32c}$$

$$\begin{pmatrix} A_{11} \\ A_{21} \end{pmatrix} (A_{11})^{-1} = \begin{pmatrix} \mathbb{1}_{11} \\ L_{21}L_{11}^{-1} \end{pmatrix}, \tag{32d}$$

$$(A_{11})^{-1} \begin{pmatrix} A_{11} & A_{12} \end{pmatrix} = \begin{pmatrix} \mathbb{1}_{11} & U_{11}^{-1}U_{12} \end{pmatrix}. \tag{32e}$$

The main advantage of prrLU over a direct CI is numerical stability, as we avoid the construction and inversion of ill-conditioned pivot matrices [31]. In our experience, prrLU is also more stable than the QR-stabilization approach to CI used in [13]. Furthermore, prrLU is updatable: new rows and columns can be added easily.

Let us note that the maximal pivot strategy of prrLU eliminates the largest contribution to the next Schur complement, hence reducing the CI error. Hence, it is a simple, greedy algorithm for constructing a near-maximum volume submatrix [42, 45].

### 3.3.2 Alternative pivot search methods: Full, rook or block rook

The above algorithm uses a *full search* for the pivots, i.e. it uses the information of the entire matrix $A$ and scales as $O(mn)$. It provides a quasi-optimal CI approximation but is expensive computationally as each new pivot is searched on the entire Schur complement $[A/(1,\dots,\widetilde{\chi})]$.

Rook search is a cheaper alternative, first proposed in [46,47]. (See Algorithm 2 of [12] and Ref. [13, Sec. III.B.3], where it was called *alternating* search). It explores the Schur complement $[A/(1,\dots,\widetilde{\chi})]$ by moving in alternating fashion along its rows and columns, similar to a chess rook. It searches along a randomly chosen initial column for the row yielding the maximum error, along that row for the column yielding the maximal error, and so on. The process terminates when a "rook condition is established", i.e. when an element is found that maximizes the error along both its row and column; that element is selected as new pivot. Compared to full pivoting, rook pivoting has the following useful properties: (i) computational cost reduced to $O[\max(m,n)]$ from $O(mn)$; (ii) comparable robustness [48]; (iii) almost as good convergence of the CI in practice.

---

**Algorithm 1:** Block rook pivoting search. Given pivot lists $\mathcal{I}, \mathcal{J}$, the algorithm updates the lists $\mathcal{I}, \mathcal{J}$ in place by alternating between searching for better pivots along the rows and columns in even or odd iterations, respectively. In each iteration, the pivot lists $\mathcal{I}, \mathcal{J}$ are updated with new, improved pivots (the 'rook move') from a prrLU decomposition with tolerance $\epsilon$ (line 8). The algorithm terminates when either the rook condition is met, i.e. when there are no better pivots along the available rows and columns, or when a maximum depth of $n_{\text{rook}}$ iterations has been reached (typically $n_{\text{rook}} \leq 5$). Upon exiting the algorithm, the updated lists $\mathcal{I}$ and $\mathcal{J}$ are of equal size.

**Input:** A matrix function $A$ with row indices $\mathbb{I}$ and column indices $\mathbb{J}$, initial pivot lists $\mathcal{I} \subseteq \mathbb{I}, \mathcal{J} \subseteq \mathbb{J}$ with $\chi$ elements each, and tolerance $\tau$.

**Output:** Updated pivot lists $\mathcal{I}, \mathcal{J}$ for the prrLU of $A$ with up to $2\chi$ elements each.

1   $\mathcal{J}' \leftarrow \mathcal{J} \cup \{\chi$ new random column indices $\in \mathbb{J} \setminus \mathcal{J}\}$
2   **for** $t \leftarrow 1$ **to** $n_{rook}$ **do**
3     **if** $t$ *is odd* **then**
4       search among the columns: set $B \leftarrow A(\mathbb{I}, \mathcal{J}')$
5     **else**
6       search among the rows: set $B \leftarrow A(\mathcal{I}', \mathbb{J})$
7     **end**
8     find new pivots: $(\mathcal{I}', \mathcal{J}') \leftarrow$ pivots of $\text{prrLU}_\epsilon(B)$
9     **if** $\mathcal{I}' = \mathcal{I}$ *and* $\mathcal{J}' = \mathcal{J}$ **then** rook condition has been established.
10       **return** $\mathcal{I}, \mathcal{J}$
11     **else**
12       update the pivots: $(\mathcal{I}, \mathcal{J}) \leftarrow (\mathcal{I}', \mathcal{J}')$
13     **end**
14 **end**

---

We now introduce *block rook search*. It is a variant of rook search which searches for all pivots simultaneously. It is useful in the common situation that a CI of a matrix $A(\mathbb{I}, \mathbb{J})$ has been obtained and then this matrix is extended to a larger matrix $A(\mathbb{I}', \mathbb{J}')$ by adding some new rows and columns. One needs to construct a new set of pivots $\mathcal{I}'$ and $\mathcal{J}'$. The previous set of pivots $\mathcal{I}$ and $\mathcal{J}$ is a very good starting point that one wishes to leverage on to construct this new set. Block rook search is described in Algorithm 1.

To find pivots, the algorithm uses a series of prrLU, applied to a subset of rows and columns in alternating fashion. It starts with a set of columns made of previously found pivots and some random ones. It then LU factorizes the corresponding sub-matrix to yield new pivot rows and columns. The algorithm is repeated, alternatingly on rows and columns, until convergence (or up to $n_{\mathrm{Rook}}$ times). In practice, we observe that $n_{\mathrm{Rook}} = 3$ is often sufficient to reach convergence. At convergence, the pivots satisfy rook conditions as if they had been sequentially found by rook search (see App. A.2 for a proof). The algorithm requires $\mathcal{O}(n_{\mathrm{Rook}}\chi^2 \max(m, n))$ to factorize the matrix $A$.

# 4 Tensor cross interpolation

We now turn to the tensor case. After introducing the TCI form of an MPS, we present the TCI algorithm and its variants. Although this section is self-contained, it is somewhat compact and we recommend users new to TCI to read a more pedagogical introduction first, such as section III of [13]. Important proofs can also be found in the appendices of [13] and/or in the mathematical literature [8–12, 18, 19, 49, 50].

The algorithm used by some of us previously (e.g. in [13, 15, 16]) will be referred to as the 2-site TCI algorithm in *accumulative mode*. Below, we introduce a number of new algorithms that evolved from this original one. Our default TCI (discussed first, in section 4.3.1) is the 2-site TCI algorithm in *reset* mode. We also introduce a 1-site TCI, a 0-site TCI and a CI-canonical algorithm and explain their specific use cases.

## 4.1 TCI form of tensor trains

Tensor trains obtained from TCI decompositions of an input tensor $F_{\boldsymbol{\sigma}}$ have a very particular, characteristic form, called *TCI form*. It is obtained, e.g., through repeated use of the CI approximation, as discussed informally in Sec. III.B.1 of [13]. Its defining characteristic is that it is built *only* from one-dimensional slices of $F_{\boldsymbol{\sigma}}$ (on which all tensor indices $\sigma_\ell$ but one are fixed). Furthermore, TCI algorithms construct the TCI form using only *local* updates of these slices, as discussed in later sections.

The most difficult part of implementing TCI algorithms lies in the book-keeping of various lists of indices. This is facilitated by the introduction of the following notations.

- An external index $\sigma_\ell$ ($\ell \in \{1, 2, \ldots, \mathcal{L}\}$) takes $d_\ell$ different values from a set $\mathbb{S}_\ell$.

- $\mathbb{I}_\ell = \mathbb{S}_1 \times \cdots \times \mathbb{S}_\ell$ denotes the set of *row multi-indices* up to site $\ell$. An element $i \in \mathbb{I}_\ell$ is a row multi-index taking the form $i = (\sigma_1, \ldots, \sigma_\ell)$.

- $\mathbb{J}_\ell = \mathbb{S}_\ell \times \cdots \times \mathbb{S}_{\mathcal{L}}$ denotes the set of *column multi-indices* from site $\ell$ upwards. An element $j \in \mathbb{J}_\ell$ is a column multi-index taking the form $j = (\sigma_\ell, \ldots, \sigma_{\mathcal{L}})$.

- $\mathbb{I}_{\mathcal{L}} = \mathbb{J}_1$ is the full configuration space. A full configuration $\boldsymbol{\sigma} \in \mathbb{I}_{\mathcal{L}}$ takes the form $\boldsymbol{\sigma} = (\sigma_1, \ldots, \sigma_{\mathcal{L}})$.

- $i_\ell \oplus j_{\ell+1} \equiv (\sigma_1, \ldots, \sigma_{\mathcal{L}})$ denotes the concatenation of complementary multi-indices.

For each $\ell$, we define a list of "pivot rows" $\mathcal{I}_\ell \subseteq \mathbb{I}_\ell$ and a list of "pivot columns" $\mathcal{J}_\ell \subseteq \mathbb{J}_\ell$. We also define $\mathcal{I}_0 = \mathcal{J}_{\mathcal{L}+1} = \{()\}$, where () is an empty tuple. Note that $\mathcal{I}_\ell$ and $\mathcal{J}_\ell$ are lists of lists of external $\sigma$ indices. Through the pivot rows and pivot columns, we define zero-, one-, and two-dimensional slices of the tensor $F$, where a $k$-dimensional slice has $k$ free indices, as follows.

- A pivot matrix $P_\ell$ is a zero-dimensional slice of the input tensor $F$:

$$[P_\ell]_{ij} = F_{i\oplus j} = \qquad \qquad \tag{33a}$$

for $i \in \mathcal{I}_\ell$ and $j \in \mathcal{J}_{\ell+1}$, or in Matlab notation, $P_\ell = F(\mathcal{I}_\ell, \mathcal{J}_{\ell+1})$. The two pivot lists have the same number of elements; $P_\ell$ is a square matrix of dimension $\chi_\ell = |\mathcal{I}_\ell| = |\mathcal{J}_{\ell+1}|$ and we will choose the pivots such that $\det P_\ell \neq 0$.

- A 3-leg T-tensor $T_\ell$ is a one-dimensional slice of $F$:

$$[T_\ell]_{i\sigma j} \equiv F_{i\oplus(\sigma)\oplus j} = \qquad \qquad \tag{33b}$$

for $i \in \mathcal{I}_{\ell-1}$, $\sigma \in \mathbb{S}_\ell$ and $j \in \mathcal{J}_{\ell+1}$, or $T_\ell \equiv F(\mathcal{I}_{\ell-1}, \mathbb{S}_\ell, \mathcal{J}_{\ell+1})$. For specified $\sigma$, the matrix $T_\ell^\sigma$ is defined as $[T_\ell^\sigma]_{ij} \equiv [T_\ell]_{i\sigma j}$.

- A 4-leg $\Pi$-tensor $\Pi_\ell$ is a two-dimensional slice of $F$:

$$[\Pi_\ell]_{i\sigma\sigma' j} \equiv F_{i\oplus(\sigma,\sigma')\oplus j} = \qquad \qquad \tag{33c}$$

for $i \in \mathcal{I}_{\ell-1}$, $\sigma \in \mathbb{S}_\ell$, $\sigma' \in \mathbb{S}_{\ell+1}$ and $j \in \mathcal{J}_{\ell+2}$, or $\Pi_\ell \equiv F(\mathcal{I}_{\ell-1}, \mathbb{S}_\ell, \mathbb{S}_{\ell+1}, \mathcal{J}_{\ell+2})$.

With these definitions, the TCI approximation $\widetilde{F}$ of $F$ is defined as

$$F_{\boldsymbol{\sigma}} \approx \widetilde{F}_{\boldsymbol{\sigma}} = T_1^{\sigma_1} P_1^{-1} \cdots T_\ell^{\sigma_\ell} P_\ell^{-1} T_{\ell+1}^{\sigma_{\ell+1}} \cdots P_{\mathcal{L}-1}^{-1} T_{\mathcal{L}}^{\sigma_{\mathcal{L}}}, \tag{34}$$

with independent summations over all $i_\ell \in \mathcal{I}_\ell$ and all $j_{\ell+1} \in \mathcal{J}_{\ell+1}$, for $\ell = 1, \ldots, \mathcal{L}-1$. Here, ◆ represents $P_\ell^{-1}$, the inverse of a pivot matrix, and ⊡ represents a $T$-tensor $T_\ell$. Such a tensor cross interpolation is entirely defined by the $T$ and $P$ tensors, i.e. by slices of $F$. In other words, if one (i) knows the pivot lists $\{\mathcal{I}_\ell, \mathcal{J}_{\ell+1} | \ell = 1, \ldots, \mathcal{L}-1\}$ and (ii) can compute $F_{\boldsymbol{\sigma}}$ for any given $\boldsymbol{\sigma}$, then one can construct $\widetilde{F}$. Equation (34) defines a genuine tensor train with rank $\chi = \max \chi_\ell$. Its form matches Eq. (1) with the identification $T_\ell P_\ell^{-1} = M_\ell$.

Equation (34) defines the *TCI form*, which is fully specified by two ingredients: (i) the sets of rows $\mathcal{I}_\ell$ and columns $\mathcal{J}_\ell$, and (ii) the corresponding values (slices) $T_\ell$ and $P_\ell$ of the input tensor $F_{\boldsymbol{\sigma}}$. Any tensor train can be converted exactly to a TCI form (see Sec. 4.5.1).

## 4.2 Nesting conditions

TCI algorithm relies on an important property of the pivot lists $\mathcal{I}_\ell$ and $\mathcal{J}_\ell$ that we now discuss, the *nesting conditions*. By definition, for any $\ell$:

- $\mathcal{I}_\ell$ is nested with respect to $\mathcal{I}_{\ell-1}$, denoted by $\mathcal{I}_{\ell-1} < \mathcal{I}_\ell$, if $\mathcal{I}_\ell \subseteq \mathcal{I}_{\ell-1} \times \mathbb{S}_\ell$, or equivalently, if removing the last index of any element of $\mathcal{I}_\ell$ yields an element of $\mathcal{I}_{\ell-1}$. $\mathcal{I}_{\ell-1} < \mathcal{I}_\ell$ implies that the pivot matrix $P_\ell$ is a slice of $T_\ell$.

- $\mathcal{J}_\ell$ is nested with respect to $\mathcal{J}_{\ell+1}$, denoted by $\mathcal{J}_\ell > \mathcal{J}_{\ell+1}$, if $\mathcal{J}_\ell \subseteq \mathbb{S}_\ell \times \mathcal{J}_{\ell+1}$, or equivalently, if removing the first index of any element of $\mathcal{J}_\ell$ yields an element of $\mathcal{J}_{\ell+1}$. $\mathcal{J}_\ell > \mathcal{J}_{\ell+1}$ implies that the pivot matrix $P_{\ell-1}$ is a slice of $T_\ell$.

Table 1: Example for a fully nested configuration of the pivot lists $\mathcal{I}_\ell$ and $\mathcal{J}_\ell$ for a TCI with 5 local indices $\sigma_1, \ldots, \sigma_5 \in \{0, 1\}$. Pivot lists that belong to the same bond are shown in the same row.

| $\ell$ | $\mathcal{I}_\ell$ | $\mathcal{J}_{\ell+1}$ |
|---|---|---|
| 1 | $\mathcal{I}_1 = ((1))$ | $\mathcal{J}_2 = ((1,0,0,1))$ |
| 2 | $\mathcal{I}_2 = ((1,0),(1,1))$ | $\mathcal{J}_3 = ((0,0,1),(1,0,1))$ |
| 3 | $\mathcal{I}_3 = ((1,1,0),(1,0,1))$ | $\mathcal{J}_4 = ((0,1),(1,1))$ |
| 4 | $\mathcal{I}_4 = ((1,1,0,0))$ | $\mathcal{J}_5 = ((1))$ |

We say that the pivots are:

- *left-nested* up to $\ell$ if

$$\mathcal{I}_0 < \mathcal{I}_1 < \cdots < \mathcal{I}_\ell, \tag{35}$$

- *right-nested* up to $\ell$ if

$$\mathcal{J}_\ell > \mathcal{J}_{\ell+1} > \cdots > \mathcal{J}_{\mathcal{L}+1}, \tag{36}$$

- *fully left-nested* if they are left-nested up to $\mathcal{L}-1$, *fully right-nested* if they are right-nested up to 2. When the pivots are both fully left- *and* right-nested they are said to be *fully nested*, i.e. one has

$$\mathcal{I}_0 < \mathcal{I}_1 < \cdots < \mathcal{I}_{\mathcal{L}-1}, \qquad \mathcal{J}_2 > \mathcal{J}_{\ell+2} > \cdots > \mathcal{J}_{\mathcal{L}+1}. \tag{37}$$

The importance of nesting conditions stems from the fact that they provides some interpolation properties. We refer to Ref. [13] or Appendix A.3 for the associated proofs. In particular, if the pivots are left-nested up to $\ell - 1$ and right-nested up to $\ell + 1$ (we say *nested w.r.t.* $T_\ell$) then the TCI form is exact on the one-dimensional slice $T_\ell$:

$$\widetilde{F}_{i \oplus (\sigma) \oplus j} = [T_\ell]_{i \sigma j} = F_{i \oplus (\sigma) \oplus j}, \quad \forall i \in \mathcal{I}_{\ell-1}, \sigma \in \mathbb{S}_\ell, j \in \mathcal{J}_{\ell+1}. \tag{38}$$

It follows that if the pivots are fully nested, then the TCI form is exact on every $T_\ell$ and $P_\ell$, i.e. on all slices used to construct it. Hence, it is an interpolation.

An example for a fully nested configuration of the pivot lists $\mathcal{I}_\ell$ and $\mathcal{J}_\ell$ for a TCI with 5 local indices $\sigma_1, \ldots, \sigma_5 \in \{0, 1\}$ is shown in Table 1. Full nesting could be broken for example by adding $(0,0)$ to $\mathcal{I}_2$, or by adding $(1,1,0)$ to $\mathcal{J}_3$.

## 4.3 2-site TCI algorithms

The goal of TCI algorithms is to obtain a TCI approximation of a given tensor $F$ at a specified tolerance $\|F - \widetilde{F}\|_\infty < \tau$ (over the maximum norm), by finding a minimal set of suitable pivots. In this section, we present various 2-site TCI algorithms and discuss their variants and options. They are all based on the fact that the TCI form (34) (with fully nested pivots) is exact on all one-dimensional slices $T_\ell$ but not on the two-dimensional slices $\Pi_\ell$. All 2-site TCI algorithms thus aim to iteratively improve the representation of the $\Pi_\ell$ slices.

### 4.3.1 Basic algorithm

We start by presenting a TCI algorithm in a version based on LU factorization. In Sec. 4.3.2 we will describe its connection to the algorithm based on CI factorizations presented in prior work [12, 13]. The algorithm proceeds as follows:

(1) Start with an index $\hat{\sigma}$ for which $F_{\hat{\sigma}} \neq 0$, and construct initial pivots from it:
$\mathcal{I}_\ell = \{(\hat{\sigma}_1, \ldots, \hat{\sigma}_\ell)\}$ and $\mathcal{J}_{\ell+1} = \{(\hat{\sigma}_{\ell+1}, \ldots, \hat{\sigma}_\mathcal{L})\}$ for all $\ell$.

(2) Sweeping back and forth over $\ell = 1, \ldots, \mathcal{L}-1$, perform the following update at each $\ell$:

- Construct the $\Pi_\ell$ tensor (33c).

- View the tensor $\Pi_\ell$ as a matrix $F(\mathcal{I}_{\ell-1} \times \mathbb{S}_\ell, \mathbb{S}_{\ell+1} \times \mathcal{J}_{\ell+2})$ and perform its prrLU decomposition which approximates it as $\Pi_\ell \approx \widetilde{\Pi}_\ell$ with

$$[\Pi_\ell]_{i_{\ell-1}\sigma_\ell\sigma_{\ell+1}j_{\ell+2}} \approx [T_\ell'^{\sigma_\ell}]_{i_{\ell-1}j'_{\ell+1}} (P_\ell')^{-1}_{j'_{\ell+1}i'_\ell} [T_\ell'^{\sigma_{\ell+1}}]_{i'_\ell j_{\ell+2}}, \tag{39}$$

$$\underset{i_{\ell-1}\ \ \sigma_\ell\ \ \sigma_{\ell+1}\ \ j_{\ell+2}}{\boxed{\ \ \Pi_\ell\ \ }} \approx \underset{i_{\ell-1}\ \ \sigma_\ell\ j'_{\ell+1}}{\boxed{T_\ell'}}\ \underset{\ \ }{\overset{P_\ell'^{-1}}{\diamond}}\ \underset{i'_\ell\ \ \sigma_{\ell+1}\ \ j_{\ell+2}}{\boxed{T_{\ell+1}'}},$$

where $i'_\ell \in \mathcal{I}'_\ell \subset \mathcal{I}_{\ell-1} \times \mathbb{S}_\ell$ and $j'_{\ell+1} \in \mathcal{J}'_{\ell+1} \subset \mathbb{S}_{\ell+1} \times \mathcal{J}_{\ell+2}$ are the new pivots.

- Replace the old pivot lists $\mathcal{I}_\ell, \mathcal{J}_{\ell+1}$ by the new ones $\mathcal{I}'_\ell, \mathcal{J}'_{\ell+1}$. By construction, the nesting conditions $\mathcal{I}_{\ell-1} < \mathcal{I}'_\ell, \mathcal{J}'_{\ell+1} > \mathcal{J}_{\ell+2}$ are satisfied. The matrices $P_\ell$, $T_\ell$ and $T_{\ell+1}$ are also updated along with the pivots, according to their definitions (33a, 33b). Note that this step may break the full nesting condition: one may have $\mathcal{I}_\ell < \mathcal{I}_{\ell+1}$ but not $\mathcal{I}'_\ell < \mathcal{I}_{\ell+1}$; similarly, one may have $\mathcal{J}_\ell > \mathcal{J}_{\ell+1}$ but not $\mathcal{J}_\ell > \mathcal{J}'_{\ell+1}$.

(3) Iterate step (2) until the specified tolerance is reached, or a specified number of times.

When pivots are left-nested up to $\ell - 1$ and right-nested up to $\ell + 2$ — a property that our algorithm actually preserves — (we say that the tensor train is *nested w.r.t.* $\Pi_\ell$), then the following crucial relation holds (for a proof, see [13, App. C.2], or our App. A.3):

$$\left[\Pi_\ell - \widetilde{\Pi}_\ell\right]_{i_{\ell-1}\sigma_\ell\sigma_{\ell+1}j_{\ell+2}} = \left[F - \widetilde{F}\right]_{i_{\ell-1}\sigma_\ell\sigma_{\ell+1}j_{\ell+2}}, \tag{40}$$

for all $\sigma_\ell, \sigma_{\ell+1}$. Thus, the error made by approximating the local tensor $\Pi_\ell$ by its prrLU decomposition $\widetilde{\Pi}_\ell$ is also the error, on this two-dimensional slice, of approximating $F_\sigma$ by the TCI decomposition $\widetilde{F}_\sigma$. By construction, the TCI form (34) (with fully nested pivots) is exact on *one-dimensional slices*, $\mathcal{I}_{\ell-1} \times \mathbb{S}_\ell \times \mathcal{J}_{\ell+1}$, but not on the *two-dimensional slices* $\mathcal{I}_{\ell-1} \times \mathbb{S}_\ell \times \mathbb{S}_{\ell+1} \times \mathcal{J}_{\ell+2}$. Hence, the algorithm chooses the pivots in order to minimize the error on the latter.

The algorithm presented in this section deviates significantly from the one used by some of us in Ref. [12, 13]: there, new pivots could be added but they were never removed in order to maintain the full nesting condition. However, a close examination of [13, App. C.2] shows that partial nesting is sufficient to ensure Eq. (40). We use this fact to use an update strategy where the pivots $\mathcal{I}'_\ell, \mathcal{J}'_{\ell+1}$ are reset at each step (2) of the algorithm. The ability to discard "bad" pivots (e.g. ones found in early iterations that later turn out to be suboptimal) significantly improves the numerical stability of the present TCI algorithm compared to the original one [12]. This point will be discussed further in Sec. 4.3.3. If desired, full nesting can be restored at the end using 1-site TCI, discussed in Sec. 4.4.

### 4.3.2 CI vs prrLU

The TCI algorithm as described in this paper is also different from the standard TCI algorithm [12, 13] in that it uses prrLU instead of the CI decomposition for the $\Pi_\ell$ tensor. While CI and prrLU are equivalent, as shown in Sec. 3.3, the prrLU yields a more stable implementation, as it avoids inverting the pivot matrices $P$, which may become ill-conditioned. We emphasize again that we have found prrLU to be more efficient and stable than the alternative QR approach used in Appendix B of [13] to address the conditioning issue of the pivot matrices.

For convenience, we explicitly rewrite the correspondence between CI and LU factorization shown in Eqs. (32) as appropriate for the update of $\widetilde{\Pi}_\ell$:

$$\widetilde{\Pi}_\ell = T_\ell (P_\ell)^{-1} T_{\ell+1} = LDU = \begin{pmatrix} P_\ell & L_{11}DU_{12} \\ L_{21}DU_{11} & L_{21}DU_{12} \end{pmatrix}, \tag{41a}$$

$$P_\ell = L_{11}DU_{11}, \tag{41b}$$

$$T_\ell = \begin{pmatrix} L_{11}DU_{11} \\ L_{21}DU_{11} \end{pmatrix}, \qquad T_\ell P_\ell^{-1} = \begin{pmatrix} \mathbb{1} \\ L_{21}L_{11}^{-1} \end{pmatrix}, \tag{41c}$$

$$T_{\ell+1} = \begin{pmatrix} L_{11}DU_{11} & L_{11}DU_{12} \end{pmatrix}, \qquad P_\ell^{-1} T_{\ell+1} = \begin{pmatrix} \mathbb{1} & U_{11}^{-1}U_{12} \end{pmatrix}. \tag{41d}$$

Since $U_{11}$ and $L_{11}$ are triangular matrices, the two terms involving a matrix inversion can be computed in a stable manner using forward/backward substitution.

### 4.3.3 Pivot update method: Reset vs accumulative

In order to update the pivots in the TCI algorithm, we can use two different methods, which we call *reset* and *accumulative*.

- In reset mode, we recompute the full prrLU decomposition of $\Pi_\ell$ at each $\ell$, hence reconstructing new pivots $\mathcal{I}_\ell, \mathcal{J}_{\ell+1}$. This version was presented in Sec. 4.3.1.

- In accumulative mode, we update the pivot lists $\mathcal{I}_\ell, \mathcal{J}_{\ell+1}$ by only *adding* pivots. Typically, pivots are added one at a time, thereby increasing $\chi_\ell$ to $\chi_\ell+1$. Once a pivot has been added, it is never removed. This strategy preserves full nesting, thus ensuring the interpolation property of the TCI approximation. This is the method presented in Ref. [12, algorithm #5].

The main advantage of reset mode is that it eliminates bad pivots which are almost linearly dependent, thereby leading to poorly conditioned $P$ matrices. These occur when the algorithm first explores configurations where $F_\sigma$ is small and only later discovers other configurations with larger values of $F_{\sigma'}$. In such cases, the late pivots correspond to a much larger absolute value of $F$ than the first, leading to ill-conditioned $P_\ell$. Therefore, in accumulative mode, it is crucial to choose as an initial pivot a point where $F$ is of the same order of magnitude as its maximum. In reset mode, the bad pivots are automatically eliminated, which yields a better TCI approximation and very stable convergence. On the other hand, accumulative mode requires a (slightly) smaller number of values of $F$, as the exploration of configurations for finding pivots is kept to a minimum.

The runtime of both approaches scales as $O(\chi^3)$. Accumulative mode requires $O(\chi^2)$ per update and $\chi$ updates to reach a rank of $\chi$. Reset mode requires $O(\chi^3)$ for each update, but typically converges within a small number of updates independently of $\chi$.

We note that the pioneering work of Ref. [10] used a method similar to reset mode, recalculating the pivots at each step. MPS recompression was performed very differently, however, using a combination of SVD and the maximum volume principle, which led to slower scaling. Here, pivot optimization is done entirely within the LU decomposition.

### 4.3.4 Pivot search method: Full, rook or block rook

A crucial component of 2-site TCI algorithms is the search for pivots, as the largest elements of the error tensor $|\Pi_\ell - \widetilde{\Pi}_\ell|$. As discussed in Sec. 3.3.2, three different search modes are available: *Full search* is the simplest and most stable mode, but also most expensive, scaling as $\mathcal{O}(d^2)$. *Rook search* is a cheaper alternative, scaling as $\mathcal{O}(d)$ (since rows and columns are

explored alternatingly), and is almost as good in practice. Rook search is well adapted to accumulative mode [12] and is advantageous when the dimension $d$ is large.

*Block rook search* is especially useful when used with reset pivot update mode. Indeed, it allows reusing previously found pivots and therefore reusing previously computed values of $F$. This is particularly useful when $F_\sigma$ is an expensive function to evaluate on $\sigma$. The algorithm requires $\mathcal{O}(n_{Rook}\chi^2 d)$ function evaluations to factorize a $\Pi$ tensor.

### 4.3.5  Proposing pivots from outside of TCI

In its normal mode, TCI constructs new pivots by making local updates of existing pivots. In several situations, it is desirable to enrich the pivot search by proposing a list of values of the indices $\sigma$ which the TCI algorithm is required to try as pivots. It is a way to incorporate prior knowledge about $F$ into TCI. We call such values of $\sigma$ global pivots. This section discusses our strategy to perform this operation in a stable way.

Given a list of global pivots, we split each index $\sigma$ as $\sigma = i_\ell \oplus j_{\ell+1}$ for all $\ell = 1, \dots \mathcal{L}-1$, and $i_\ell$ and $j_{\ell+1}$ are added to the corresponding pivot lists $\mathcal{I}_\ell$ and $\mathcal{J}_{\ell+1}$. This operation preserves nesting conditions. Next, we perform a prrLU decomposition of the pivot matrices $P_\ell$ to remove possible spurious pivots. Last, we perform a few sweeps using 2-sites TCI in reset mode to stabilize the pivots lists. We provide a simple example of global pivot addition in Appendix B.3.5.

Global pivot proposals can be useful in several situations. First, the TCI algorithm can experience some ergodicity issues as discussed in Sec. 4.3.6, which can be solved by adding some pivots explicitly. The construction of the Matrix Product Operators discussed in Section 7 belongs to this category. Second, the TCI decomposition of a tensor $F_2$ close to another $F_1$ for which the TCI is already known, e.g. due to an adiabatic change of some parameter, can benefit from initialization with the pivots of $\widetilde{F}_1$. Third, global pivot proposal can be used to separate the exploration of the configuration space (the way these global pivots are constructed) from the algorithm used to update the tensor train. For instance, one could use a separate algorithm to globally look for pivots where the TCI error is large using a separate global optimizer; then propose these pivots to TCI; and iteratively repeat the process until convergence.

The above algorithm, which we call *StrictlyNested*, works well but suffers from one (albeit relatively rare) problem: it occasionally discards perfectly valid proposed global pivots. This may happen when $\chi_\ell$ depends on $\ell$ in such a manner that the MPS has a "constriction", i.e. a bond with a smaller dimension $\chi_\ell$ than all others. Upon sweeping through this bond, some pivots will be deleted (which is fine), but that deletion will propagate upon continuing to sweep (which is a weakness of the algorithm).

A simple fix is to construct an *enlarged* tensor $\bar{\Pi}_\ell$ that extends $\Pi_\ell$ with additional rows and columns containing deleted pivots, thus retaining these for consideration as potential pivots. Concretely, denoting pivots obtained in a previous sweep by $\bar{\mathcal{I}}_\ell$ and $\bar{\mathcal{J}}_\ell$, we define

$$\bar{\Pi}_\ell = F([\mathcal{I}_{\ell-1} \times \mathbb{S}_\ell] \cup \bar{\mathcal{I}}_\ell \, , \, [\mathbb{S}_{\ell+1} \times \mathcal{J}_{\ell+2}] \cup \bar{\mathcal{J}}_{\ell+1}), \tag{42}$$

and use $\bar{\Pi}_\ell$ instead of $\Pi_\ell$ for the prrLU decomposition. We note that such enlargements can break nesting conditions, i.e. this is an *UnStrictlyNested* mode. However, we have not observed this to cause any problems in our numerical experiments.

### 4.3.6  Ergodicity

The construction of tensor trains using TCI is based on the exploration of configuration space. In analogy with what can happen with Monte Carlo techniques, this exploration may encounter ergodicity problems, remaining stuck in a subpart of the configuration space and not visiting other relevant parts. Examples where this may occur include: very sparse tensors $F_\sigma$, where

TCI might miss some nonzero entries (see the Matrix Product Operator construction section 7 for an example); tensors with discrete symmetries, where the exploration may remain in one symmetry sector (relevant for the partition function of the Ising model, see Sec. 5.3); or multivariate functions with very narrow peaks.

All ergodicity problems that we have encountered so far could be fixed by proposing global pivots, as described in Sec. 4.3.5. For sparse tensors, one feeds the algorithm with a list of nonzero entries. For discrete symmetries, one initializes the algorithm with one configuration per symmetry sector. One could also consider more elaborate strategies that use a dedicated algorithm to explore new configurations, in analogy to the construction of complex moves when building a Monte Carlo algorithm. In fact, existing Monte Carlo algorithms could be used directly as way to propose global pivots. Such an algorithm would separate entirely the pivot exploration strategy from the way the tensor train is updated.

Let us illustrate the above ideas with a toy example. Consider a fermionic operator $c$ ($c^\dagger$) that destroys (creates) an electron on a unique site ($\{c, c\} = \{c^\dagger, c^\dagger\} = 0; \{c, c^\dagger\} = 1$). We want to factorize

$$F_{\boldsymbol{\sigma}} = \langle a_{\sigma_1} \cdots a_{\sigma_{\mathcal{L}}} \rangle, \tag{43}$$

into a tensor train, where $a_0 = c$ and $a_1 = c^\dagger$ and the average is taken with respect to the state $\frac{1}{\sqrt{2}}|0\rangle + \frac{1}{\sqrt{2}}c^\dagger|0\rangle$. For even $\mathcal{L}$, this tensor has only two non-zero elements, namely $F_{\boldsymbol{\sigma}} = 1/2$ for $\boldsymbol{\sigma}_1 = (1, 0, 1, 0, \ldots, 1, 0)$ and $\boldsymbol{\sigma}_2 = (0, 1, 0, 1, \ldots, 0, 1)$. This is due to the fermionic algebra, which implies $a_0 a_0 = cc = 0$ and $a_1 a_1 = c^\dagger c^\dagger = 0$. Using TCI in a standard way with one of the two elements as the starting pivot, TCI fails to find the second one. The reason is that the TCI updates are local, thus TCI quickly (wrongly) concludes that it correctly describes all configurations, whereas it correctly describes only the configurations that it has seen. A simple cure is to propose both $\boldsymbol{\sigma}_1$ and $\boldsymbol{\sigma}_2$ as global pivots. This works and is the easiest solution when the important configurations are known. An alternative cure is to *enlarge the configuration space* to obtain a larger but less sparse tensor. This idea is analogous to the concept of worms in Monte Carlo, where the configuration space is enlarged to remove constrains and allow for non-local updates. Here, we enlarge the local dimension from $d = 2$ to $d = 3$ by adding identity as a third operator, $a_2 = 1$. The new tensor is much less sparse and is correctly reconstructed using TCI with $(2, 2, \ldots, 2)$ as initial pivot. Restricting the resulting tensor train to $\sigma_i \in \{0, 1\}$ yields the correct factorization.

### 4.3.7 Error estimation: Bare vs. environment

In the prrLU decomposition of the $\Pi_\ell$ tensor described in Sec. 4.3.1 above, each new pivot is chosen in order to minimize the *bare error* $|\Pi_\ell - \widetilde{\Pi}_\ell|_{i_{\ell-1}\sigma_\ell\sigma_{\ell+1}j_{\ell+2}}$. An alternative choice is to define an *environment error* whose minimization aims to find the best approximation of the "integrated" tensor $\sum_{\boldsymbol{\sigma}} F_{\boldsymbol{\sigma}}$, i.e. summed over all external indices (see Sec. III.B.4 of Ref. [13]). The environment error has the form $|L_{i_{\ell-1}} R_{j_{\ell+2}}| |\Pi_\ell - \widetilde{\Pi}_\ell|_{i_{\ell-1}\sigma_\ell\sigma_{\ell+1}j_{\ell+2}}$, with left and right environment tensors defined as

$$L_{i_{\ell-1}} = \sum_{\sigma_1, \ldots, \sigma_{\ell-1}} [T_1^{\sigma_1} P_1^{-1} \cdots T_{\ell-1}^{\sigma_{\ell-1}} P_{\ell-1}^{-1}]_{1 i_{\ell-1}}, \qquad R_{j_{\ell+2}} = \sum_{\sigma_{\ell+2}, \ldots, \sigma_{\mathcal{L}}} [P_{\ell+1}^{-1} T_{\ell+2}^{\sigma_{\ell+2}} \cdots P_{\mathcal{L}-1}^{-1} T_{\mathcal{L}}^{\sigma_{\mathcal{L}}}]_{j_{\ell+2} 1}. \tag{44}$$

Minimization of the environment error can be very efficient for the computation of integrals involving integrands with long tails. An example of improved accuracy using this *environment mode* is given in Fig. 7 of Ref. [13].

### 4.4 The 1-site and 0-site TCI algorithms

In this section, we propose two more algorithms complementing 2-site TCI: the 1-site and 0-site TCI algorithms. The names reflect the number $\sigma$-indices of the objects decomposed with

LU: $\Pi$, $T$ or $P$ tensors with 2, 1 or 0 $\sigma$-indices, respectively. The 2-site algorithms described above are more versatile, and only they can increase the bond dimension $\chi_\ell$, so they are almost always needed during the initial learning stage (unless global pivots are used to start with a large enough rank). However, the 1-site and 0-site TCI algorithms are faster than 2-site TCI, and the former can also be used to achieve full nesting.

### 4.4.1 The 1-site TCI algorithm

The 1-site TCI algorithm sweeps through the tensor train and compresses its $T$ tensors using prrLU. In a forward sweep we view $T_\ell$ as a matrix with indices $(\mathcal{I}_{\ell-1} \times \mathbb{S}_\ell, \mathcal{J}_{\ell+1})$, regrouping the $\sigma_\ell$ index with the left index $i_{\ell-1}$. Using prrLU, we obtain new pivots $\mathcal{I}'_\ell$, $\mathcal{J}'_{\ell+1}$ to replace $\mathcal{I}_\ell$, $\mathcal{J}_{\ell+1}$, satisfying $\mathcal{I}'_\ell > \mathcal{I}_{\ell-1}$ and $\mathcal{J}'_{\ell+1} \subseteq \mathcal{J}_{\ell+1}$, and update $T_\ell$, $P_\ell$ and $T_{\ell+1}$ accordingly. After the forward sweep, the pivots are fully left-nested, i.e. $\mathcal{I}_0 < \cdots < \mathcal{I}_{\mathcal{L}-1}$.

In a backward sweep, $T_\ell$ is viewed as a matrix with indices $(\mathcal{I}_{\ell-1}, \mathbb{S}_\ell \times \mathcal{J}_{\ell+1})$, so prrLU yields new pivots $\mathcal{I}'_{\ell-1} \subseteq \mathcal{I}_{\ell-1}$, $\mathcal{J}'_\ell > \mathcal{J}_{\ell+1}$, and corresponding updates of $T_\ell$, $P_{\ell-1}$ and $T_{\ell-1}$. After the backward sweep, the pivots are fully right-nested, i.e. $\mathcal{J}_2 > \cdots > \mathcal{J}_{\mathcal{L}+1}$, and all bond dimensions meet the tolerance (i.e. are suitable for achieving the specified tolerance). However, the backward sweep preserves left-nesting only if taking the subset $\mathcal{I}'_{\ell-1} \subseteq \mathcal{I}_{\ell-1}$ does not remove any pivots, i.e. if actually $\mathcal{I}'_{\ell-1} = \mathcal{I}_{\ell-1}$. To achieve full nesting, left nesting can be restored by performing one more forward sweep at the same tolerance. This preserves right-nesting, because all bond dimensions already meet the tolerance, thus the last forward sweep removes no pivots from $\mathcal{J}_{\ell+1}$ for $\ell = 1, \ldots, \mathcal{L}-1$. For a related discussion in a different context, see Sec. 4.5.

1-site TCI can be used to (i) compress a TCI to a smaller rank; (ii) restore full nesting; (iii) improve the pivots at lower computational cost than its 2-site counterpart.

### 4.4.2 The 0-site TCI algorithm

The 0-site TCI algorithm sweeps through the pivot matrices $P_\ell$, prrLU decomposing each to yield updated pivot lists $\mathcal{I}'_\ell$, $\mathcal{J}'_{\ell+1}$ that replace $\mathcal{I}_\ell$, $\mathcal{J}_{\ell+1}$. 0-site TCI breaks nesting conditions. Its main usage is to improving the conditioning of $P_\ell$, by removing "spurious" pivots. For example, if a very large list of global pivots has been proposed, 0-site TCI can be used as a first filter to keep only the most relevant ones. It does not require new calls to $F$ tensor elements and hence can be used even when $F$ is no longer available.

### 4.5 CI- and LU-canonicalization

The MPS form $F_{\boldsymbol{\sigma}} = M_1^{\sigma_1} M_2^{\sigma_2} \cdots M_{\mathcal{L}}^{\sigma_{\mathcal{L}}}$ of a tensor is not unique. Indeed one can always replace $M_\ell \leftarrow M_\ell N_\ell$ and $M_{\ell+1} \leftarrow N_\ell^{-1} M_{\ell+1}$ for any $\ell$ and invertible matrix $N_\ell$ of appropriate dimension ($\chi_\ell \times \chi_\ell$). This is known as the gauge freedom. One can exploit this freedom to write the MPS into *canonical forms*. A standard way is to express it as a product of left- and right-unitary matrices around an *orthogonality center*, using the SVD decomposition [2] (the SVD-canonical form). In this section, we show how an arbitrary MPS can be put in TCI form, described uniquely in terms of pivot lists and corresponding slices of $F$. We call the corresponding algorithm CI-canonicalization. LU-canonicalization is a variant thereof.

The different canonical forms offer different advantages for subsequent operations on the tensor train. The SVD-canonical form is widely used in the tensor network community to improve performance of certain contractions by exploiting the unitarity properties of the MPS matrices. It is also very useful for algorithms such as DMRG as it provides a degree of non-locality to an otherwise local optimization. The CI-canonical form, on the other hand, is made up entirely of slices of the original MPS, i.e. a selection of values of the function through the

index sets $\mathcal{I}_\ell$ and $\mathcal{J}_\ell$. These set of points may have a value by themselves, e.g. as the starting point of a multi-variate optimization or to perform transformations (rotations, translations) in the case of quantics. Bringing a tensor into CI-canonical form is also a necessary step to enable the application of other TCI algorithms, such as TCI optimization (Sec. 4.3) or global pivot insertion (Sec. 4.3.5), which rely on the property that all core tensors of the MPS are defined through $\mathcal{I}_\ell$ and $\mathcal{J}_\ell$. LU canonicalization is a minor modification of CI canonicalization, and is mentioned here for completeness. The authors are not currently aware of any application unique to the LU-canonical form.

A simple way to put the MPS in a TCI form would be to apply the 2-site TCI to $F_{\boldsymbol{\sigma}}$, considered as a function of $\boldsymbol{\sigma}$. However, we present here a specific and *direct CI-canonicalization* algorithm to achieve this, based on the MPS structure. This algorithm has several advantages over the 2-site TCI: first, it is faster, taking only $O(\chi^3)$ operations (like the usual SVD-canonicalization) instead of $O(\chi^4)$;[1] second, it bypasses all the potential issues of the 2-site TCI algorithm discussed above, like ergodicity. Let us emphasize that while the CI-canonicalization algorithm can seem similar to the 1-site TCI algorithm, the two algorithms are actually different, as the former directly exploits the MPS structure of $F_{\boldsymbol{\sigma}}$.

### 4.5.1 CI-canonicalization.

Let us consider a MPS of the form

$$F_{\boldsymbol{\sigma}} = [M_1^{\sigma_1}]_{1a_1}[M_2^{\sigma_2}]_{a_1 a_2} \cdots [M_{\mathcal{L}}^{\sigma_{\mathcal{L}}}]_{a_{\mathcal{L}-1} 1} = \quad . \tag{45}$$

Here, the indices $a_\ell$ are ordinary MPS indices, *not* multi-indices $i_\ell$ or $j_\ell$ from pivot lists. CI-canonicalization is a sequence of exact transformations that convert the MPS to the TCI form of Eq. (34), built from $T_\ell$ and $P_\ell$ tensors that are slices of $F$ carrying multi-indices $i_\ell$, $j_\ell$ and that constitute full-rank matrices. We achieve this through three half-sweeps, involving exact (i.e. at machine precision) CI decompositions. A first forward sweep introduces left-nested lists $\widehat{\mathcal{I}}_\ell$ of row pivot multi-indices $\hat{\imath}_\ell$. Then, a backward sweep introduces right-nested lists $\mathcal{J}_\ell$ of column pivot multi-indices $j_\ell$ and matching subsets $\mathcal{I}_\ell \subset \widehat{\mathcal{I}}_\ell$ of row pivots $i_\ell$ (no longer left-nested). Finally, a second forward sweep restores left-nesting of row pivots. Important here is tracking the conversion from regular indices ($a_\ell$) to row ($i_\ell$, $\hat{\imath}_\ell$) and column ($j_\ell$) multi-indices. We thus display these indices explicitly below.

**First forward sweep.** We start with an exact CI decomposition (8) of $M_1$:

$$[M_1^{\sigma_1}]_{1a_1} = [C_1]_{\sigma_1 \hat{a}_1}[\widehat{P}_1^{-1}]_{\hat{a}_1 \hat{\imath}_1}[R_1]_{\hat{\imath}_1 a_1}, \qquad . \tag{46}$$

Here, $\hat{\imath}_1 \in \widehat{\mathcal{I}}_1 \subseteq \{\sigma_1\}$ are new multi-indices labeling pivot rows. The hat on $\widehat{P}_1$ emphasizes that it is *not* a slice of $F$, since the $\hat{a}_1$ are not multi-indices. Defining matrices $C_1^{\sigma_1}$ with elements $[C_1^{\sigma_1}]_{1\hat{a}_1} \equiv [C_1]_{\sigma_1 \hat{a}_1}$ we obtain

$$F_{\boldsymbol{\sigma}} = \left[ C_1^{\sigma_1}\widehat{P}_1^{-1}R_1 M_2^{\sigma_2} M_3^{\sigma_3} \cdots M_{\mathcal{L}}^{\sigma_{\mathcal{L}}} \right]_{11} = \quad . \tag{47}$$

---

[1]The complexity of using TCI for this purpose splits into $O(\chi^2)$ evaluations of the MPS which require $O(\chi^2)$ operations each. There is a possibility to cache the partial contractions of the MPS to bring the global cost down to $O(\chi^3)$ but the resulting algorithm is still inferior to the CI-canonicalization algorithm.

For $\ell \geq 2$ we iteratively define $\widetilde{M}_\ell^{\sigma_\ell} = R_{\ell-1} M_\ell^{\sigma_\ell}$ and group $\sigma_\ell$ with $\hat{\imath}_{\ell-1}$ to reshape $\widetilde{M}_\ell$ into a matrix which we factorize exactly with CI:

$$[R_{\ell-1} M_\ell^{\sigma_\ell}]_{\hat{\imath}_{\ell-1} a_\ell} = [\widetilde{M}_\ell]_{(\hat{\imath}_{\ell-1},\sigma_\ell) a_\ell} = [C_\ell^{\sigma_\ell}]_{\hat{\imath}_{\ell-1} \hat{a}_\ell} [\widehat{P}_\ell^{-1}]_{\hat{a}_\ell \hat{\imath}_\ell} [R_\ell]_{\hat{\imath}_\ell a_\ell}, \tag{48}$$

The tensor $C_\ell$ can be viewed as a matrix $C_\ell^{\sigma_\ell}$ with elements $[C_\ell^{\sigma_\ell}]_{\hat{\imath}_{\ell-1} \hat{a}_\ell} = [C_\ell]_{(\hat{\imath}_{\ell-1},\sigma_\ell) \hat{a}_\ell}$. The new row pivots are left-nested, $\hat{\imath}_\ell \in \widehat{\mathcal{I}}_\ell > \widehat{\mathcal{I}}_{\ell-1}$.

In practice, we do not calculate $C_\ell$ and $\widehat{P}_\ell$ separately. Instead, the prrLU decomposition directly yields the combination $A_\ell^{\sigma_\ell} = C_\ell^{\sigma_\ell} \widehat{P}_\ell^{-1}$:

$$[A_\ell^{\sigma_\ell}]_{\hat{\imath}_{\ell-1} \hat{\imath}_\ell} = [C_\ell^{\sigma_\ell}]_{\hat{\imath}_{\ell-1} \hat{a}_\ell} [\widehat{P}_\ell^{-1}]_{\hat{a}_\ell \hat{\imath}_\ell}, \tag{49}$$

By construction, see Eq. (10a), this product collapses to $[A_\ell^{\sigma_\ell}]_{\hat{\imath}_{\ell-1} \hat{\imath}_\ell} = \delta_{\hat{\imath}_{\ell-1} \oplus (\sigma_\ell), \hat{\imath}_\ell}$ whenever $\hat{\imath}_{\ell-1} \oplus (\sigma_\ell) \in \widehat{\mathcal{I}}_\ell$ (see also App. A.3).

After a full forward sweep to the very right we arrive at a tensor train of the form

$$F_{\boldsymbol{\sigma}} = \left[A_1^{\sigma_1} \cdots A_{\mathcal{L}-1}^{\sigma_{\mathcal{L}-1}} \widetilde{M}_{\mathcal{L}}^{\sigma_{\mathcal{L}}}\right]_{11} = \tag{50}$$

Here, the row pivots are by construction all left-nested as $\widehat{\mathcal{I}}_0 < \cdots < \widehat{\mathcal{I}}_{\mathcal{L}-1}$. This ensures the following important property: for any $\ell \leq \mathcal{L}-1$, the product $A_1 \cdots A_\ell$ collapses telescopically (starting from $A_1 A_2$) if evaluated on any pivot $\bar{\imath}_\ell = (\bar{\sigma}_1, \ldots, \bar{\sigma}_\ell) \in \widehat{\mathcal{I}}_\ell$ (cf. Eq. (A.12)):

$$= [A_1^{\bar{\sigma}_1} A_2^{\bar{\sigma}_2} \cdots A_\ell^{\bar{\sigma}_\ell}]_{1 \hat{\imath}_\ell} = \delta_{\bar{\imath}_\ell \hat{\imath}_\ell} \quad \text{if} \quad \hat{\imath}_\ell \in \widehat{\mathcal{I}}_\ell. \tag{51}$$

If Eq. (50) is evaluated on pivot configurations of $\widetilde{M}_{\mathcal{L}}$, having $\bar{\imath}_{\mathcal{L}-1} \in \widehat{\mathcal{I}}_{\mathcal{L}-1}$, we find via Eq. (51) that $F_{\bar{\imath}_{\mathcal{L}-1} \oplus (\sigma_{\mathcal{L}})} = [\widetilde{M}_{\mathcal{L}}^{\sigma_{\mathcal{L}}}]_{\bar{\imath}_{\mathcal{L}-1}, 1}$. Thus, $\widetilde{M}_{\mathcal{L}}$ is a slice of $F$, namely $\widetilde{M}_{\mathcal{L}} = F(\widehat{\mathcal{I}}_{\mathcal{L}-1}, \mathbb{S}_{\mathcal{L}})$. All $C_\ell$ and $\widehat{P}_\ell$ have full rank when viewed as matrices $[C_\ell]_{(\hat{\imath}_{\ell-1},\sigma_\ell) \hat{a}_\ell}$ and $\widehat{P}_{\hat{\imath}_\ell \hat{a}_\ell}$. However, $C_\ell$ and $\widetilde{M}_{\mathcal{L}}$ may still be rank-deficient when viewed as matrices $[C_\ell]_{\hat{\imath}_{\ell-1}(\sigma_\ell, \hat{a}_\ell)}$ or $[\widetilde{M}_{\mathcal{L}}]_{\hat{\imath}_{\mathcal{L}-1} \sigma_{\mathcal{L}}}$.

**Backward sweep.** Starting from Eq. (50), we sweep backward to generate right-nested column multi-indices $j_\ell$. The CI factorizations are analogous to those of the forward sweep, with two differences: they group $\sigma_\ell$ with column (not row) indices prior to factorization; the resulting $P_\ell$ and $R_\ell$ matrices are slices of $F$, thus revealing the bond dimensions of $F$.

We initialize the backward sweep by factorizing $\widetilde{M}_{\mathcal{L}}^{\sigma_{\mathcal{L}}}$ exactly as $C_{\mathcal{L}-1} P_{\mathcal{L}-1}^{-1} R_{\mathcal{L}}^{\sigma_{\mathcal{L}}}$:

$$[\widetilde{M}_{\mathcal{L}}^{\sigma_{\mathcal{L}}}]_{\hat{\imath}_{\mathcal{L}-1} 1} = [C_{\mathcal{L}-1}]_{\hat{\imath}_{\mathcal{L}-1} j_{\mathcal{L}}} [P_{\mathcal{L}-1}^{-1}]_{j_{\mathcal{L}} i_{\mathcal{L}-1}} [R_{\mathcal{L}}]_{i_{\mathcal{L}-1} \sigma_{\mathcal{L}}}, \tag{52}$$

Here, $j_{\mathcal{L}} \in \mathcal{J}_{\mathcal{L}} \subseteq \{\sigma_{\mathcal{L}}\}$ are multi-indices labeling pivot columns; $i_{\mathcal{L}-1} \in \mathcal{I}_{\mathcal{L}-1} \subseteq \widehat{\mathcal{I}}_{\mathcal{L}-1}$ are row pivots. Note that $R_{\mathcal{L}}$ and $P_{\mathcal{L}-1}$, being subslices of $\widetilde{M}_{\mathcal{L}}$, are slices of $F$, namely $R_{\mathcal{L}} = F(\mathcal{I}_{\mathcal{L}-1}, \mathbb{S}_{\mathcal{L}})$ and $P_{\mathcal{L}} = F(\mathcal{I}_{\mathcal{L}-1}, \mathcal{J}_{\mathcal{L}})$. We thus make the identification $T_{\mathcal{L}} = R_{\mathcal{L}}$.

For $\ell \leq \mathcal{L}-1$ we iteratively define $\widetilde{N}_\ell^{\sigma_\ell} = A_\ell^{\sigma_\ell} C_\ell$ and factorize it as $C_{\ell-1} P_{\ell-1}^{-1} R_\ell^{\sigma_\ell}$:

$$[A_\ell^{\sigma_\ell}]_{\hat{\imath}_{\ell-1} \hat{\imath}_\ell} [C_\ell]_{\hat{\imath}_\ell j_{\ell+1}} = [\widetilde{N}_\ell]_{\hat{\imath}_{\ell-1}(\sigma_\ell, j_{\ell+1})} = [C_{\ell-1}]_{\hat{\imath}_{\ell-1} j_\ell} [P_{\ell-1}^{-1}]_{j_\ell i_{\ell-1}} [R_\ell^{\sigma_\ell}]_{i_{\ell-1} j_{\ell+1}}, \tag{53}$$

Here, the new column multi-indices are right-nested, $j_\ell \in \mathcal{J}_\ell > \mathcal{J}_{\ell+1}$, while the row multi-indices are a subset of the previous ones, $i_{\mathcal{L}-1} \in \mathcal{I}_{\mathcal{L}-1} \subseteq \widehat{\mathcal{I}}_{\mathcal{L}-1}$ (thus possibly breaking left-nesting, $\mathcal{I}_{\ell-1} \not< \mathcal{I}_\ell$). We show below that $R_\ell$ is a slice of $F$, thus we rename it $T_\ell = R_\ell$, and that $P_{\ell-1}$, too, is a slice of $F$. We also define $B_\ell^{\sigma_\ell} = P_{\ell-1}^{-1} T_\ell^{\sigma_\ell}$,

$$[B_\ell^{\sigma_\ell}]_{j_\ell j_{\ell+1}} = [P_{\ell-1}^{-1}]_{j_\ell i_{\ell-1}} [T_\ell^{\sigma_\ell}]_{i_{\ell-1} j_{\ell+1}}, \qquad \frac{B_\ell}{j_\ell \underset{\sigma_\ell}{\,\Vert\,} j_{\ell+1}} = \frac{P_{\ell-1}^{-1} \quad T_\ell}{j_\ell \diamond i_{\ell-1} \underset{\sigma_\ell}{\,\Vert\,} j_{\ell+1}}. \tag{54}$$

Via Eq. (10b) it collapses to $[B_\ell^{\sigma_\ell}]_{j_\ell, j_{\ell+1}} = \delta_{j_\ell, (\sigma_\ell) \oplus j_{\ell+1}}$ if $(\sigma_\ell) \oplus j_{\ell+1} \in \mathcal{J}_\ell$. Importantly, the inner summation for $B_\ell$ now involves multi-indices $i_{\ell-1}$ (for $A_\ell$ it still involved $\hat{a}_\ell$ indices).

Sweeping backward up to site $\ell$, and then all the way to the very left, we obtain

$$F_{\boldsymbol{\sigma}} = \left[ A_1^{\sigma_1} \cdots A_{\ell-1}^{\sigma_{\ell-1}} \widetilde{N}_\ell^{\sigma_\ell} B_{\ell+1}^{\sigma_{\ell+1}} \cdots B_{\mathcal{L}}^{\sigma_{\mathcal{L}}} \right]_{11} = \frac{A_1}{1 \underset{\sigma_1}{\,\Vert\,} \hat{i}_1} \cdots \frac{A_{\ell-1} \quad \widetilde{N}_\ell \quad B_{\ell+1}}{i_{\ell-2} \underset{\sigma_{\ell-1}}{\,\Vert\,} \hat{i}_{\ell-1} \underset{\sigma_\ell}{\,\circ\,} j_{\ell+1} \underset{\sigma_{\ell+1}}{\,\Vert\,} j_{\ell+2}} \cdots \frac{B_{\mathcal{L}}}{j_{\mathcal{L}} \underset{\sigma_{\mathcal{L}}}{\,\Vert\,} 1} \tag{55}$$

$$= [\widetilde{N}_1^{\sigma_1} B_2^{\sigma_2} \cdots B_{\mathcal{L}}^{\sigma_{\mathcal{L}}}]_{11} = \frac{\widetilde{N}_1 \quad B_2}{1 \underset{\sigma_1}{\,\square\,} j_2 \underset{\sigma_2}{\,\Vert\,}} \cdots \frac{B_{\mathcal{L}}}{j_{\mathcal{L}} \underset{\sigma_{\mathcal{L}}}{\,\Vert\,} 1}. \tag{56}$$

In Eq. (55), the column pivots are by construction all right-nested as $\mathcal{J}_{\ell+1} > \cdots > \mathcal{J}_{\mathcal{L}+1}$, and in Eq. (56) they are fully right-nested, $\mathcal{J}_2 > \cdots > \mathcal{J}_{\mathcal{L}+1}$. Importantly, this ensures that for any $\ell \geq 2$ the product $B_\ell \cdots B_{\mathcal{L}}$ collapses telescopically (starting from $B_{\mathcal{L}-1} B_{\mathcal{L}}$) if it is evaluated on any pivot $\bar{j}_\ell = (\bar{\sigma}_\ell, \ldots, \bar{\sigma}_{\mathcal{L}}) \in \mathcal{J}_\ell$ (cf. Eq. (A.12b)):

$$\frac{B_\ell}{j_\ell \underset{\bar{\sigma}_\ell}{\,\Vert\,} j_{\ell+1}} \cdots \frac{B_{\mathcal{L}-1} \quad B_{\mathcal{L}}}{j_{\mathcal{L}-1} \underset{\bar{\sigma}_{\mathcal{L}-1}}{\,\Vert\,} j_{\mathcal{L}} \underset{\bar{\sigma}_{\mathcal{L}}}{\,\Vert\,} 1} = [B_\ell^{\bar{\sigma}_\ell} \cdots B_{\mathcal{L}-1}^{\bar{\sigma}_{\mathcal{L}-1}} B_{\mathcal{L}}^{\bar{\sigma}_{\mathcal{L}}}]_{j_\ell 1} = \delta_{j_\ell \bar{j}_\ell}, \quad \forall \bar{j}_\ell \in \mathcal{J}_\ell. \tag{57}$$

Consider Eq. (55) with $\ell > 1$. If evaluated on pivot configurations of $\widetilde{N}_\ell$, having $\bar{i}_{\ell-1} \in \widehat{\mathcal{I}}_{\ell-1}$ and $\bar{j}_{\ell+1} \in \mathcal{J}_{\ell+1}$, it collapses telescopically via Eqs. (50) and (57) to $F_{\bar{i}_{\ell-1} \oplus \sigma_\ell \oplus \bar{j}_{\ell+1}} = [\widetilde{N}_\ell^{\sigma_\ell}]_{\bar{i}_{\ell-1} \bar{j}_{\ell+1}}$. Therefore, $\widetilde{N}_\ell$ is a slice of $F$, namely $\widetilde{N}_\ell = F(\widehat{\mathcal{I}}_{\ell-1}, \mathbb{S}_\ell, \mathcal{J}_{\ell+1})$. It follows that the same is true for its subslices, $T_\ell = R_\ell = F(\mathcal{I}_{\ell-1}, \mathbb{S}_\ell, \mathcal{J}_{\ell+1})$ and $P_{\ell-1}(\mathcal{I}_{\ell-1}, \mathcal{J}_\ell)$, as announced above. Therefore, the CI factorization of $\widetilde{N}_\ell$ reveals the bond dimension of $F$ for bond $\ell-1$, namely $\chi_{\ell-1} = |\mathcal{I}_{\ell-1}| = |\mathcal{J}_\ell|$. The latter is an intrinsic property of $F$ and will remain unchanged under arbitrary gauge transformations (e.g. exact SVDs or CIs) on its bonds. A telescope argument shows that $\widetilde{N}_1$ in (56) is a slice of $F$, too, thus we identify $T_1 = \widetilde{N}_1 = F(\mathbb{S}_1, \mathcal{J}_2)$.

Using $B_\ell^{\sigma_\ell} = P_{\ell-1}^{-1} T_\ell^{\sigma_\ell}$ in Eq. (56), we obtain a tensor train in the TCI form of Eq. (34), namely $F_{\boldsymbol{\sigma}} = [T_1^{\sigma_1} P_1^{-1} T_2^{\sigma_2} \cdots P_{\mathcal{L}-1}^{-1} T_{\mathcal{L}}^{\sigma_{\mathcal{L}}}]_{11}$. Here, all ingredients are slices of $F$, labeled by multi-indices, and each $T_\ell$ is full rank for both ways of viewing it as a matrix, $[T]_{(i_{\ell-1}, \sigma_\ell) j_{\ell+1}}$ or $[T]_{i_{\ell-1}(\sigma_\ell, j_{\ell+1})}$. The column pivots are fully right-nested. However, the row pivots are *not* fully left-nested, since the backward sweep dropped some row pivots.

**Second forward sweep.** To obtain a tensor train in fully nested TCI form, we perform a second exact forward sweep, using the 1-site TCI algorithm of Sec. 4.4.1. This generates fully left-nested row pivots. Moreover, since all bond dimensions have already been revealed during the backward sweep, no column pivots are lost during the second forward sweep, thus the column pivots remain fully right-nested. More explicitly: during the second forward sweep, the rank of $[T_\ell]_{(i_{\ell-1}, \sigma_\ell) i_\ell}$ is equal to the number of its columns, $\chi_\ell = |\mathcal{I}_i|$, hence this matrix has full rank. Therefore, its exact CI decomposition retains all columns, loosing none. The resulting tensor train is fully nested, as desired.

Table 2: Computational cost of the main TCI algorithms in `xfac` / `tci.jl`.

| action | variant | | calls to $F_\sigma$ | algebra cost |
|---|---|---|---|---|
| iterate | rook piv. | 2-site | $\mathcal{O}(\chi^2 d n_{\text{rook}}\mathcal{L})$ | $\mathcal{O}(\chi^3 d n_{\text{rook}}\mathcal{L})$ |
| | full piv. | 2-site | $\mathcal{O}(\chi^2 d^2\mathcal{L})$ | $\mathcal{O}(\chi^3 d^2\mathcal{L})$ |
| | full piv. | 1-site | $\mathcal{O}(\chi^2 d\mathcal{L})$ | $\mathcal{O}(\chi^3 d\mathcal{L})$ |
| | full piv. | 0-site | $0$ | $\mathcal{O}(\chi^3\mathcal{L})$ |
| achieve full nesting | | | $\mathcal{O}(\chi^2 d\mathcal{L})$ | $\mathcal{O}(\chi^3 d\mathcal{L})$ |
| add $n_p$ global pivots | | | $\mathcal{O}\big((2\chi + n_p)n_p\mathcal{L}\big)$ | $\mathcal{O}\big((\chi + n_p)^3\mathcal{L}\big)$ |
| compress tensor train | SVD | | $0$ | $\mathcal{O}(\chi^3 d\mathcal{L})$ |
| | LU | | | |
| | CI | | | |

**CI-canonicalization with compression.** CI-canonicalization can optionally be combined with compression at the cost of an extra half-sweep. Then, the sequence becomes: (i) An exact forward sweep builds row indices $\{\hat{\imath}_\ell\}$. (ii) A backward sweep with compression builds column indices $\{j_\ell\}$ according to a specified tolerance $\tau$ and/or rank $\chi$, while possibly reducing row indices from $\{\hat{\imath}_\ell\}$ to $\{i_\ell\}$. (iii) A forward sweep with compression finalizes row indices according to the specifications while possibly further reducing column indices; this yields a proper TCI form with the specified $\tau$ and/or $\chi$. (iv) A final optional backward sweep without compression restores full nesting.

### 4.5.2 LU-canonicalization

*LU*-canonicalization proceeds in a similar manner, but instead of the CI decomposition $CP^{-1}R$ it iteratively uses the corresponding LU decomposition $LDU$, where $L$ is lower-triangular, $U$ upper-triangular and $D$ diagonal. Forward sweeps generate $LLL\cdots$ products while absorbing $DU$ factors rightwards; backward sweeps generate $\cdots UUU$ products while absorbing $LD$ factors leftwards. In this manner, one can express $F$ in the form $L_1\cdots L_{\ell-1}\tilde{N}_\ell U_{\ell+1}\cdots U_\mathcal{L}$, for any $\ell = 1,\ldots,\mathcal{L}$, if desired.

## 4.6 High-level algorithms

We have now enlarged our toolbox with several flavors of TCI algorithms and canonical forms with various options and variants. These algorithms can be combined in numerous ways to provide more abstract, high-level algorithms for different tasks. The best combination will depend on the intended application, and we provide some rough practical guidelines below. The corresponding computational costs are listed in Table 2.

- 2-site TCI in *accumulative* plus *rook pivoting* mode is the fastest technique. It requires the least pivot exploration and very often provides very good results on its own. The accuracy can be improved, if desired, by following this with a few (cheap) 1-site TCI sweeps to reset the pivots.

- 2-site TCI in *reset* plus *rook pivoting* mode is marginally more costly than the above but more stable. It is a good default. For small $d$, one should use the *full* search, which is even more stable and involves almost no additional cost if $d \leq 2n_{\text{rook}}$.

- If good heuristics for proposing pivots are available or ergodicity issues arise, one should consider switching to *global pivot proposal* followed by 2-site TCI.

- To obtain the best final accuracy at fixed $\chi$, one can build a TCI with a higher rank $\chi' > \chi$, then compress it using either SVD or CI recompression.

- For calculations of integrals or sums, we recommend the environment mode. In some calculations, we have observed it to increase the accuracy by two digits for the same computational cost.

## 4.7 Operations on tensor trains

The various TCI algorithms can be combined with other MPS algorithms [2,9] in various ways. Let us mention a few examples.

**Function composition.** Given a TCI $\widetilde{F}_{\sigma}$ approximating a function $f$, its composition with another function $g(f(x))$, can be performed by constructing another TCI, $\widetilde{G}_{\sigma} \approx g(\widetilde{F}_{\sigma})$. The repeated evaluations of $\widetilde{F}_{\sigma}$ required for this can be accelerated by caching partial contractions of the tensor train. This gives a runtime complexity of $\mathcal{O}(\chi_{\widetilde{F}}\chi_{\widetilde{G}}^3 d\mathcal{L})$, where $\chi_{\widetilde{F}}$ and $\chi_{\widetilde{G}}$ are the ranks of $\widetilde{F}$ and $\widetilde{G}$. Since the tensors $T_\ell$ are slices of $\widetilde{F}$, the new TCI $\widetilde{G}$ can be initialized by applying $g$ to each element of $T_\ell$. For simple, monotonically increasing functions $g$, the subsequent optimization typically converges very quickly.

**Element-wise tensor addition.** Given two tensor trains, $\widetilde{F} = M_1 M_2 \cdots M_{\mathcal{L}}$ and $\widetilde{F}' = M_1' M_2' \cdots M_{\mathcal{L}}'$, their element-wise sum $\widetilde{F}''_{\sigma} = \widetilde{F}_{\sigma} + \widetilde{F}'_{\sigma}$ can be computed by creating block matrices,

$$M_\ell''^{\sigma_\ell} = \begin{pmatrix} M_\ell^{\sigma_\ell} & 0 \\ 0 & M_\ell'^{\sigma_\ell} \end{pmatrix}, \tag{58}$$

and recompressing the resulting tensor train $\widetilde{F}''_{\sigma} = \mathrm{Tr}(M_1''^{\sigma_1} M_2''^{\sigma_2} \cdots M_{\mathcal{L}}''^{\sigma_{\mathcal{L}}})$ using the CI-canonicalization algorithm. The total runtime complexity is dominated by that of the recompression, namely $\mathcal{O}\big((\chi + \chi')^3 d\mathcal{L}\big)$, where $\chi$ and $\chi'$ are the ranks of $\widetilde{F}$ and $\widetilde{F}'$. An advantage over the conventional SVD-based recompression is that the resulting MPS is truncated in terms of the maximum norm rather than the Frobenius norm, which can be more accurate for certain applications (see Sec. 7 for an example).

**Matrix-vector contractions.** Consider the contraction $G_{\sigma'\sigma}F_{\sigma}$ in a $d^{\mathcal{L}}$-dimensional space. If $G$ and $F$ are compressible tensors, TCI can be used to approximate them by an MPO and MPS, respectively, where the former is of the form

$$G_{\sigma'\sigma} \approx \widetilde{G}_{\sigma'\sigma} = [W_1]_{1i_1}^{\sigma_1'\sigma_1}[W_2]_{i_1i_2}^{\sigma_2'\sigma_2}\cdots[W_{\mathcal{L}}]_{i_{\mathcal{L}-1}1}^{\sigma_{\mathcal{L}}'\sigma_{\mathcal{L}}} = \quad . \tag{59}$$

Their contraction yields another MPS:

$$G_{\sigma'\sigma}F_{\sigma} \approx \widetilde{G}_{\sigma'\sigma}\widetilde{F}_{\sigma} = \quad = \quad . \tag{60}$$

The MPO-MPS contraction can be computed exactly by performing the sum $\sum_{\sigma}$, yielding an MPS with bond dimensions $\chi_{\ell,\widetilde{G}}\chi_{\ell,\widetilde{F}}$. The standard, SVD-based MPS toolbox offers two ways to obtain a compressed version of this result: (i) Fitting the exact result to an MPS with reduced bond dimensions; or (ii) zip-up compression, where the MPO-MPS contraction is performed one site at a time, followed by a local compression before proceeding to the next site [51–53]. TCI in principle offers further options, e.g. zip-up compression as in (ii), but performing all compressions using CI instead of SVD. The computational times of all these options are $\mathcal{O}(\chi^4\mathcal{L})$ for $\chi_{\widetilde{G}} = \chi_{\widetilde{F}} = \chi_{\widetilde{G}\widetilde{F}} = \chi$. The potential advantages of TCI- or CI-based contractions are twofold: the resulting MPS is truncated in terms of the maximum norm; and we can use the rook search, which can be efficient for large local dimensions $d$. To what extent TCI-based MPO-MPS contraction schemes have a chance of outperforming SVD-based ones will depend on context and is a question to be explored in future work.

## 4.8 Relation to machine learning

In this section we briefly compare and contrast TCI with other learning approaches such as deep neural network approaches.

TCI unfolding algorithms construct MPS representations $\widetilde{F}$ for $F$ by systematically learning its structure. *Learning* the tensor $F$ in the traditional machine learning sense would amount to the following sequence: (1) draw a training set of configurations/values $\{\boldsymbol{\sigma}, F_{\boldsymbol{\sigma}}\}$; (2) design a model $\widetilde{F}_{\boldsymbol{\sigma}}$ (typically a deep neural network); (3) fit the model to the training set by minimizing the error $\|F - \widetilde{F}\|$, measured w.r.t. to some norm (typically using a variant of stochastic gradient descent); and (4) use the model to evaluate $\widetilde{F}_{\boldsymbol{\sigma}}$ for new configurations. TCI implements this program with a few very important differences:

(1) TCI does not work with a given data set; instead, it actively requests the configurations that are likely to bring the most new information on the tensor (active learning).

(2) The model is not a neural network but a tensor train, i.e. a tensor network (a highly structured model). If $F$ has a low-rank structure it can be accurately approximated by a low-rank tensor train $\widetilde{F}$, with an exponentially smaller memory footprint. For TCI to learn $\widetilde{F}$, the number of samples of $F_{\boldsymbol{\sigma}}$ requested by TCI will be $\ll d^{\mathcal{L}}$.

(3) The actual TCI algorithm used to minimize the error $\|F - \widetilde{F}\|$ is conceptually very different from gradient descent. It guarantees that the error is smaller than a specified tolerance $\tau$ for all known samples.

(4) Once $\widetilde{F}$ has been found, its elements $\widetilde{F}_{\boldsymbol{\sigma}}$ can be computed for *all* configurations $\boldsymbol{\sigma}$. This by itself may not seem like progress, since we had assumed that one could call any $F_{\boldsymbol{\sigma}}$ to begin with. Nevertheless, access to any $\widetilde{F}_{\boldsymbol{\sigma}}$ may be useful in cases where accessing $F_{\boldsymbol{\sigma}}$ is computationally expensive (e.g. the result of a complex simulation), or possible only in a limited time window (e.g. while collecting experimental data). Much more importantly, the tensor train structure of $\widetilde{F}$ permits subsequent operations (such as computing $\sum_{\boldsymbol{\sigma}} F_{\boldsymbol{\sigma}}$ over all configurations) to be performed exponentially faster.

# 5 Application: Computing integrals and sums

We now turn to practical illustrations of TCI in action. The following three sections give examples of various TCI applications, together with code listings illustrating how they can be coded using `xfac` or `TCI.jl` libraries.

The present section deals with the most obvious application of TCI: computing large integrals and sums. The basic idea has already been briefly introduced in Sec. 2.2. Here, we provide more details, a further example and the code listing used to compute it.

For historical reasons the `xfac` library implements two sets of algorithms corresponding to two classes `TensorCI1` and `TensorCI2`. The former is based on CI in accumulative mode and will eventually be deprecated while the latter is based on prrLU and supports many different modes. The Julia package `TCI.jl` follows closely the implementation of `TensorCI2`.

## 5.1 Quadratures for multivariate integrals

Consider a multi-dimensional integral, $\int_D d^{\mathcal{N}}\mathbf{x} f(\mathbf{x})$, with $\mathbf{x} = (x_1, \dots, x_{\mathcal{N}})$, over a domain $D = D_1 \times \cdots \times D_{\mathcal{N}}$. (We here denote the number of variables by $\mathcal{N}$ (not $\mathcal{L}$), for notational consistency with Sec. 6 and Refs. [13, 15].) For each variable $x_\ell \in D_\ell$ we choose a grid of discretization points $\{x_\ell(\sigma_\ell)\}$, enumerated by an index $\sigma_\ell = 1, \dots, d_\ell$, and an associated grid of quadrature weights $\{w_\ell(\sigma_\ell)\}$, such that its 1D integral is represented by the quadrature

rule $\int_{D_\ell} dx_\ell f(x_\ell) \approx \sum_{\sigma_\ell=1}^{d_\ell} w_\ell(\sigma_\ell) f(x_\ell(\sigma_\ell))$. A typical choice would be the Gauss–Kronrod or Gauss–Legendre quadrature. Then, we use the natural tensor representation $F$ (Eq. (2)) of $f$ and its TCI unfolding $\widetilde{F}$ to obtain a factorized representation of the function,

$$f(\mathbf{x}(\boldsymbol{\sigma})) = F_{\boldsymbol{\sigma}} \simeq \widetilde{F}_{\boldsymbol{\sigma}} = \prod_{\ell=1}^{\mathcal{N}} M_\ell^{\sigma_\ell}. \tag{61}$$

Since $\widetilde{F}$ does not incorporate quadrature weights, this is called an *unweighted* unfolding. The $\mathcal{N}$-fold integral over $f$ can thus be computed as [8, 12, 13]

$$\int_D d^{\mathcal{N}}\mathbf{x} f(\mathbf{x}) \approx \sum_{\boldsymbol{\sigma}} \Big( \prod_{\ell=1}^{\mathcal{N}} w_\ell(\sigma_\ell) \Big) f(\mathbf{x}(\boldsymbol{\sigma})) \approx \prod_{\ell=1}^{\mathcal{N}} \Big[ \sum_{\sigma_\ell=1}^{d_\ell} w_\ell(\sigma_\ell) M_\ell^{\sigma_\ell} \Big]. \tag{62}$$

The first approximation $\approx$ refers to the error of the quadrature rule (controlled by the number of points $d_\ell$ in the discretization of each variable). The second $\approx$ is the factorization error (controlled by the rank $\chi$) of the unfolding (61). Thus, the computation of one $\mathcal{N}$-dimensional integral has been replaced by $\mathcal{N}\chi^2$ exponentially easier problems, namely 1-dimensional integrals that each amount to performing a sum $\sum_{\sigma_\ell}$.

An alternative to unweighted unfolding is *weighted unfolding*, which unfolds the weighted tensor $\Big( \prod_{\ell=1}^{\mathcal{N}} w_\ell(\sigma_\ell) \Big) f(\mathbf{x}(\boldsymbol{\sigma})) = F_{\boldsymbol{\sigma}} \simeq \widetilde{F}_{\boldsymbol{\sigma}} = \prod_{\ell=1}^{\mathcal{N}} M_\ell^{\sigma_\ell}$. Then, the integral is given by

$$\int_D d^{\mathcal{N}}\mathbf{x} f(\mathbf{x}) \approx \sum_{\boldsymbol{\sigma}} \Big( \prod_{\ell=1}^{\mathcal{N}} w_\ell(\sigma_\ell) \Big) f(\mathbf{x}(\boldsymbol{\sigma})) \approx \prod_{\ell=1}^{\mathcal{N}} \Big[ \sum_{\sigma_\ell=1}^{d_\ell} M_\ell^{\sigma_\ell} \Big]. \tag{63}$$

The weighted tensor has the same rank as the unweighted one since the weights form a rank-1 MPS. The weighted unfolding can sometimes be more efficient than unweighted unfolding—achieving higher accuracy for a given $\chi$—since the error estimation during the TCI construction includes information about the weights. The weighted unfolding is typically combined with the use of the environment error that directly targets the best error for the calculation of integrals.

## 5.2 Example code for integrating multivariate functions

Next, we illustrate how TCI computations of multivariate integrals can be performed using the `xfac` toolbox. For definiteness, we consider a toy example from Ref. [54] for which the result is known analytically: the computation of the following integral over a hypercube:

$$I^{(\mathcal{N})} = \int_{[0,1]^{\mathcal{N}}} dx_1 \cdots dx_{\mathcal{N}} f(\mathbf{x}), \qquad f(\mathbf{x}) = \frac{2^{\mathcal{N}}}{1 + 2\sum_{\ell=1}^{\mathcal{N}} x_\ell}. \tag{64}$$

For $\mathcal{N} = 5$, the analytical solution of above integral is

$$I^{(5)} = [-65205 \log(3) - 6250 \log(5) + 24010 \log(7) + 14641 \log(11)]/24. \tag{65}$$

```python
import xfacpy
from math import log

N = 5  # Number of dimensions

def f(x):  # Integrand function
    f.neval += 1
    return 2**N / (1 + 2 * sum(x))
```

```
10
11
12  f.neval = 0
13
14  # Exact integral value in 5 dimensions
15  i5 = (- 65205 * log(3) - 6250 * log(5) + 24010 * log(7) + 14641 * log(11)) / 24
16
17  # Gauss-Kronrod abscissas (xell) and weights (well)
18  xell, well = xfacpy.GK15(0, 1)
19
20  # TCI1 Tensor factorization, no environment
21  tci = xfacpy.CTensorCI1(f, [xell] * N)
22
23  # Estimate integral and error
24  for hsweep in range(14):
25    tci.iterate()
26    # calculate the integal over the hypercube
27    itci = tci.get_TensorTrain().sum([well] * N)
28    print("hsweep= {}, neval= {}, I_tci= {:e}, |I_tci - I_exact|= {:e}, in-sample err= {:e}"
29          .format(hsweep+1, f.neval, itci, abs(itci - i5), tci.pivotError[-1]))
```

Listing 1: Python code to numerically compute the integral $I^{(\mathcal{N}=5)}$ (Eq. (64)) using the `xfac` package with `TensorCI1`. The script performs 14 half-sweeps using continuous TCI on a 15 point Gauss–Kronrod grid. For each half-sweep (`hsweep`), the number of function evaluations (`neval`), the approximate integral value (`itci`), the absolute error with respect to the exact integral value (`i5`) from Eq. (65) and the in-sample error (`insample err`) is printed. These values are shown in Figs. 4(a-c).

The Python script to perform the integration numerically using the Python bindings of `xfac` (package `xfacpy`) is shown in code Listing 1; see Listing 11 for an equivalent Julia code using `TCI.jl`. Both codes can be trivially adapted to compute the integral of any function which is known explicitly by just modifying the definition of $f(\mathbf{x})$.

In the Python code, lines 1 and 2 import the packages `xfacpy` and the `log` function (needed for comparison with Eq. (65)). Lines 7–9 define the user-supplied function $f$; line 8 defines an (optional) attribute of $f$, `neval`, counting the number of times the integrand is called; line 9 defines the integrand. Here $x$ is a list of floats or a numpy array. For each argument $x_\ell$, the user specifies a grid $\{x_\ell(\sigma_\ell)\}$ of $d_\ell$ quadrature nodes, enumerated by an index $\sigma_\ell$, and an associated grid of quadrature weights $\{w_\ell(\sigma_\ell)\}$ (cf. Sec. 5.1). Here, we use the nodes and weights of the Gauss–Kronrod quadrature, with $d_\ell = 15$ for all $\ell$. For convenience, the Gauss–Kronrod quadrature is included in `xfac` so that the GK15 function in line 18 returns two lists, `xell` and `well`, containing the quadrature nodes $\{x_\ell(\sigma_\ell)\}$ and weights $\{w_\ell(\sigma_\ell)\}$, respectively (chosen the same for all $\ell$).

The `CTensorCI()` object created in line 21 is the basic object used to perform TCI on a continuous function, discretized as $F_{\boldsymbol{\sigma}} = f(\mathbf{x}(\boldsymbol{\sigma}))$. This class performs the factorization in accumulative mode. Note that `CTensorCI()` is a thin wrapper over the corresponding discrete class `TensorCI()` that creates $F_{\boldsymbol{\sigma}}$ from $f(\mathbf{x})$ and the grids $\mathbf{x}_\ell(\sigma_\ell)$. To instantiate the class, two arguments must be provided: the function `f`, and the grid on which the function will be called, `[xell] * N`. For $\mathcal{N} = 5$, the latter is equivalent to `[xell, xell, xell, xell, xell]`, i.e. five copies of the GK15 grid (a list of list of points).

The loop in lines 24–29 performs a series of half-sweeps, alternating left-to-right and right-to-left, 14 in total (i.e. 7 full sweeps), to iteratively improve the TCI approximation $\widetilde{F}_{\boldsymbol{\sigma}}$ of the tensor $F_{\boldsymbol{\sigma}}$. In line 25, `tci.iterate()` performs one half-sweep, and in line 27, `tci.get_TensorTrain().sum([well] * N)` calculates the integral according to Eq. (62). Finally, lines 28 and 29 print the results: the number of half-sweeps, `hsweep`; the number of calls to $f$, `neval`; the calculated value of the integral, `itci`; its error with respect to the exact calculation, $|I^{(\mathcal{N})} - \widetilde{I}^{(\mathcal{N})}|$; and the "in-sample error", `in-sample err`, defined as the maximum difference

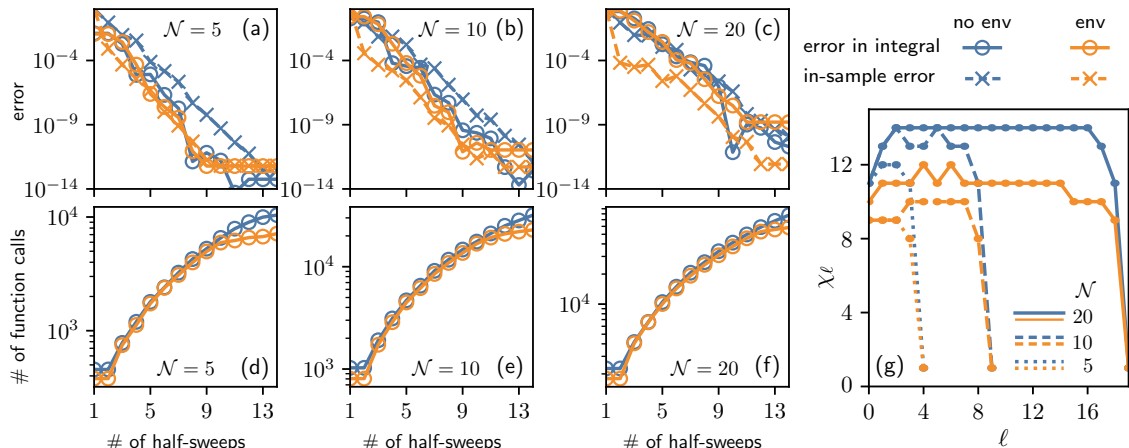

Figure 4: Performance metrics for the TCI computation of the $\mathcal{N}$-dimensional integral $I^{(\mathcal{N})} = \int d^{\mathcal{N}}\mathbf{x} f(\mathbf{x})$ of Eq. (64) using the natural tensor representation (2), for $\mathcal{N} = 5, 10, 20$. (a–c) The relative error for the integral $|1 - I^{(\mathcal{N})}/\widetilde{I}^{(\mathcal{N})}|$ (solid lines with circles), the maximum (over all sampled pivots) of the relative in-sample error $|1 - F_{\boldsymbol{\sigma}}/\widetilde{F}_{\boldsymbol{\sigma}}|_{\infty}$ (dashed lines with crosses), and (d–f) the number of function calls, all plotted versus the number of half-sweeps. The TCI computation of $\widetilde{I}^{(\mathcal{N})}$ has been performed on a 15-point Gauss–Kronrod grid (i.e. $d_{\ell} = 15$), either in the no environment mode ("no env", blue) or in the environment mode ("env", orange). (g) The final bond dimension $\chi_{\ell}$ plotted vs. $\ell \in [1, \mathcal{N}]$, for $\mathcal{N} = 5, 10, 20$. The growth of $\chi_{\ell}$ with increasing $\ell$ or $\mathcal{N} - \ell$ flattens off at rather small values of $\chi = \max\{\chi_{\ell}\}$, indicating that the function $f(\mathbf{x})$ is strongly compressible. [Code: Listing 1 (Python), 11 (Julia)]

$|F_{\boldsymbol{\sigma}} - \widetilde{F}_{\boldsymbol{\sigma}}|$ during the half-sweep (a "training set error", albeit a very conservative one because the algorithm is actively looking for points with large errors). The code above performs the bare variant (no environment) of the factorization of $F_{\boldsymbol{\sigma}}$. For comparison, we have also computed the factorisation in environment mode (see Sec. B.1.1 for the corresponding syntax).

Figure 4 shows the two errors (upper panel) and number of function calls (lower panel) as a function of the number of half-sweeps for $\mathcal{N} = 5, 10$, and 20. The convergence of the integral is very fast and depends only weakly on the number of dimensions. It turns out that, in this example, the environment mode (orange) does not bring much advantage over the bare mode (blue). To highlight the strength of TCI we note that for $\mathcal{N} = 5$ (or $\mathcal{N} = 20$) the 14 half-sweeps needed to reach an absolute error below $10^{-10}$ (or $10^{-8}$) required roughly $10^4$ (or $10^5$) function calls, hence the ratio of the number of sampled points to all points of $F_{\boldsymbol{\sigma}}$ was only $10^4/15^5 \approx 10^{-2}$ (or $10^5/15^{20} \approx 10^{-19}$). In general, if the rank of the MPS unfolding of the integrand remains roughly constant as the number of dimensions increases, then the gain in favor of TCI increases exponentially.

Finally, let us state that the method presented above only works if the chosen quadrature model (e.g. the Gauss–Kronrod quadrature) is suitable for the integrand in question. A variant of this method using the quantics representation is presented in section 6.3.2.

## 5.3 Example of computation of partition functions

Our second example is very similar to the previous one except that we now consider an object that is already a (discrete) tensor, without any need to perform a discretization. This example was implemented in C++, and the code used to generate all data can be found in Listing 10 in App. B.2.1.

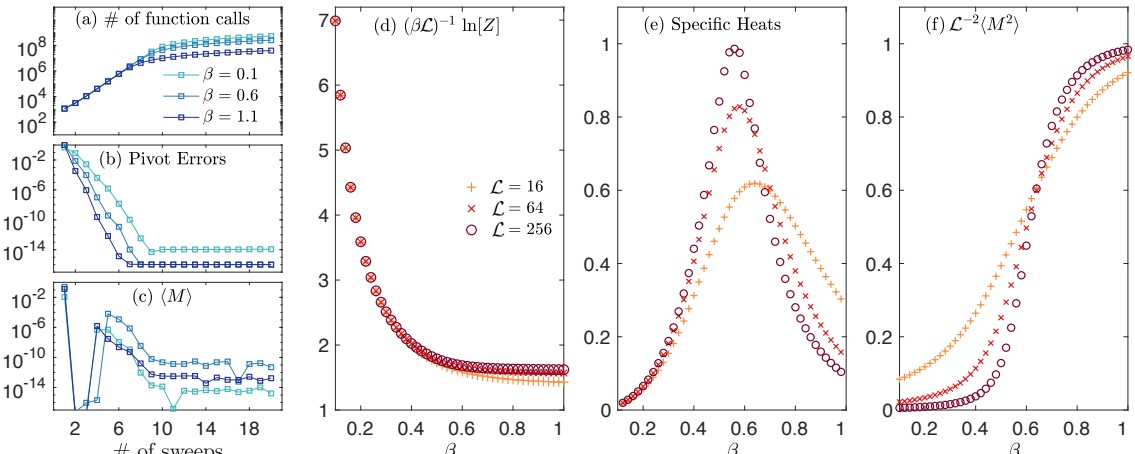

Figure 5: Unfolding the Boltzmann distribution function for the inverse square Ising chain via 2-site TCI in reset mode. The first three panels show the evolution of (a) the number of function calls, (b) pivot errors, and (c) magnetization with TCI full-sweeps for $\mathcal{L} = 64$ for $\beta = 0.1, 0.6$ and $1.1$. (d) The free energy density, (e) specific heat, and (f) the second-order moment, for $\mathcal{L} = 16, 64$ and $256$, computed for temperatures in the vicinity of the phase transition at $\beta_c \approx 0.62$. [Code: Listing 10 (C++)]

We consider the calculation of a classical partition function of the form $Z = \sum_{\sigma} W_{\sigma}$, where $W_{\sigma} = e^{-\beta E_{\sigma}}$ and $E_{\sigma}$ are the Boltzmann weight and energy, respectively, of a configuration $\boldsymbol{\sigma} = (\sigma_1, \ldots, \sigma_{\mathcal{L}})$ and $\beta = 1/T$ is the inverse temperature of the system. Once the Boltzmann weight has been put in TCI form, $W_{\sigma} \simeq \widetilde{W}_{\sigma} = \prod_{\ell=1}^{\mathcal{L}} M_{\ell}^{\sigma_{\ell}}$, the partition function can be expressed in factorized form, allowing its evaluation in polynomial time:

$$Z = \sum_{\sigma} W_{\sigma} \approx \sum_{\sigma} \widetilde{W}_{\sigma} = \sum_{\sigma} \prod_{\ell=1}^{\mathcal{L}} M_{\ell}^{\sigma_{\ell}} = \prod_{\ell=1}^{\mathcal{L}} \sum_{\sigma_{\ell}} M_{\ell}^{\sigma_{\ell}}. \tag{66}$$

This direct access to $Z$ stands in contrast to Monte Carlo approaches: these typically evaluate ratios of sums, giving easy access only to observables such as magnetization but not directly to the partition function itself. From $Z$, one can calculate the free energy per site, $F = (\beta \mathcal{L})^{-1} \ln Z$, and the specific heat, $C = \beta^2 \frac{\partial \ln Z}{\partial \beta^2}$ (evaluated through finite differences). Other quantities can also be calculated directly using appropriate weights.

Our example is a ferromagnetic Ising chain with a long-range interaction decaying as the inverse square of the distance. The energy of a configuration reads

$$E_{\sigma} = -\sum_{\ell < \ell'} J_{\ell\ell'} \sigma_{\ell} \sigma_{\ell'}, \tag{67}$$

where $\sigma_{\ell} = \pm 1$ is a classical spin variable at site $\ell$ and $J_{\ell\ell'} = |\ell - \ell'|^{-2}$ the coupling constant between sites $\ell$ and $\ell'$. This system is sufficiently complex to display a Kosterlitz-Thouless transition [55–57] at $\beta_c \approx 0.62$. Beyond the free energy, we also calculate the magnetization $M = \sum_{\ell=1}^{\mathcal{L}} \sigma_{\ell}/\mathcal{L}$ and its variance, using suitably modified versions of Eq. (66).

In Figs. 5(a–c), we first inspect the accuracy of the TCI at three different temperatures with $\mathcal{L} = 64$. Fig. 5(a) shows the accumulated number of function calls to the Boltzmann weight $W_{\sigma}$ over several sweeps. The total number of function calls initially grows exponentially, then the growth slows down significantly once the TCI's pivot error [Fig. 5(b)] approaches convergence. In Fig. 5(c), we see that irrespective of $\beta$, the average of the on-site magnetization reduces to almost zero (smaller than $10^{-8}$) when the TCI's pivot error is sufficiently small.

This is due to symmetry since we did not use a (small) magnetic field to break the global $Z_2$ symmetry of the problem. To preserve this symmetry during TCI, we start the algorithm with two global pivots: $\boldsymbol{\sigma} = (1, 1, \ldots, 1)$ and $(-1, -1, \ldots, -1)$. This is very important at low temperature. Indeed, if we use only a single global pivot, then the pivot exploration gets stuck in the corresponding sector and we obtain the same result as if we *had* broken the symmetry with a small magnetic field. Even though the initial pivots correspond to fully polarized configurations ($\beta \to \infty$), TCI converges well at all temperatures, including in the paramagnetic phase. This is a indication of the robustness of the algorithm.

Figures 5(d–f) compare physical observables, such as the free energy, the specific heat, and the second magnetic moment, for $\mathcal{L} = 16$, 64 and 256. The smoothness of the free energy curve versus $\beta$ [Fig. 5(d)] rules out the possibility of a first-order transition. Yet a phase transition is clearly seen in Fig. 5(e), as the specific heat develops an increasingly sharp peak when increasing the system size. Figure 5(f), showing the second magnetic moment, likewise indicates that a phase transition occurs at $\beta \approx 0.62$, where the three sets of data points for different system sizes intersect.

# 6 Application: Quantics representation of functions

When working with functions $f(\mathbf{x})$ for which a very high resolution of the variables $\mathbf{x}$ is desired, e.g. functions having structures with widely different length scales, using the *quantics tensor representation* [18,19] can be advantageous. It achieves exponential resolution by representing the function variables $\mathbf{x}$ through binary digits $\boldsymbol{\sigma}$. The resulting binary representation of the function can be viewed as a tensor, $F_{\boldsymbol{\sigma}} = f(\mathbf{x}(\boldsymbol{\sigma}))$. Many functions are represented by a low-rank tensor, including some functions involving vastly different scales [15,21,49]. This section discusses various applications of quantics TCI.

## 6.1 Definition

We begin by discussing the quantics representation of a function of one variable, $f(x)$. The variable is rescaled such that $x \in [0, 1)$ and discretized on a uniform grid $x(m) = m/M$, with $M = 2^{\mathcal{R}}$ and $m = 0, 1, \ldots, M - 1$. We express the grid index $m$ in binary form using $\mathcal{R}$ bits $\sigma_r \in \{0, 1\}$ as follows (the second expression is standard binary notation)

$$m(\sigma_1, \ldots, \sigma_{\mathcal{R}}) = (\sigma_1 \sigma_2 \cdots \sigma_{\mathcal{R}})_2 \equiv \sum_{r=1}^{\mathcal{R}} \sigma_r 2^{\mathcal{R}-r}. \tag{68}$$

We define $\boldsymbol{\sigma} = (\sigma_1, \ldots, \sigma_{\mathcal{R}})$ and $x(\boldsymbol{\sigma}) = x(m(\boldsymbol{\sigma}))$. Bit $\sigma_r$ now resolves $x$ at the scale $2^{-r}$. Thus, the discretized function $f$ is a tensor $F_{\boldsymbol{\sigma}} = f(x(\boldsymbol{\sigma}))$, the *quantics* representation of $f$. It has $\mathcal{L} = \mathcal{R}$ indices, each of dimension $d = 2$.

For a function of $\mathcal{N}$ variables, $f(\mathbf{x}) = f(x_1, \ldots, x_{\mathcal{N}})$, we rescale and discretize each variable as $x_n(m_n) = m_n/M = m_n/2^{\mathcal{R}}$, then express $m_n$ through $\mathcal{R}$ bits $\sigma_{nr} \in \{0, 1\}$ as

$$m_n(\sigma_{n1}, \ldots, \sigma_{n\mathcal{R}}) = (\sigma_{n1} \sigma_{n2} \cdots \sigma_{n\mathcal{R}})_2 = \sum_{r=1}^{\mathcal{R}} \sigma_{nr} 2^{\mathcal{R}-r}. \tag{69}$$

The vector $\mathbf{x}$ is represented by a tuple of $\mathcal{L} = \mathcal{N}\mathcal{R}$ bits, where bit $\sigma_{nr}$ resolves $x_n$ at the scale $2^{-r}$. The rank of the tensor train $\widetilde{F}_{\boldsymbol{\sigma}}$ obtained by unfolding $F_{\boldsymbol{\sigma}}$ can strongly depend on the way we order the different bits. In the *interleaved* quantics representation, we group all the bits that address the same scale together and relabel the bits as $\boldsymbol{\sigma} = (\sigma_1, \ldots, \sigma_{\mathcal{L}})$, with $\sigma_{\ell(n,r)} = \sigma_{nr}$

and $\ell = n+(r-1)\mathcal{N} = 1, \ldots, \mathcal{L}$, such that

$$
F_{\boldsymbol{\sigma}} = f(\mathbf{x}(\boldsymbol{\sigma})) =
$$

$$
=
\tag{70}
$$

If the variables at the same scale are strongly entangled, which is the case in many physical applications, using the interleaved quantics representation can lead to a more compressible tensor [15, 18, 19, 21]. An alternative is the *fused quantics* representation, $F_{\tilde{\boldsymbol{\sigma}}} = f(\mathbf{x}(\tilde{\boldsymbol{\sigma}}))$, where we "fuse" all bits for scale $2^{-r}$ into a single variable

$$
\tilde{\sigma}_r = (\sigma_{\mathcal{N}r} \cdots \sigma_{2r} \sigma_{1r})_2 = \sum_{n=1}^{\mathcal{N}} 2^{n-1} \sigma_{nr},
\tag{71}
$$

taking the values $0, \ldots, 2^{\mathcal{N}}-1$. One can also arrange these variables as $\tilde{\boldsymbol{\sigma}} = (\tilde{\sigma}_1, \ldots, \tilde{\sigma}_{\mathcal{R}})$. One can also group together all bits addressing a given variable $x_n$, as done in the natural representation.

Once a quantics representation $F$ of $f$ has been defined, TCI can be applied to $F$ to obtain a tensor train $\tilde{F}$ interpolating $f$ with exponential resolution. We dub this algorithm *quantics TCI* (QTCI), and the resulting tensor train a *quantics tensor train* (QTT) [18–20].

Some simple analytic functions are approximated well as a QTT with $\chi < 10$. For instance, a pure exponential, $f(x) = e^{\lambda x}$, has $\chi = 1$, since its quantics tensor factorizes completely, $F_{\boldsymbol{\sigma}} = \prod_{r=1}^{\mathcal{R}} e^{\lambda \sigma_r 2^{\mathcal{R}-r}}$. Similarly, sine and cosine functions have $\chi = 2$, since they can be expressed as sums of two exponentials, i.e. sums of two rank-1 tensors. Some discontinuous functions likewise have low-rank in quantics representations, such as the Dirac delta ($\chi = 1$) and Heaviside step function ($\chi = 2$) [19]. By contrast, random noise is incompressible and leads to $\chi \sim d^{\mathcal{L}/2}$. More generally, if a function has low quantics rank $\chi$, the sites representing different scales are not strongly "entangled". In this sense, the quantics rank of a function quantifies the degree of scale separation inherent in the function [15, 21].

An interesting example of low-rank analytic functions of two variables is the Kronecker delta function $f(m_1, m_2) = \delta_{m_1 m_2}$ defined on a discrete 2D grid. Its matrix representation, the $2^{\mathcal{R}} \times 2^{\mathcal{R}}$ unit matrix, is incompressible (in the sense of SVD) because all its singular values are 1. In the quantics representation, $f(m_1, m_2) = \delta_{\sigma_{11}\sigma_{21}} \cdots \delta_{\sigma_{1r}\sigma_{2r}} \cdots \delta_{\sigma_{1\mathcal{R}},\sigma_{2\mathcal{R}}}$, which can be regarded as a rank-1 MPS by fusing $\sigma_{1r}$ and $\sigma_{2r}$.

Other examples for functions of multiple variables that can be approximated as a low-rank QTT are multivariate analogues of the 1D examples above, with $\mathbf{x} \in \mathbb{R}^{\mathcal{N}}$: a single exponential $f(\mathbf{x}) = \exp(\mathbf{v} \cdot \mathbf{x})$ with arbitrary $\mathbf{v}$ has bond dimension $\chi = 1$; a Dirac delta $\delta(\mathbf{x})$ reduces to the Kronecker delta above and therefore has bond dimension $\chi = 1$ as well; a step function $f(\mathbf{x}) = \theta(\mathbf{v} \cdot \mathbf{x} - \mathbf{b})$ for given $\mathbf{v}$ and $\mathbf{b}$ has bond dimension $\chi = 2$. In all examples mentioned here, the small bond dimension is due to separability of length scales. An example where length scales are not separable is the function $f(\mathbf{x}) = \theta(1 - \|\mathbf{x}\|_2)$, which is equal to 1 inside the unit sphere and 0 outside. Since the surface of the sphere is curved, the maximum bond dimension, $\chi_{\max}$, needed to represent this function with a QTT will depend on the resolution with which the surface is resolved, increasing as the resolution is refined. This is illustrated in Fig. 6 for the case $\mathcal{N} = 2$.

## 6.2 Operating on quantics tensor trains

Given a function represented by a quantics tensor train, various operations on these functions can be performed within the tensor train form. In the following, we describe how to calculate integrals, convolutions and symmetry transforms within the quantics representation; quantics

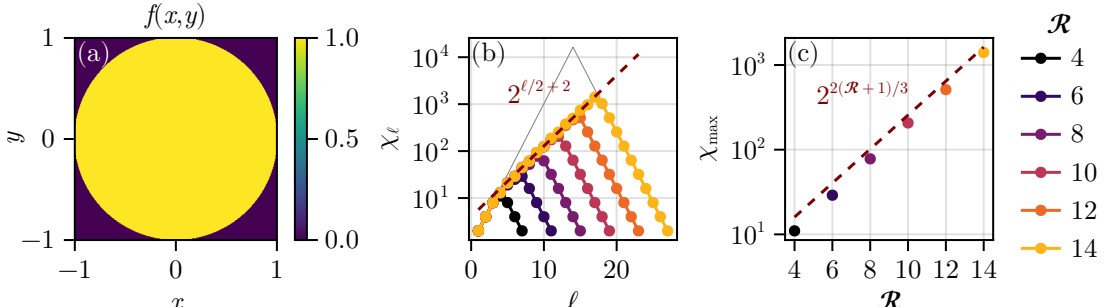

Figure 6: (a) The function $f(\mathbf{x}) = \theta(1 - \|\mathbf{x}\|_2)$, in $\mathcal{N} = 2$ dimensions. (b) The bond dimension $\chi_\ell$ of a QTT representation of $f$ with interleaved index ordering, plotted for several values of $\mathcal{R}$. Along the chain, the bond dimension scales as $\chi_\ell \sim 2^{\ell/2}$. Intuitively, this is because each additional pair of bits $\sigma_{1r}, \sigma_{2r}$ doubles the number of points close to the circle, which are those that contain additional information. (c) The maximal bond dimension, $\chi_{\max}$, increases exponentially with $\mathcal{R}$, as $\chi_{\max} \approx 2^{2(\mathcal{R}+1)/3}$. This behavior is independent of the specified tolerance, because the step function changes abruptly. If the step function is broadened, the maximum bond dimension decreases significantly, in a manner depending on the tolerance.

Fourier transforms are described in detail in Sec. 6.2. In addition, the methods for element-wise operations and addition of tensor trains that have already been introduced in Sec. 4.7 work just as well here. These basic 'building blocks' can be combined to formulate more complicated algorithms entirely within the quantics tensor train form.

**Integrals** are approximated as Riemann sums, then factorized over the quantics bits as

$$\int_{[0,1]^{\mathcal{N}}} d^{\mathcal{N}}\mathbf{x} f(\mathbf{x}) \approx \frac{1}{2^{\mathcal{L}}} \sum_{\boldsymbol{\sigma}} f(\mathbf{x}(\boldsymbol{\sigma})) = \frac{1}{2^{\mathcal{L}}} \sum_{\boldsymbol{\sigma}} F_{\boldsymbol{\sigma}} \approx \frac{1}{2^{\mathcal{L}}} \sum_{\boldsymbol{\sigma}} \widetilde{F}_{\boldsymbol{\sigma}} = \frac{1}{2^{\mathcal{L}}} \prod_{\ell=1}^{\mathcal{L}} \left[ \sum_{\sigma_\ell=1}^{2} M_\ell^{\sigma_\ell} \right], \quad (72)$$

where $1/2^{\mathcal{L}}$ is the integration volume element. Since the number of discretization points is exponential in $\mathcal{L}$, the discretization error of this integral decreases as $O(1/2^{\mathcal{L}})$, whereas the cost of the factorized sum is $\mathcal{O}(\chi^2 d\mathcal{L})$, i.e. linear in $\mathcal{L}$.

**Matrix products** of the form $f(\mathbf{x}, \mathbf{z}) = \int_D d^{\mathcal{N}}\mathbf{y}\, g(\mathbf{x}, \mathbf{y}) h(\mathbf{y}, \mathbf{z})$ can be performed as follows. We use quantics representations for each of the variables $\mathbf{x}$, $\mathbf{y}$ and $\mathbf{z}$, e.g. $\mathbf{x} = \mathbf{x}(\boldsymbol{\sigma}_x)$ with $\boldsymbol{\sigma}_x = (\sigma_{1x}, \ldots, \sigma_{\mathcal{L}x})$ and $\mathcal{L} = \mathcal{N}\mathcal{R}$. We unfold the tensors for $g$ and $h$ as MPOs,

$$\widetilde{G}_{\boldsymbol{\sigma}_x \boldsymbol{\sigma}_y} = \begin{array}{c} \sigma_{1x} \quad \sigma_{2x} \quad \cdots \quad \sigma_{\mathcal{L}x} \\ \times\!\!-\!\!\bullet\!\!-\!\!\bullet\!\!-\!\!\bullet\!\!-\!\!\bullet\!\!-\!\!\times \\ \sigma_{1y} \quad \sigma_{2y} \quad \cdots \quad \sigma_{\mathcal{L}y} \end{array}, \qquad \widetilde{H}_{\boldsymbol{\sigma}_y \boldsymbol{\sigma}_z} = \begin{array}{c} \sigma_{1y} \quad \sigma_{2y} \quad \cdots \quad \sigma_{\mathcal{L}y} \\ \times\!\!-\!\!\bullet\!\!-\!\!\bullet\!\!-\!\!\bullet\!\!-\!\!\bullet\!\!-\!\!\times \\ \sigma_{1z} \quad \sigma_{2z} \quad \cdots \quad \sigma_{\mathcal{L}z} \end{array}, \qquad (73)$$

with indices at matching scales, $(\sigma_{\ell x}, \sigma_{\ell y})$ or $(\sigma_{\ell y}, \sigma_{\ell z})$, assigned to the same site $\ell$. We then approximate the integral $\int d^{\mathcal{N}}\mathbf{y}$ by a factorized sum over each $\sigma_{\ell y}$, cf. (72), $f(\mathbf{x}, \mathbf{y}) \approx 2^{-\mathcal{L}} \sum_{\boldsymbol{\sigma}_y} \widetilde{G}_{\boldsymbol{\sigma}_x \boldsymbol{\sigma}_y} \widetilde{H}_{\boldsymbol{\sigma}_y \boldsymbol{\sigma}_z}$, which can be computed and compressed in several ways, see Sec. 4.7.

**Quantics Fourier transform** can be performed using a simple MPO-MPS contraction, where the MPO is of surprisingly low rank ($\chi \approx 11$ for machine precision in one dimension) [21, 58]. This means that taking the Fourier transform of a function that has a low-rank quantics tensor train can be done exponentially faster than with FFT. Calculating $\hat{f}(\mathbf{k}) = \int d\mathbf{x} f(\mathbf{x}) e^{-i\mathbf{k}\cdot\mathbf{x}}$

on a quantics tensor train representing $f(\mathbf{x})$ is equivalent to the quantum Fourier transform algorithm ubiquitous in quantum computing [18].

Consider a discrete function $f_m \in \mathbb{C}^M$, e.g. the discretization, $f_m = f(x(m))$, of a one-dimensional function $f(x)$ on a grid $x(m)$. Its discrete Fourier transform (DFT) is

$$\hat{f}_k = \sum_{m=0}^{M-1} T_{km} f_m, \qquad T_{km} = \tfrac{1}{\sqrt{M}} e^{-i2\pi k \cdot m/M}. \tag{74}$$

For a quantics grid, $M = 2^{\mathcal{R}}$ is exponentially large and the DFT exponentially expensive to evaluate. We seek a quantics tensor train representing $T$, because then $\hat{f} = Tf$ can be computed by simply contracting the tensor trains for $T$ and $f$ and recompressing [18, 19, 21].

We start by expressing $m$ and $k$ in their quantics form

$$m(\boldsymbol{\sigma}) = (\sigma_1 \sigma_2 \cdots \sigma_{\mathcal{R}})_2 = \sum_{\ell=1}^{\mathcal{R}} \sigma_\ell 2^{\mathcal{R}-\ell}, \qquad k(\boldsymbol{\sigma}') = (\sigma'_1 \sigma'_2 \cdots \sigma'_{\mathcal{R}})_2 = \sum_{\ell'=1}^{\mathcal{R}} \sigma_{\ell'} 2^{\mathcal{R}-\ell'}. \tag{75}$$

Then, $T$ has the quantics representation

$$T_{\boldsymbol{\mu}} = T_{\boldsymbol{\sigma}'\boldsymbol{\sigma}} = T_{k(\boldsymbol{\sigma}')m(\boldsymbol{\sigma})} = \tfrac{1}{\sqrt{M}} \exp\left[-i2\pi \sum_{\ell\ell'} 2^{\mathcal{R}-\ell'-\ell} \sigma_{\ell'}' \sigma_\ell\right], \tag{76}$$

where we introduced the fused index $\boldsymbol{\mu} = (\mu_1, \ldots \mu_{\mathcal{R}})$, with $\mu_\ell = (\sigma'_{\mathcal{R}-\ell+1}, \sigma_\ell)$. We thereby arrange $\boldsymbol{\sigma}'$ and $\boldsymbol{\sigma}$ indices in *scale-reversed* order [21], so that $\sigma'_{\mathcal{R}-\ell+1}$, describing the scale $2^{\ell-1}$ in the $k$ domain, is fused with $\sigma_\ell$, describing the scale $2^{\mathcal{R}-\ell}$ in the $m$ domain, in accordance with Fourier reciprocity (small $k$ scales match large $m$ scales and vice versa):

$$
\begin{array}{ccc}
\overline{T_{\boldsymbol{\mu}}} & = & \begin{array}{ccccccc} \sigma_1 & \sigma_2 & \cdots & \sigma_\ell & \cdots & \sigma_{\mathcal{R}-1} & \sigma_{\mathcal{R}} \\ \hline \sigma'_{\mathcal{R}} & \sigma'_{\mathcal{R}-1} & \cdots & \sigma'_{\mathcal{R}-\ell+1} & \cdots & \sigma'_2 & \sigma'_1 \end{array} \\
\mu_1 \quad \mu_2 \quad \cdots \quad \mu_\ell \quad \cdots \quad \mu_{\mathcal{R}-1} \quad \mu_{\mathcal{R}} & &
\end{array} \tag{77a}
$$

The tensor $T_{\boldsymbol{\mu}}$ turns out to have a remarkably *low rank* [21, 58]: when unfolded as a MPO $\widetilde{T}_{\boldsymbol{\mu}}$,

 (77b)

a rank of $\chi = 11$ suffices to yield machine precision, i.e. errors $|T_{\boldsymbol{\mu}} - \widetilde{T}_{\boldsymbol{\mu}}|_\infty / |T_{\boldsymbol{\mu}}|_\infty < 10^{-10}$, irrespective of $\mathcal{R}$ [21, 58]. By contrast, if a scale-reversed order is not used (i.e., $\mu_\ell = (\sigma'_\ell, \sigma_\ell)$), the resulting tensor $T_{\boldsymbol{\mu}}$ has exponentially large rank [59]. An intuitive explanation for the scale-reversed order is given in Appendix A.5, which also verifies through numerical experiment that the small-rank representation is found by TCI.

It follows that for a 1D function with rank $\chi'$ in quantics representation, the DFT can be obtained in $O(\chi^2 \chi'^2 \mathcal{R}) = O(\chi^2 \chi'^2 \log M)$ operations, where $M = 2^{\mathcal{R}}$ is the number of points in the grid. This is exponentially faster than the $O(M \log M)$ of the fast Fourier transform [21, 58, 60].

## 6.3 Example: High-resolution compression of functions

In this section we illustrate the use of quantics TCI for representing functions in 1, 2 and 3 dimensions, and for computing multi-dimensional integrals.

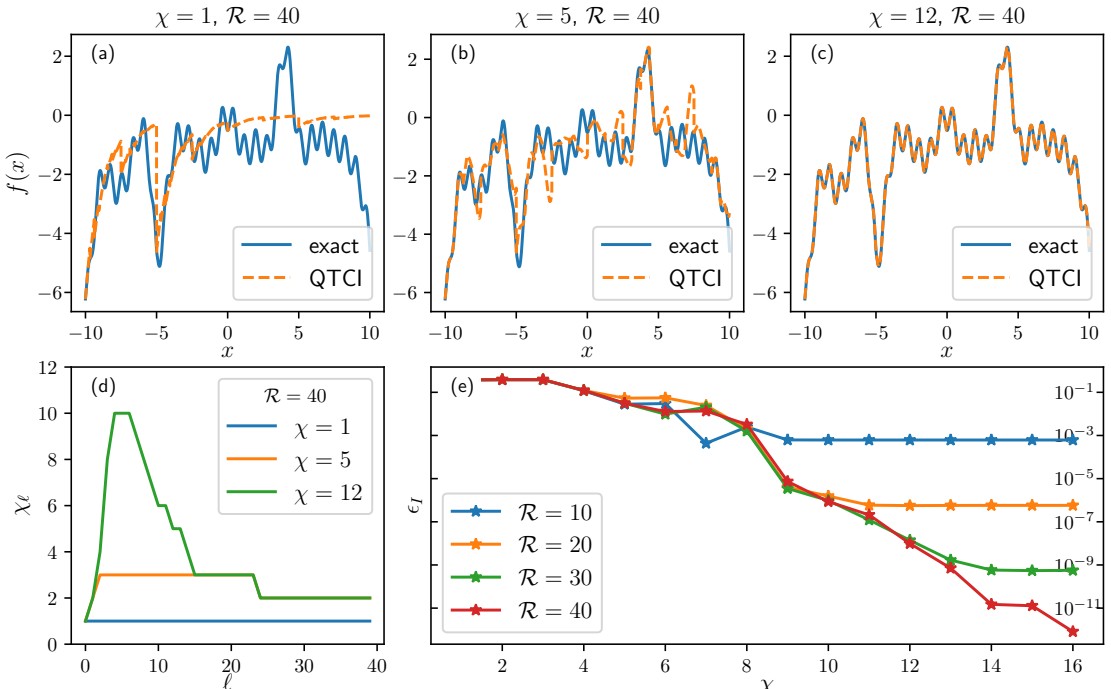

Figure 7: (a-c) The function $f(x)$ of Eq. (78) (solid blue) and its quantics representation (orange dashed) with $\mathcal{R} = 40$ for $\chi = 1$, 5 and 12. Although the TCI is performed on $2^{40} \approx 10^{12}$ points, only a small fraction are actually shown in the plot. (d) Bond dimension $\chi_\ell$ as a function of $\ell$, for $\chi = 1, 5, 12$. (e) Error on the integral $I = \int_{-10}^{10} dx f(x)$ calculated from its QTCI approximation, $\widetilde{I}$. The plot shows the relative error $\epsilon_I = |\widetilde{I}/I - 1|$, plotted versus the rank $\chi$ of the QTCI approximation, for $\mathcal{R} = 10, 20, 30, 40$. [Code: Listing 2 (Python), 3 (Julia)]

### 6.3.1 Oscillating functions in 1, 2 and 3 dimensions

**1d oscillating function** As a first example, we consider a simple function with large oscillations:

$$f(x) = \text{sinc}(x) + 3e^{-0.3(x-4)^2}\,\text{sinc}(x-4) - \cos(4x)^2 - 2\,\text{sinc}(x+10)e^{-0.6(x+9)}$$
$$+ 4\cos(2x)e^{-|x+5|} + \frac{6}{x-11} + \sqrt{(|x|)}\arctan(x/15), \tag{78}$$

where $\text{sinc}(x) = \sin x/x$ is the sinus cardinal and $x \in [-10, 10]$.

The code of Listing 2 discretizes the function on a quantics grid of $2^{40}$ points $\{x(\boldsymbol{\sigma})\}$, defines the tensor $F_{\boldsymbol{\sigma}} = f(x(\boldsymbol{\sigma}))$ and uses xfac to TCI it, $\widetilde{F}_{\boldsymbol{\sigma}} \approx F_{\boldsymbol{\sigma}}$. Listing 3 shows TCI.jl code performing the same task. Figs. 7(a–c) show the resulting QTCI approximation: it converges very quickly with increasing $\chi$. Fig. 7(d) shows the bond dimension $\chi_\ell$ as a function of $\ell$. It remains small ($\leq 10$), hence the function is strongly compressible. Figure 7(e) shows that the integral $I = \int_{-10}^{+10} dx f(x)$ converges rapidly with increasing $\chi$ even though the function is highly oscillatory.

```
1  import xfacpy
2  import numpy as np
3
4  # Grid parameters
5  R = 40                          # Number of bits <@$\cR$@>
6  M = 2**R                        # Number of grid points <@$M$@>
```

```
 7  xmin, xmax = -10.0, +10.0   # Domain of function <@$f$@>
 8
 9
10  def m_to_sigma(m):          # Convert grid index <@$m$@> to quantics multi-index
        <@$\bsigma(m)$@>
11      return [int(k) for k in np.binary_repr(m, width=R)]
12
13
14  def sigma_to_x(sigma):      # Convert quantics multi-index <@$\bsigma$@> to grid point
        <@$x(\bsigma)$@>
15      ind = int(''.join(map(str, sigma)), 2)
16      return xmin + (xmax-xmin)*ind/M
17
18
19  def f(x):                   # Function of interest <@$f(x)$@>
20      return (np.sinc(x)+3*np.exp(-0.3*(x-4)**2)*np.sinc(x-4)-np.cos(4*x)**2 -
21              2*np.sinc(x+10)*np.exp(-0.6*(x+9))+4*np.cos(2*x)*np.exp(-abs(x+5)) +
22              6*1/(x-11)+abs(x)**0.5*np.arctan(x/15))
23
24
25  def f_tensor(sigma):        # Quantics tensor <@$F_\bsigma$@>
26      return f(sigma_to_x(sigma))
27
28
29  # Set first pivot to <@$\bar\bsigma=(0, \ldots, 0)$@> and initialize TCI <@$\tF_\bsigma$@>
30  p = xfacpy.TensorCI1Param()
31  p.pivot1 = [0 for ind in range(R)]
32  f_tci = xfacpy.TensorCI1(f_tensor, [2]*R, p)
33
34  # Optimize <@$\tF_\bsigma$@>
35  for sweep in range(12):
36      f_tci.iterate()         # Perform a half sweep
37
38  f_tt = f_tci.get_TensorTrain()  # Obtain the TT <@$M_1 M_2 \ldots M_\scR$@>
39  # Print a table to compare <@$f(x)$@> and <@$\tF_\bsigma$@> on some regularly spaced
        points
40  print("x\t f(x)\t f_tt(x)")
41  for m in range(0, M, 2**(R-5)):
42      sigma = m_to_sigma(m)
43      x = xmin + (xmax-xmin)*m/M
44      print(f"{x}\t{f(x)}\t{f_tt.eval(sigma)}")
```

Listing 2: Python code using xfac to compute construct a quantics tensor train for the function $f(x)$ of Eq. (78), shown in Fig. 7, using $2^{40}$ grid points. Note that this code could also have used pre-defined functions that are part of xfac to generate quantics grids $x(\boldsymbol{\sigma})$ and convert between $x$, $m$, and $\boldsymbol{\sigma}$, see App. B.1.3.

**2d oscillating function**   The above construction generalizes straightforwardly to more than one dimension. Let us consider the following simple 2d function with features at vastly different scales:

$$f(x,y) = 1 + e^{-0.4(x^2+y^2)} + \sin(xy)e^{-x^2} + \cos(3xy)e^{-y^2} + \cos(x+y) \tag{79}$$
$$+ 0.05\cos\left[10^2 \cdot (2x-4y)\right] + 5 \cdot 10^{-4}\cos\left[10^3 \cdot (-2x+7y)\right] + 10^{-5}\cos\left(2 \cdot 10^8 x\right).$$

We use a quantics unfolding of $f(x,y)$ with $\mathcal{R} = 40$, which discretizes $f$ on a $10^{12} \times 10^{12}$ grid. The corresponding quantics tensor $F_{\boldsymbol{\sigma}}$ has $\mathcal{L} = 2\mathcal{R}$ indices, interleaved so that even indices $\sigma_{2\ell}$ encode $x$ and odd indices $\sigma_{2\ell+1}$ encode $y$. A tensor train approximation is then obtained using standard TCI, which yields an efficient low-rank representation that rapidly converges, as shown in Fig. 8 (a–d). At rank $\chi \approx 110$, the MPS becomes a numerically exact (within machine precision) representation of the original function at all scales. It requires only $10^5$ numbers ($\sim 1$ MB of RAM), which is trivial to store in memory, and 19 orders of magnitude smaller than needed for a naive regular grid ($\sim 10^{13}$ TB of RAM). Furthermore,

```
1  using QuanticsTCI
2  import QuanticsGrids as QG
3
4  R = 40                                      # Number of bits <@$\cR$@>
5  M = 2^R                                     # Number of discretization points <@$M$@>
6  xgrid = QG.DiscretizedGrid{1}(R, -10, 10)   # Discretization grid <@$x(\bsigma)$@>
7
8  function f(x)                               # Function of interest <@$f(x)$@>
9      return (
10          sinc(x) + 3 * exp(-0.3 * (x - 4)^2) * sinc(x - 4) - cos(4 * x)^2 -
11          2 * sinc(x + 10) * exp(-0.6 * (x + 9)) + 4 * cos(2 * x) * exp(-abs(x + 5)) +
12          6 * 1 / (x - 11) + sqrt(abs(x)) * atan(x / 15))
13  end
14
15  # Construct and optimize quantics TCI <@$\tF_\bsigma$@>
16  f_tci, ranks, errors = quanticscrossinterpolate(Float64, f, xgrid; maxbonddim=12)
17  # Print a table to compare <@$f(x)$@> and <@$\tF_\bsigma$@> on some regularly spaced
       points
18  println("x\t f(x)\t\t\t f_tt(x)")
19  for m in 1:2^(R-5):M
20      x = QG.grididx_to_origcoord(xgrid, m)
21      println("$x\t$(f(x))\t$(f_tci(m))")
22  end
```

Listing 3: Julia code using `TCI.jl` to construct a quantics tensor train for $f(x)$ of Eq. (78), plotted in Fig. 7. The function `quanticscrossinterpolate` includes code to convert $f$ to quantics form, see Sec. 8.3. The `xgrid` object constructed on line 6 is a lazy object that does not create an exponentially large object.

it can be manipulated exponentially faster than for the regular grid, including most common operations such as Fourier transform, convolution or integration.

**3d integral**     Figure 9 shows the last example of this series: the computation of the 3D integral $I = \int_{\mathbb{R}^3} d^3\mathbf{x}\, e^{-\sqrt{x^2+y^2+z^2}}$ using the quantics representation. TCI in both accumulative and reset mode converges exponentially fast towards the exact integral $I = 8\pi$, almost reaching machine precision, an indication of excellent numerical stability.

### 6.3.2   Quantics for multi-dimensional integration

Let us return in this section to the example of the multi-dimensional integral from Sec. 5.2. In the initial approach, a quadrature has been choosen in order to factorize the function on the quadrature grid, and then, in a second step, to perform the one-dimensional integrations. Here, we consider the interleaved quantics representation described in Sec. 6.1, with $\mathcal{L} = \mathcal{N}\mathcal{R}$ legs of dimension $d = 2$. After obtaining the QTT from TCI, the integral can be evaluated efficiently by a factorized sum over the MPS tensors, as shown in Eq. (72). This corresponds to a Riemann sum with exponentially many discretization points. The results are shown in Fig. 10. In this example, we observe a fast convergence of the results. Note that using the fused instead of interleaved representation here would lead to $d = 2^{\mathcal{N}}$, which quickly becomes prohibitive for large $\mathcal{N}$.

Listing 8 in App. B.1.3 contains the python code (using `xfac`) yielding the results shownin Fig. 10. The code is very similar to Listing 1, but replaces the Gauss–Kronrod helper functions with corresponding functions for a quantics grid. A more detailed discussion can be found in App. B.1.3. Listing 13 contains an equivalent code using `TCI.jl`.

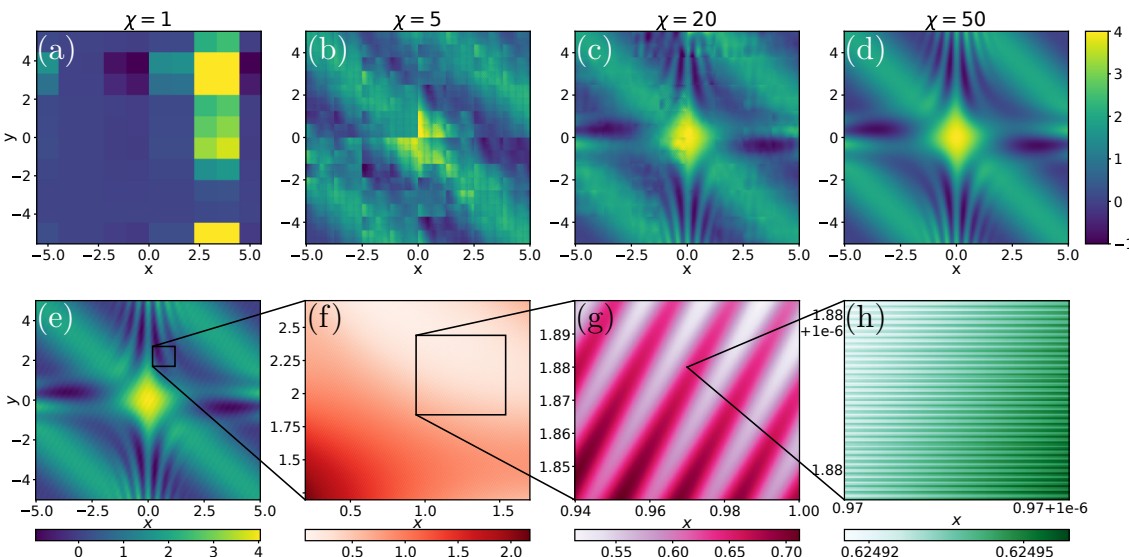

Figure 8: Quantics TCI (QTCI) representation of the function $f(x,y)$ defined in Eq. (79). (a–d) Approximations obtained for 4 different values of the MPS rank $\chi$ using $\mathcal{R} = 40$ plotted on a coarse grid of $300 \times 300$ points. (e–h) From left to right, the panels show different levels of zoom into the QTCI at $\chi = 50$ from coarse to very fine. At this rank, the compressed representation is numerically exact at all scales. [Code: Listing 7 (Python), 12 (Julia)]

## 6.4 Example: Heat equation using superfast Fourier transforms

In this section, we show how the different operations described earlier can be combined for a nontrivial application: solving a partial differential equation on a grid with exponentially many grid points [20].

Our example is the solution of the heat equation in 1D,

$$\partial_t u(x,t) = \partial_x^2 u(x,t), \tag{80}$$

with a billion grid points and a complex initial condition with features at different scales. Since its solution is trivial in Fourier space, $u(k,t) = e^{-k^2 t} u(k,0)$, our strategy is simple: put $u(x,0)$ in quantics form using TCI, Fourier transform it (in ultrafast way), evolve it up to time $t$ and Fourier transform back to real space.

We discretize the spatial variable as $x(m) = x_{\min} + m\delta$ with $\delta = (x_{\max} - x_{\min})/M$, $M = 2^{\mathcal{R}}$. Then, we view $u$ as a vector with components $u_m(t) = u(x(m), t)$, satisfying the equation

$$\partial_t u_m(t) = (u_{m-1} - 2u_m + u_{m+1})/\delta^2. \tag{81}$$

Taking the discrete Fourier transform of this equation using $u^{\text{FT}} = Tu$ one obtains

$$\partial_t u_k^{\text{FT}}(t) = -(2/\delta)^2 \sin^2(\pi k/M) u_k^{\text{FT}}(t). \tag{82}$$

For a given initial condition $u_m(0)$, with Fourier transform $u_k^{\text{FT}}(0)$, this can be solved as

$$u_k^{\text{FT}}(t) = g_k(t) u_k^{\text{FT}}(0), \qquad g_k(t) = \exp\left[-(2/\delta)^2 \sin^2(\pi k/M) t\right]. \tag{83}$$

The algorithm to solve the heat equation using the quantics representation is now straightforward. It is summarized through the following mappings:

$$u_m(0) \xrightarrow{\text{QTCI}} \widetilde{U}_{\boldsymbol{\sigma}}(0), \qquad g_k(t) \xrightarrow{\text{QTCI}} \widetilde{G}_{\boldsymbol{\sigma}'}(t), \qquad T_{km} \xrightarrow{\text{QTCI}} \widetilde{T}_{\boldsymbol{\sigma}'\boldsymbol{\sigma}}, \tag{84a}$$

$$\widetilde{U}_{\boldsymbol{\sigma}}(0) \xrightarrow{\times \widetilde{T}_{\boldsymbol{\sigma}'\boldsymbol{\sigma}}} \widetilde{U}_{\boldsymbol{\sigma}'}^{\text{FT}}(0) \xrightarrow{\times \widetilde{G}_{\boldsymbol{\sigma}'}(t)} \widetilde{U}_{\boldsymbol{\sigma}'}^{\text{FT}}(t) \xrightarrow{\times \widetilde{T}_{\boldsymbol{\sigma}\boldsymbol{\sigma}'}^{-1}} \widetilde{U}_{\boldsymbol{\sigma}}(t). \tag{84b}$$

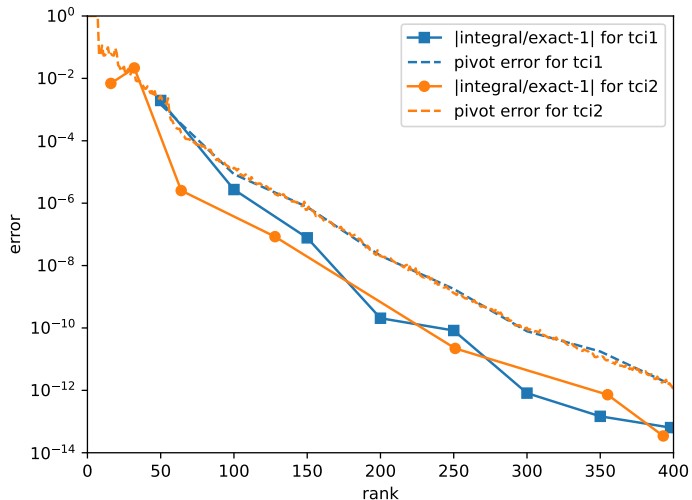

Figure 9: Consider the 3D integral $I = \int_{\mathbb{R}^3} d^3\mathbf{x}\, e^{-\sqrt{x^2+y^2+z^2}} = 8\pi$. We use `xfac` to compute its QTCI approximation, $\widetilde{I}$, on a uniform grid of $2^{3\mathcal{R}}$ points with $\mathcal{R} = 30$ in the cube $[-40, 40]^3$. The plot shows the error $\epsilon_I = |\widetilde{I}/I - 1|$ (solid lines with symbols) and the pivot error (dashed lines) as function of the MPS rank $\chi$, computed using accumulative mode (blue) and reset mode (orange).

By Eq. (84a), we first QTCI all relevant objects; by Eq. (84b), we then Fourier transform the initial condition, time-evolve it in momentum space, and then Fourier transform it back to position space. The third step of Eq. (84b) involves element-wise multiplication of two tensor trains, $\widetilde{U}^{\mathrm{FT}}_{\sigma'}(t) = \widetilde{G}_{\sigma'}(t)\widetilde{U}^{\mathrm{FT}}_{\sigma'}(0)$, performed separately for every $\sigma'$. Note that the application of the tensor train operators $\widetilde{T}$ and $\widetilde{T}^{-1}$ are understood to each be followed by TCI recompressions.

We consider an initial condition with tiny, rapid oscillations added to a large, box-shaped background described by Heaviside $\theta$-functions:

$$u(x,0) = \tfrac{1}{100}\left[1 + \cos(120x)\sin(180x)\right] + \theta\left(x - \tfrac{7}{2}\right)\left[1 - \theta\left(x - \tfrac{13}{2}\right)\right]. \tag{85}$$

Figure 11 shows the subsequent solution $u(x,t)$ at several different times. With increasing time, the initial oscillations die out (see inset) and in the long-time limit diffusive spreading is observed, as expected. The computation was performed for $\mathcal{R} = 30$, implying a very dense grid with $M = 2^{30}$ points, beyond the reach of usual numerical simulation techniques. Remarkably, however, the computational costs scale only linearly (not exponentially!) with $\mathcal{R}$. Indeed, even though the grid has around one billion points, obtaining the solution for one value of the time takes about one second on a single computing core.

The python code used to produce the data for Fig. 11 is shown in listing 9, App. B.1.4.

# 7 Application: Matrix product operators (MPOs)

A linear tensor operator $H_{\sigma'\sigma}$ can be unfolded into a matrix product operator (MPO) (also known as tensor train operator) using TCI. This is done by grouping the input and output indices together, $\mu_\ell = (\sigma'_\ell, \sigma_\ell)$, and performing TCI on the resulting tensor train $F_\mu \equiv H_{\sigma'\sigma}$.

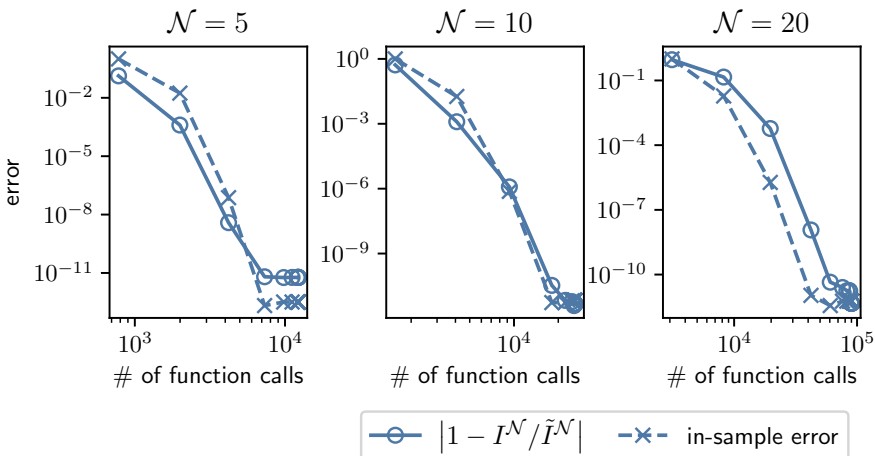

Figure 10: Performance metrics for the TCI2 computation of the integral $I^{(\mathcal{N})}$ of Eq. (64), for $\mathcal{N} = 5, 10, 20$, (left, middle, right). In all panels, the relative error $|1-I^{(\mathcal{N})}/\widetilde{I}^{(\mathcal{N})}|$ (straight lines with circles) and the relative in-sample error $|1-F_{\boldsymbol{\sigma}}/\widetilde{F}_{\boldsymbol{\sigma}}|$ (dashed lines with crosses) is plotted versus the number of function evaluations. The TCI2 computation of $\widetilde{I}^{(\mathcal{N})}$ has been obtained using $\mathcal{R} = 40$ quantics bits per variable in interleaved representation and a maximal bond dimension $\chi = 30$. [Code: Listing 8 (Python), 13 (Julia)]

One obtains an MPO, $H \approx \widetilde{H} = \prod_{\ell=1}^{\mathcal{L}} W_\ell$, with tensor elements of the form

$$[H]_{\boldsymbol{\sigma}'\boldsymbol{\sigma}} \approx \Big[\prod_{\ell=1}^{\mathcal{L}} W_\ell\Big]_{\boldsymbol{\sigma}'\boldsymbol{\sigma}} = [W_1]_{1i_1}^{\sigma_1'\sigma_1}[W_2]_{i_1 i_2}^{\sigma_2'\sigma_2}\cdots[W_{\mathcal{L}}]_{i_{\mathcal{L}-1}1}^{\sigma_{\mathcal{L}}'\sigma_{\mathcal{L}}} = \quad . \tag{86}$$

In this section, we discuss a specific algorithm to perform this unfolding for the construction of the Hamiltonian MPO for quantum many-body problems. This construction is the first step of a DMRG many-body calculation. For this application, the Hamiltonian $H$ is very sparse and a naive usage of TCI may fail there due to the ergodicity problem discussed in section 4.3.6. To avoid this issue, the algorithm and associated code (C++ header `autompo.h`) discussed below generates a MPO representation from a sum of rank-1 terms using element-wise tensor addition.

## 7.1 Formulation of the problem

Consider an $\mathcal{L}$-site quantum system whose many-body Hamiltonian is the sum of $N_H$ rank-1 MPOs $H_a$,

$$H = \sum_{a=1}^{N_H} H_a\,, \qquad H_a = \prod_{\ell=1}^{\mathcal{L}} H_{a\ell}\,, \tag{87}$$

where each $H_{a\ell}$ is a local operator acting non-trivially only on site $\ell$ (see Eqs. (91) or (93) below for examples). Each term $H_a$ in the sum is, by construction, a MPO of rank 1, but their sum $\sum_a H_a$ is not. The number of terms in the sum typically is exponentially smaller than the size of the Hilbert space in which the Hamiltonian lives, hence the operator of interest is very sparse. For instance, in quantum chemistry applications involving, say, $\mathcal{L}$ spin-orbitals, the number of terms is $\mathcal{O}(\mathcal{L}^4)$ while the size of the Hilbert space is $2^{\mathcal{L}}$. Naively, $H$ is an MPO

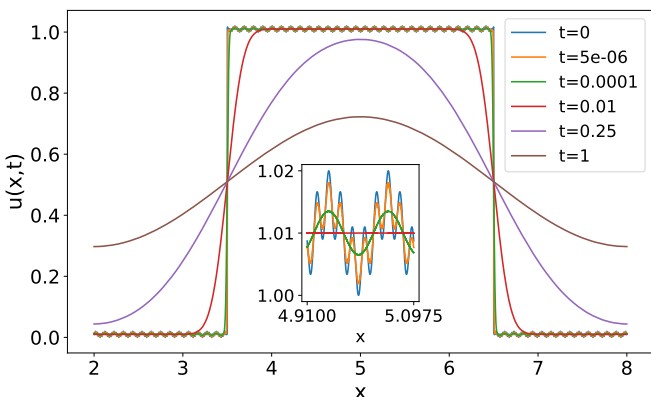

Figure 11: Solution of the heat equation (80) using quantics TCI. The plot shows $u(x, t)$ versus $x$ for different times. We used a 1D grid with $M = 2^{\mathcal{R}}$ points and $\mathcal{R} = 30$, at a computational cost of $\mathcal{O}(\mathcal{R})$. The inset shows a zoom close to $x = 5$.

of rank $N_H$ as one may express it as $\prod_{\ell=1}^{\mathcal{L}} W_\ell$, with

$$W_1 = (H_{1\mathcal{L}}, \ldots, H_{N_H \mathcal{L}}), \qquad W_{1 < \ell < \mathcal{L}} = \begin{pmatrix} H_{1\ell} & 0 & \ldots & \ldots \\ 0 & H_{2\ell} & 0 & \ldots \\ \ldots & \ldots & \ldots & \ldots \\ \ldots & \ldots & 0 & H_{N_H \ell} \end{pmatrix}, \qquad W_{\mathcal{L}} = \begin{pmatrix} H_{1\mathcal{L}} \\ H_{2\mathcal{L}} \\ \ldots \\ H_{N_H \mathcal{L}} \end{pmatrix}. \quad (88)$$

However, in many situation the actual rank is much smaller.

The problem of generating a compressed MPO from a sum of products of local operators is as old as the field of tensor networks itself. There are essentially three standard approaches to perform this task (see [61–64] for an in-depth discussion):

- Manual construction of the MPO, in particular using complementary operators. This method, pioneered in DMRG, is suitable for simple problems but not for general ones.

- Symbolic compression of the naive-sum MPO, in particular using bipartite graph theory. This powerful, automatic approach is exact. However, this approach makes implementing approximate compression (within a certain tolerance) rather complex and does not exploit specific relations between the values of matrix elements (all that matters is whether a term is present or not).

- Compression of the naive-sum MPO using SVD. This approach is widely used but has a well-known stability issue for large systems due to a numerical truncation error.[2]

The stability issue of the SVD compression can be understood as arising from the fact that SVD finds the best low-rank approximation of an $N \times N$ matrix $A$ with respect to the *Frobenius* norm $|A|_F = (\sum_{ij} |A_{ij}|^2)^{1/2}$. When $A$ is the sum of terms having very different Frobenius norms, numerical truncation errors may lead the algorithm to wrongly discard those with small norms. As an illustration, consider $A = \mathbb{1} + \psi\psi^\dagger$ where $\mathbb{1}$ is the identity matrix and $\psi$ a normalized vector ($\psi^\dagger \psi = 1$). Here, we have $|A|_F = N + 3$. When $N$ is very large (as in many-body problems, where $N \approx 2^{\mathcal{L}}$), the $\mathcal{O}(1)$ contribution from $\psi\psi^\dagger$ may be lost in numerical noise. By contrast, the prrLU does not suffer from this problem since it optimizes a different target norm (the *maximum* norm of the Schur complement).

---

[2]In Ref. [64], it is shown that this issue can be resolved for certain local Hamiltonians in the DMRG context by exploiting their specific structure.

## 7.2 MPO algorithm for quantum many-body problems

We propose to compress the naive-sum MPO using prrLU instead of SVD. Very importantly, in this approach, the full naive-sum MPO is never built. Our auto-MPO algorithm follows a divide-and-conquer strategy:

(1) Collect a fixed number $N_a \ll N_H$ of terms $H_a$.

(2) Construct the naive MPO of their sum using Eq. (88).

(3) Compress the resulting tensor train using CI canonicalization.

(4) Repeat steps (1-3) until all $N_H$ terms have been processed.

(5) Sum and pairwise compress (formally in a binary tree) the $N_H/N_a$ partial sums from (4).

The validity of the final MPO can be checked explicitly making use of the fact that $H$ is a sparse matrix. Considering $F_\mu = H_{\sigma'\sigma}$ as a large vector, the tensor $\widetilde{F}_\mu$ is a correct unfolding of $F_\mu$ if and only if

$$\sum_\mu F_\mu^* \widetilde{F}_\mu = \sum_\mu |F_\mu|^2 = \sum_\mu |\widetilde{F}_\mu|^2 \tag{89}$$

(this guarantees that $|F - \widetilde{F}|_F = 0$ hence $F = \widetilde{F}$). This translates into

$$\sum_{a=1}^{N_H} \left( \sum_\mu [H_a]_\mu^* \widetilde{F}_\mu \right) = \sum_{a,a'=1}^{N_H} \left( \sum_\mu [H_{a'}^*]_\mu [H_a]_\mu \right) = \sum_\mu |\widetilde{F}_\mu|^2 . \tag{90}$$

Computing these expressions involves $\mathcal{O}(N_H)$ MPS contractions for the left side, enumerating the nonzero elements of the sparse matrices for the central part, and taking the trace of an MPO-MPO product for the right side. The same approach can be applied to the compression obtained by SVD or to compare the results of SVD and prrLU compressions.

We have tested the above algorithm against the same divide-and-conquer approach but with prrLU replaced by SVD, for the example $A = I_d + \psi\psi^\dagger$ where $\psi$ is the rank-1 MPS $\psi_{\sigma_1\cdots\sigma_\mathcal{L}} = \prod_\ell \delta_{\sigma_\ell,1}$. We found that SVD yields the correct rank-2 MPO for $\mathcal{L} < 103$ but fails for larger values of $\mathcal{L}$, incorrectly yielding a rank-1 MPO (the identity MPO). By contrast, the prrLU variant is stable for all values of $\mathcal{L}$ (up to 1000) that we have tested.

## 7.3 Illustration on Heisenberg and generic chemistry Hamiltonians

We illustrate the auto-MPO algorithm with two iconic Hamiltonian examples here: the Heisenberg Hamiltonian for a spin chain and a generic quantum chemistry Hamiltonian. The full code can be found in the folder `example/autoMPO/autoMPO.cpp` of the `xfac` library.

We start with the spin-$\frac{1}{2}$ Heisenberg Hamiltonian for an $\mathcal{L}$-site ring of spins:

$$H = \sum_{\ell=1}^{\mathcal{L}} S_\ell^z S_{\ell+1}^z + \frac{1}{2} \sum_{\ell=1}^{\mathcal{L}} \left( S_\ell^+ S_{\ell+1}^- + S_\ell^- S_{\ell+1}^+ \right), \tag{91}$$

$$S_\ell^\alpha = \underbrace{\mathbb{1} \otimes \mathbb{1} \otimes \cdots \otimes \mathbb{1}}_{\ell-1 \text{ times}} \otimes s^\alpha \otimes \underbrace{\mathbb{1} \otimes \cdots \otimes \mathbb{1}}_{\mathcal{L}-\ell \text{ times}} . \tag{92}$$

Here, the matrices $\mathbb{1} = \left(\begin{smallmatrix} 1 & 0 \\ 0 & 1 \end{smallmatrix}\right), s^z = \frac{1}{2}\left(\begin{smallmatrix} 1 & 0 \\ 0 & -1 \end{smallmatrix}\right), s^+ = \left(\begin{smallmatrix} 0 & 1 \\ 0 & 0 \end{smallmatrix}\right), s^- = \left(\begin{smallmatrix} 0 & 0 \\ 1 & 0 \end{smallmatrix}\right)$ represent the single-site identity and spin operators for a spin-$\frac{1}{2}$ Hilbert space, while $S_\ell^{\alpha=z,\pm}$ represent site-$\ell$ spin operators for the full Hilbert space of the $\mathcal{L}$-site chain, acting non-trivially only on site $\ell$. We use periodic boundary conditions, defining $S_{\mathcal{L}+1}^\alpha = S_1^\alpha$. Listing 4 shows a C++ code that first constructs the

Hamiltonian as a sum of local operators (an instance of the `polyOp` class), then generates the MPO (using the `to_tensorTrain()` method).

Our second example is a fully general quantum chemistry Hamiltonian of the form

$$H = \sum_{\ell_1\ell_2} K_{\ell_1\ell_2}\, c^\dagger_{\ell_1} c_{\ell_2} + \sum_{\ell_1<\ell_2,\ell_3<\ell_4} V_{\ell_1\ell_2\ell_3\ell_4}\, c^\dagger_{\ell_1} c^\dagger_{\ell_2} c_{\ell_3} c_{\ell_4}. \tag{93}$$

The fermionic operators $c^\dagger_\ell$ ($c_\ell$) create (destroy) an electron at spin-orbital $\ell$. They satisfy standard anti-commutation relations, which we implement using a Jordan-Wigner transformation. We take all the coefficients $K_{\ell_1\ell_2}$ and $V_{\ell_1\ell_2\ell_3\ell_4}$ as random numbers for our benchmark (in a real application, the number of significant Coulomb elements would be smaller, typically $\mathcal{L}^3$ instead of $\mathcal{L}^4$ here). The example code is given in Listing 5 below. Table 3 shows the obtained ranks for up to $\mathcal{L} = 50$ orbitals which match the theoretical expectation. Note that the number of terms $N_H$ for the larger size is greater than $10^6$, hence a naive approach would fail here.

```cpp
1  #include <xfac/tensor/auto_mpo.h>
2
3
4  using namespace std;
5  using namespace xfac;
6  using namespace xfac::autompo;
7
8
9  /// Heisenberg Hamiltonian (periodic boundary condition)
10 polyOp HeisenbergHam(int L)
11 {
12     auto Sz=[=](int i) { return prodOp {{ i%L, locOp {{-0.5,0},{0,0.5}} }}; };
13     auto Sp=[=](int i) { return prodOp {{ i%L, locOp {{0 ,0},{1,0}} }}; };
14     auto Sm=[=](int i) { return prodOp {{ i%L, locOp {{0 ,1},{0,0}} }}; };
15
16     polyOp H;
17     for(int i=0; i<L; i++) {
18         H += Sz(i)*Sz(i+1) ;
19         H += Sp(i)*Sm(i+1)*0.5 ;
20         H += Sm(i)*Sp(i+1)*0.5;
21     }
22     return H;
23 }
24
25
26 int main() {
27     int len=50;
28     auto H=HeisenbergHam(len);
29     TensorTrain mpo=H.to_tensorTrain();
30
31     cout<< "|1-<mpo|H>/<mpo|mpo>|=" << abs(1-H.overlap(mpo)/mpo.norm2()) << endl;
32
33     return 0;
34 }
```

Listing 4: C++ code to generate the MPO of the periodic Heisenberg Hamiltonian of Eq. (91). Lines 1–6 load the `xfac` library and namespaces. Lines 12–14 construct the spin operators of Eq. (92); note that only the single-site $2\times2$ matrices need to be specified explicitly. Lines 16–20 construct the sum $\sum_{\ell=1}^{\mathcal{L}}$ over all chain sites of the Hamiltonian Eq. (91). The maximun bond dimension obtained is 8 as it should be. This listing showcases the close similarity between formulae and corresponding code, which was one of the design goals of the `xfac` library.

Our C++ implementation is found in the namespace `xfac::autompo`. We define three classes: `locOp`, `prodOp`, and `polyOp`, corresponding to a local operator (i.e. a $2 \times 2$ matrix), a direct product of `locOp`, and a sum of `prodOp`, respectively. Our `prodOp` is a std::map going from int to locOp, while `polyOp` contains a std::vector of `prodOp`. The operators * and += are conveniently overloaded. Each of the classes `prodOp` and `polyOp` possesses the methods `to_tensorTrain()` (the actual algorithm to construct the MPO) and `overlap(mpo)` (to compute the left hand side of Eq. (89)).

```cpp
1  polyOp ChemistryHam(arma::mat const& K, arma::mat const& Vijkl)
2  {
3      auto Fermi=[=](int i, bool dagger)
4      {
5          locOp create={{0,1},{0,0}};
6          auto ci=prodOp {{ i, dagger ? create : create.t() }};
7          for(auto j=0; j<i; j++) ci[j]=locOp {{1,0},{0,-1}};    // fermionic sign
8          return ci;
9      };
10
11     auto L=K.n_rows;
12     polyOp H;
13
14     for(auto i=0u; i<L; i++)
15         for(auto j=0u; j<L; j++)
16             if (fabs(K(i,j))>1e-14)
17                 H += Fermi(i,true)*Fermi(j,false)*K(i,j);   // kinetic energy
18
19     for(auto i=0; i<L; i++)
20         for(auto j=i+1; j<L; j++)
21             for(auto k=0; k<L; k++)
22                 for(auto l=k+1; l<L; l++)
23                     if (fabs(Vijkl(i+j*L,k+l*L))>1e-14)
24                         H += Fermi(i,true)*Fermi(j,true)*Fermi(k,false)*
25                             Fermi(l,false)*Vijkl(i+j*L,k+l*L);
26     return H;
27 }
```

Listing 5: C++ code to generate the MPO of the quantum chemistry Hamiltonian of Eq. (93).

Table 3: Performance of our Auto-MPO construction for the quantum chemistry Hamiltonian of Eq. (93), for $\mathcal{L}$ orbitals, computed with an error tolerance of $\tau = 10^{-9}$. The third column is the bond dimension found with our approach. As a check, the fourth column gives the expected bond dimension obtained via the complementary-operator approach [63]. A naively constructed MPO would have bond dimension equal to the number of terms (2nd column), making it practically impossible to compress using SVD for $\mathcal{L} = 50$. [Code: Listing 5 (C++)]

| $\mathcal{L}$ | number of terms | bond dimension | $\mathcal{L}^2/2 + 3\mathcal{L}/2 + 2$ |
|---|---|---|---|
| 10 | 2125 | 67 | 67 |
| 30 | 190125 | 497 | 497 |
| 50 | 1503125 | 1327 | 1327 |

Table 4: Features supported by the main algorithms in `xfac/TCI.jl`.

| feature | TensorCI1 | TensorCI2 | | |
|---|---|---|---|---|
| | | 0-site | 1-site | 2-site |
| update mode | accumulative | accumulative & reset | | |
| pivot search | full & rook | full | full | full & rook |
| nesting condition | full | no | full | partial |
| environment error | supported | no | no | planned support |
| recompression | not supported | supported | supported | supported |
| global pivots | not supported | supported | supported | supported |

# 8 API and implementation details

We have presented a variety of use cases for our libraries `xfac/TCI.jl` in the examples above. After reading the present section, prospective users should be able to use our libraries in their own applications. In Sec. 8.1, we overview common features of the `xfac/TCI.jl` libraries. In Secs. 8.2 and 8.3 we provide language-specific information for C++ and Julia, respectively. We refer the reader to the tensor4all website [65] for the full documentation of the libraries.

Code for most examples contained in this paper is shown in Appendix B, and can be used as a starting point for implementations of new use cases. For more advanced use cases, it may be necessary to refer to the online documentation. We also encourage the readers to directly read the code of the library, in either language. It is indeed rather compact and often conveys the algorithms more transparently than lenghty explanations.

## 8.1 Implementation

For legacy reasons, `xfac/TCI.jl` contain two main classes for computing TCIs: `TensorCI1` and `TensorCI2`. `TensorCI1` is a variation of algorithm 5 of Ref. [12] and has been discussed in great detail in Ref. [13, Sec. III] (for a summary, see Sec. S-2 of the supplemental material of Ref. [15]). It is based on the conventional CI formula [38] and iteratively adds pivots one by one without ever removing any pivots (accumulative mode). `TensorCI2` is based on the more stable prrLU decomposition and implements 2-, 1- and 0-site TCI as described in this paper.

The numerical stability of the prrLU decomposition is inherited by `TensorCI2`, which often shows more reliable convergence. It is therefore used as a default in our codes. Nevertheless, since all TCI algorithms involve sampling, none of them is fully immune against missing some features of the tensor of interest, as already discussed above. Therefore, it may be necessary to enrich the sampling by proposing relevant global pivots before or during iteration (see Sec. 4.3.5). For instance, for the results shown in Fig. 9, we proposed 8 initial pivots according to the symmetry of the problem. Because of their different sampling patterns, it may also happen that `TensorCI1` finds much better approximations than `TensorCI2`. We have found at least one example where this was the case, and manual addition of some global pivots during initialization of `TensorCI2` solved the issue. There are minor differences in other features supported by `TensorCI1` and `TensorCI2`, which are summarized in Table 4. Most importantly, 0-site and 1-site optimization is only available in `TensorCI2`. Therefore, we offer convenient conversion between both classes.

General tensor trains, possibly obtained from an external source, are represented by a class `TensorTrain`. It supports related algorithms that are agnostic to the specific index structure of a TCI, such as evaluation, summation or compression using LU, CI or SVD. It also serves as an interface to other tensor network algorithms, such as those implemented in ITensor [33], to allow for quick incorporation of the TCI libraries into existing code. A `TensorCI1`/`TensorCI2` object

Table 5: Features supported by the main TCI algorithms in `xfac`. In the code, ci1 is a `TensorCI1`, ci2 is a `TensorCI2`, p is a `TensorCIParam` used to build a `TensorCI`, and tt is a `TensorTrain`. The method iterate(nIter, nSite) receives the number of iterations nIter to perform and the number of physical sites nSite to use for the matrix CI (can be 0, 1, or 2).

| Section | feature | variant | example C++ code |
|---------|---------|---------|------------------|
| 4.2 | nesting | no | ci2.iterate() |
| | | full | ci1.iterate() |
| | | | ci2.makeCanonical() |
| 4.3.3 | pivot update | accumulative | ci1.iterate() |
| | | reset | ci2.iterate() |
| 4.3.7 | environment | active if | p.weight=... |
| 4.3.4 | pivot search | rook | p.fullPiv=false |
| | | full | p.fullPiv=true |
| 4.3.5 | global pivots | | ci2.addGlobalPivots(...) |
| 4.4 | 0-site | | ci2.iterate(1,0) |
| | 1-site | | ci2.iterate(1,1) |
| 4.5 | compression | SVD | tt.compressSVD() |
| | | LU | tt.compressLU() |
| | | CI | tt.compressCI() |
| 4.5 | conversion | tci1 → tci2 | to_tci2(tci1) |
| | | tci2 → tci1 | to_tci1(tci2) |

can be trivially converted to a `TensorTrain` object. Conversion in the inverse direction is done by making the `TensorTrain` CI-canonical using the algorithm described in Sec. 4.5, resulting in a `TensorCI2` object.

## 8.2  C++ API (`xfac`)

The file "readme.txt" explains the installation procedure and how to generate the detailed documentation. The main components of the library are represented in Figure 12. As mentioned above, the classes `TensorCI1` and `TensorCI2` build a TCI of an input function. The main output is the tensor train, stored in the class `TensorTrain`, which represents a list of 3-leg tensors.

Below, we summarize the C++ API especially focusing on `TensorCI2`; the API for `TensorCI1` is similar and can be found in the documentation. A `TensorCI2` can be constructed from a tensor function $f : (x_1, x_2, \ldots, x_{\mathcal{L}}) \rightarrow \mathbb{C}$ and its local dimensions $\{d_\ell\}$ where the index $x_\ell \in \{1, \ldots, d_\ell\}$ with $\ell = 1, 2, \ldots, \mathcal{L}$. This is the main constructor:

```
1  TensorCI2(
2      function<T(vector<int>)> f,
3      vector<int> localDim,
4      TensorCI2Param param={}
5  );
```

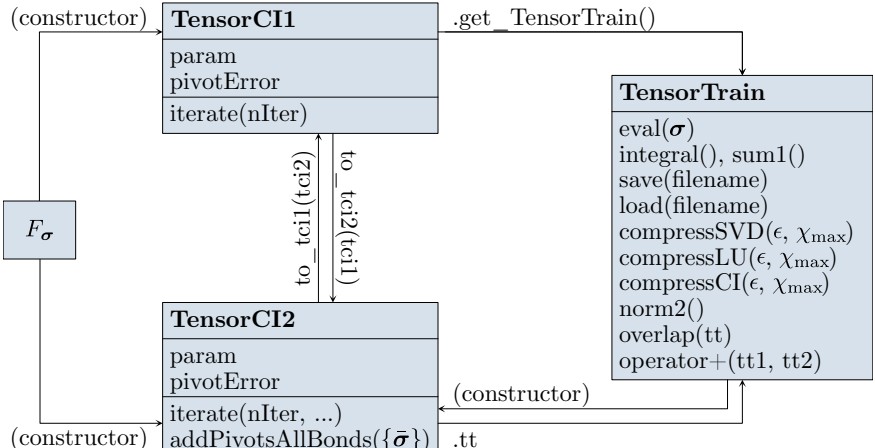

Figure 12: Scheme of the main conversions implemented in xfac.

The parameters of the cross interpolation can be set in the constructor by the class `TensorCI2Param`:

```
 1 struct TensorCI2Param {
 2   int bondDim=30;                        ///< max bond dimension of tensor train
 3   double reltol=1e-12;                   ///< expected relative tolerance of CI
 4   vector<int> pivot1;                    ///< first pivot (optional)
 5   bool fullPiv=false;                    ///< whether to use full pivoting
 6   int nRookIter=3;                       ///< number of rook pivoting iterations
 7   vector<vector<double>> weight;         ///< activates the ENV learning
 8   function<bool(vector<int>)> cond;      ///< cond(x)=false when x should not be a pivot
 9   bool useCachedFunction=true;           ///< whether to use internal caching
10 };
```

For `TensorCI1`, `TensorCI1Param` is used to set the parameters. We refer to the documentation for more details.

To factorize a continuous function $f : \mathbb{R}^{\mathcal{L}} \to \mathbb{R}$, xfac introduces the class `CTensorCI2` (or `CTensorCI1`). `CTensorCI2` is a `TensorCI2` that can be constructed from a multidimensional function $f$ by providing also the grid of points for each component:

```
 1 CTensorCI2(
 2   function<T(vector<double>)> f,
 3   vector<vector<double>> const& xi,
 4   TensorCI2Param param={}
 5 );
```

The main output of `CTensorCI2` is a continuous tensor train `CTensorTrain`, which can be evaluated at any point in $\mathbb{R}^{\mathcal{L}}$, including those outside the original grid (cf. App. A.4).

As discussed in Sec. 6.1, functions of continuous variables can also be discretized using the quantics representation. For that, xfac introduces the helper class `QTensorCI2`, currently available only for `TensorCI2`. It can be constructed from a multidimensional function $f$ by providing the quantics grid in addition to the parameters required for `TensorCI2`:

```
 1 QTensorCI2(
 2   function<T(vector<double>)> f,
 3   grid::Quantics const& qgrid,
 4   TensorCI2Param param={}
 5 );
```

Specifically, the `grid::Quantics` type represents an uniform grid on the hypercube $[a, b)^{\mathcal{N}}$ with $2^{\mathcal{RN}}$ points:

```
1 struct Quantics {
2     double a=0, b=1;        ///< start and end points of interval
3     int nBit=10;            ///< number of bits for each variable
4     int dim=1;              ///< dimension of hypercube
5     bool fused=false;       ///< whether to fuse the bits for the same scale (default:
           false)
6 }
```

The main output of `QTensorCI2` is a quantics tensor train `QTensorTrain`, which is a cheap representation of the function that can be evaluated, and saved/loaded to file.

## 8.3 Julia libraries

The Julia implementation of TCI is subdivided into several parts:

- `TensorCrossInterpolation.jl` (referred to as `TCI.jl`) contains only TCI and associated algorithms for tensor cross interpolation.

- `QuanticsGrids.jl` contains functionality to construct quantics grids, and to convert indices between direct and quantics representations.

- `QuanticsTCI.jl` is a thin wrapper around `TCI.jl` and `QuanticsGrids.jl` to allow for convenient quantics tensor cross interpolation in the most common use cases.

- `TCIITensorConversion.jl` is a small helper library to convert between tensor train objects and MPS/MPO objects of the `ITensors.jl` library.

All four libraries are available through Julia's general registry and can thus be installed by

```
1 import Pkg; Pkg.install("TensorCrossInterpolation")
```

and analogous commands. Below, we present only the main functionalities that were used for the examples in this paper. A complete documentation can be found online [65].

### 8.3.1 TensorCrossInterpolation.jl

Similar to xfac, `TCI.jl` has classes `TensorCI1` and `TensorCI2` that build a TCI of an input function, as well as a general-purpose `TensorTrain` class. These three classes and their main functions are shown in Fig. 13. Given a function of interest, $f : (x_1, x_2, \ldots, x_{\mathcal{L}}) \to \mathbb{C}$ and its local dimensions $\{d_\ell\}$, the most convenient way to obtain a `TensorCI1`/`TensorCI2` is by calling `crossinterpolate1`/`crossinterpolate2`. Since the algorithm based on prrLU is usually more stable, we recommend using `crossinterpolate2` as a default.

```
1 function crossinterpolate2(
2   ::Type{ValueType},        # Return type of f, usually Float64 or ComplexF64
3   f,                        # Function of interest: <@$f_\bsigma$@>
4   localdims::Union{Vector{Int},NTuple{N,Int}},  # Local dimensions <@$(d_1, \ldots,
        d_\scL)$@>
5   initialpivots::Vector{MultiIndex};  # List of initial pivots <@$\{\hat\bsigma\}$@>.
        Default: <@$\{(1, \ldots, 1)\}$@>
6   tolerance::Float64,       # Global error tolerance <@$\tau$@> for TCI. Default:
        <@$10^{-8}$@>
```



```
7    pivottolerance::Float64,   # Local error tolerance <@$\tau_{\mathrm{loc}}$@> for prrLU.
         Default: <@$\tau$@>
8    maxbonddim::Int,           # Maximum bond dimension <@$\chi_{\max}$@>. Default: no limit
9    maxiter::Int,              # Maximum number of half-sweeps. Default: <@$20$@>
10   pivotsearch::Symbol,       # Full or rook pivot search? Default: :full
11   normalizeerror::Bool,      # Normalize <@$\varepsilon$@> by <@$\max_{\bsigma \in
         \mathrm{samples}} F_\bsigma$@>? Default: true
12   ncheckhistory::Int         # Convergence criterion: <@$\varepsilon < \tau$@> for how
         many iterations? Default: 3
13 ) where {ValueType,N}
```

The three required positional arguments specify basic features of the tensor to be approximated. `f` is a function that produces tensor components when called with a vector of indices. For instance, `f([1, 2, 3, 4])` should return the value of $f_{1234}$. If appropriate pivots are known beforehand, they can be put in the list `initialpivots`, which is used to initialize the TCI. The convergence of TCI is controlled by mainly by the arguments `tolerance`, which is the global error tolerance $\tau$ of the TCI approximation, and `pivottolerance`, which determines the local error tolerance during 2-site updates. Usually, it is best to set `pivottolerance` to `tolerance` or slightly below `tolerance`. Both should be larger than the numerical accuracy, else the cross approximation may become numerically unstable. The maximum number of sweeps, controlled by `maxiter`, can be chosen rather small in reset mode, as the algorithm requires only a few sweeps.

After convergence, `crossinterpolate2` returns an object of type `TensorCI2` that represents the tensor train, as well as two vectors: `ranks` contains the bond dimension $\chi$, and `errors` the error estimate $\varepsilon$, both as a function of iteration number. For example, a possible call to `crossinterpolate2` to approximate a complex tensor $f_{\sigma_1\cdots\sigma_4}$ with 4 indices $\sigma_\ell \in \{1, 2, \ldots, 8\}$ up to tolerance $10^{-5}$ with TCI would be:

```
1  tci, ranks, errors = crossinterpolate2(ComplexF64, f, fill(8, 4); tolerance=1e-5)
```

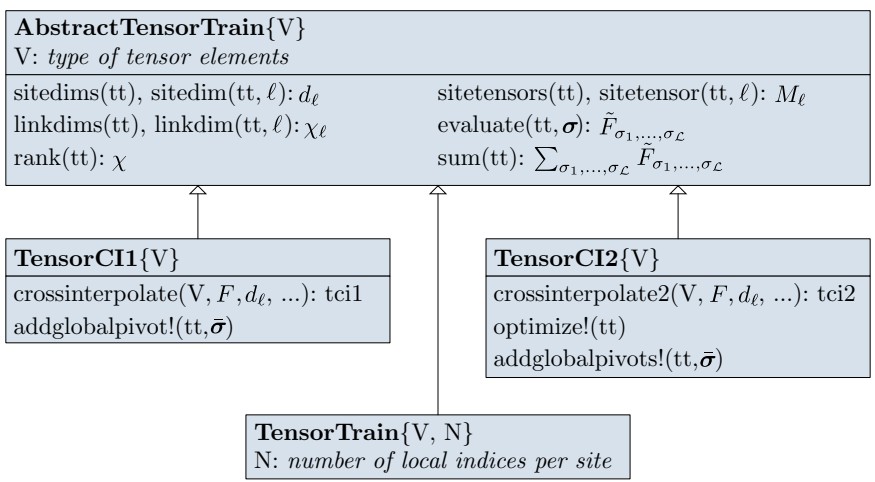

Figure 13: Schematic of the relations between the most important types in `TensorCrossInterpolation.jl`. Here, functions associated to types do not signify member functions, but rather functions operating on these types. By convention, functions ending with an exclamation mark '!' modify the object, while all other functions leave the object unchanged.

Table 6: Features supported by the main TCI algorithms in `TCI.jl`. In the code, tci1 is a `TensorCI1`, tci2 is a `TensorCI2`, and tt is a `TensorTrain`.

| Section | feature | variant | example julia code |
|---|---|---|---|
| 4.2 | nesting | no | crossinterpolate2(ValueType, f; ...) |
| | | full | crossinterpolate1(ValueType, f; ...) |
| | | | makecanonical!(tci2, f; ...) |
| 4.3.3 | pivot update | accumulative | crossinterpolate1(ValueType, f; ...) |
| | | reset | crossinterpolate2(ValueType, f; ...) |
| 4.3.4 | pivot search | rook | crossinterpolate2(..., pivotsearch=:rook, ...) |
| | | full | crossinterpolate2(..., pivotsearch=:full, ...) |
| 4.3.5 | global pivots | | addglobalpivot!(tci1, ...) |
| | | | addglobalpivots!(tci2, ...) |
| 4.4 | 0-site | | sweep0site!(tci2, ...) |
| | 1-site | | sweep1site!(tci2, ...) |
| 4.5 | compression | SVD | compress!(tt, :SVD, ...) |
| | | LU | compress!(tt, :LU, ...) |
| | | CI | compress!(tt, :CI, ...) |
| 4.5 | conversion | tci1 → tci2 | TensorCI1{ValueType}(tci2, f; ...) |
| | | tci2 → tci1 | TensorCI2{ValueType}(tci1) |

To evaluate the resulting TCI, call the object as a functor in the same way as the original function. For example, `tci([1, 2, 3, 4])` should be approximately equal to `f([1, 2, 3, 4])`. This is equivalent to a call `evaluate(tci, [1, 2, 3, 4])`. A sum over the TCI, e.g. to calculate an integral, is obtained by calling `sum(tci)`. If the only objective is to calculate an integral, it is more convenient to use the function `integrate(...)`, which calculates the integral of a function by building a weighted TCI on a Gauss–Kronrod grid and performing efficient weighted summation. Alternatively, quantics schemes described in the next section can be used for this task.

To apply more complicated tensor network algorithms to `tci`, it is useful to convert it into a `TensorTrain` object, which gives access to functions that do not preserve the CI-canonical gauge, such as SVD-based compression. With `TCIITensorConversion.jl`, all tensor train like objects can also be converted to actual `MPS` and `MPO` objects of the `ITensors.jl` library, which contains much more functionality [33].

### 8.3.2 Quantics grids and QTCI

Two associated libraries, `QuanticsGrids.jl` and `QuanticsTCI.jl`, offer convenient functionality to perform computations in quantics representation. `QuanticsGrids.jl` offers conversion between quantics indices, linear indices and function variables on (multidimensional) quantics grids. For example, fused quantics indices for a $\mathcal{R} = 10$ bit quantics grid on a hypercube $[-1, +1]^3$ can be obtained using the following code:

```julia
1 import QuanticsGrids as QG
2 grid = QG.DiscretizedGrid{3}(10, (-1.0, -1.0, -1.0), (1.0, 1.0, 1.0);
      unfoldingscheme=:fused)
3 sigma = QG.grididx_to_quantics(grid, (3, 4, 5)) # Translate <@$\vec{m} = (3, 4, 5)
      \rightarrow \bsigma(\vec{m})$@>
4 m = QG.quantics_to_grididx(grid, sigma)        # <@$\vec{m}(\bsigma)$@>
5 x = QG.quantics_to_origcoord(grid, sigma)      # <@$\vec{x}(\bsigma)$@>
```

To create a quantics TCI of a user-supplied function, these grids can be used together with `quanticscrossinterpolate(...)` from `QuanticsTCI.jl`, which translates a given function $f(x_1, \ldots, x_\mathcal{N})$ to its quantics representation $F_\sigma$, and applies the `crossinterpolate2` to $F_\sigma$ in a single call. The function signature is

```julia
function quanticscrossinterpolate(
  ::Type{ValueType},         # Return type of f, usually Float64 or ComplexF64
  f,                         # Function of interest <@$f(x)$@>
  grid,                      # Discretization grid, as QuanticsGrids.Grid, Array, or Range
  initialpivots::Vector{MultiIndex};  # List of initial pivots <@$\{\bar{\vec{m}}\}$@>.
       Default: <@$\{(1, \ldots, 1)\}$@>
  unfoldingscheme::Symbol,   # Fused or interleaved representation? Default: :interleaved
  kwargs...                  # All other arguments are forwarded to crossinterpolate2().
) where {ValueType}
```

The vector `initialpivots` enumerates the pivots used to initialize the TCI, as indices into `xvals` that are automatically translated to quantics form. Thus, this function takes care of all conversions to quantics representation that the user would otherwise have to do manually. It returns a QTCI, a vector of ranks, and a vector of errors, similar to `crossinterpolate2`, for example:

```julia
qtci, ranks, errors = quanticscrossinterpolate(Float64, f, grid, tolerance=1e-5)
```

Here, `qtci` is a `QuanticsTensorCI2` object, a thin wrapper around `TensorCI2` that translates between regular indices and their quantics representation. Similar to `TensorCI2`, objects of this type can be evaluated using function call syntax. For example, `qtci(m)` should be approximately equal to `f(QG.grididx_to_origcoord(grid, m))`. The object's components can be accessed as `qtci.tci` and `qtci.grid`. For a complete documentation of all functionality, see the online documentation of the respective libraries [65].

## 9 Perspectives

In this article, we have presented old and new variants of the tensor cross interpolation (TCI) algorithm, their open source C++, python and Julia implementations as well as a wide range of applications (integration in high dimension, solving partial differential equations, construction of matrix product operators, ...).

TCI has a very peculiar position among other tensor network algorithms: it provides an automatic way to map a very large variety of physics and applied mathematics problems onto the MPS toolbox. Of course not all mathematical objects admit a low-rank representation. But some problems do, and those will strongly benefit from being mapped onto the tensor network framework. Progress in computational sciences often corresponds to exploiting a particular structure of the problem. TCI belongs to the rare class of algorithms capable of discovering such structures for us. We surmise that TCI and related tools will play a major role in extending the scope of the MPS toolbox to applications beyond its original purpose of manipulating many-body wavefunctions.

An interesting side aspect of TCI is that offers a simple Go/No-Go test for the feasibility of speeding up computations using the MPS toolbox. Suppose, e.g., that one is in possession of a solver for a partial differential equation. One can feed some typical solutions into TCI to check whether they are strongly compressible—if so, a faster solver can likely be built using MPS tools. Using this very simple approach, the authors of this article have already identified numerous compressible objects in a wide range of contexts.

# Acknowledgments

J.v.D., H.S. and M.R. thank Markus Wallerberger for very interesting discussions on a relation between prrLU and CI decompositions. X.W. and O.P. thank Miles Stoudenmire for valuable feedback on the manuscript. M.R. thanks the Center for Computational Quantum Physics at the Flatiron Institute of the Simons Foundation for hospitality during an extended visit.

**Author contributions**   Y.N.F. and X.W. initiated the project. Y.N.F. conceived the TCI-via-prrLU algorithms with the help of X.W. and developed the xfac library. M.R., S.T. and H.S. developed the TCI.jl library based on xfac. Y.N.F., M.R., M.J., J.W.L., T.L. and T.K. contributed examples of applications of these libraries. X.W., M.R., O.P., J.v.D., Y.N.F. and H.S. wrote the paper and contributed to the proofs. Y.N.F. and M.R. contributed comparable amounts of work.

**Funding information**   H.S. was supported by JSPS KAKENHI Grants No. 21H01041, No. 21H01003, No. 22KK0226, and No. 23H03817, as well as JST PRESTO Grant No. JP-MJPR2012 and JST FOREST Grant No. JPMJFR2232, Japan. This work was supported by Institute of Mathematics for Industry, Joint Usage/Research Center in Kyushu University. (FY2023 CATEGORY "IUse of Julia in Mathematics and Physics" (2023a015).) X.W. acknowledges the funding of Plan France 2030 ANR-22-PETQ-0007 "EPIQ", the PEPR "EQUBITFLY", the ANR "DADI" and the CEA-FZJ French-German project AIDAS for funding. J.v.D. acknowledges funding from the Deutsche Forschungsgemeinschaft under grant INST 86/1885-1 FUGG and under Germany's Excellence Strategy EXC-2111 (Project No. 390814868), and the Munich Quantum Valley, supported by the Bavarian state government with funds from the Hightech Agenda Bayern Plus. The Flatiron Institute is a division of the Simons Foundation.

# A   Proofs of statements in the main text

## A.1   Proof of the quotient identity for the Schur complement

Below, we give a simple proof of the quotient identity (17), i.e. that taking the Schur complement with respect to multiple blocks either simultaneously or sequentially yields the same result. Consider two block matrices

$$A = \begin{pmatrix} A_{11} & A_{12} & A_{13} \\ A_{21} & A_{22} & A_{23} \\ A_{31} & A_{32} & A_{33} \end{pmatrix}, \qquad B \equiv \begin{pmatrix} A_{11} & A_{12} \\ A_{21} & A_{22} \end{pmatrix}, \tag{A.1}$$

where $A_{11}$ and $B$ are invertible submatrices of $A$. From (13), we factorize the $B$ matrix as

$$B = \begin{pmatrix} \mathbb{1}_{11} & 0 \\ A_{21}A_{11}^{-1} & \mathbb{1}_{22} \end{pmatrix} \begin{pmatrix} A_{11} & 0 \\ 0 & [B/A_{11}] \end{pmatrix} \begin{pmatrix} \mathbb{1}_{11} & A_{11}^{-1}A_{12} \\ 0 & \mathbb{1}_{22} \end{pmatrix},$$

which is easy to invert as

$$B^{-1} = \begin{pmatrix} \mathbb{1}_{11} & -A_{11}^{-1}A_{12} \\ 0 & \mathbb{1}_{22} \end{pmatrix} \begin{pmatrix} A_{11}^{-1} & 0 \\ 0 & [B/A_{11}]^{-1} \end{pmatrix} \begin{pmatrix} \mathbb{1}_{11} & 0 \\ -A_{21}A_{11}^{-1} & \mathbb{1}_{22} \end{pmatrix}.$$

This implies that

$$\left(B^{-1}\right)_{22} = [B/A_{11}]^{-1}. \tag{A.2}$$

We then form the Schur complement

$$[A/B] = A_{33} - (A_{31}, A_{32})B^{-1}\begin{pmatrix} A_{13} \\ A_{23} \end{pmatrix} \tag{A.3}$$

$$= A_{33} - (A_{31}, A_{32})\begin{pmatrix} \mathbb{1}_{11} & -A_{11}^{-1}A_{12} \\ 0 & \mathbb{1}_{22} \end{pmatrix}\begin{pmatrix} A_{11}^{-1} & 0 \\ 0 & [B/A_{11}]^{-1} \end{pmatrix}\begin{pmatrix} \mathbb{1}_{11} & 0 \\ -A_{21}A_{11}^{-1} & \mathbb{1}_{22} \end{pmatrix}\begin{pmatrix} A_{13} \\ A_{23} \end{pmatrix}$$

$$= A_{33} - \left(A_{31}, (A_{32} - A_{31}A_{11}^{-1}A_{12})\right)\begin{pmatrix} A_{11}^{-1} & 0 \\ 0 & [B/A_{11}]^{-1} \end{pmatrix}\begin{pmatrix} A_{13} \\ A_{23} - A_{21}A_{11}^{-1}A_{13} \end{pmatrix}$$

$$= A_{33} - A_{31}A_{11}^{-1}A_{13} - (A_{32} - A_{31}A_{11}^{-1}A_{12})[B/A_{11}]^{-1}(A_{23} - A_{21}A_{11}^{-1}A_{13}).$$

On the other hand, the Schur complement $[A/A_{11}]$ has the explicit block form

$$[A/A_{11}] = \begin{pmatrix} A_{22} & A_{23} \\ A_{32} & A_{33} \end{pmatrix} - \begin{pmatrix} A_{21} \\ A_{31} \end{pmatrix}(A_{11})^{-1}(A_{12} \; A_{13})$$

$$= \begin{pmatrix} [B/A_{11}] & A_{23} - A_{21}A_{11}^{-1}A_{13} \\ (A_{32} - A_{31}A_{11}^{-1}A_{12}) & A_{33} - A_{31}A_{11}^{-1}A_{13} \end{pmatrix}. \tag{A.4}$$

Taking the Schur complement of the above matrix with respect to its upper left block $[B/A_{11}]$ yields an expression which we recognize as the last line of Eq.(A.3). This proves the Schur quotient identity (17):

$$\left[[A/A_{11}]/[B/A_{11}]\right] = [A/B]. \tag{A.5}$$

## A.2 Convergence and rook conditions in block rook search

This section proves that the block rook search Algorithm 1 (see p. 14) converges, and that upon convergence, the pivots satisfy rook conditions.

**Definition: Block rook conditions** Given lists $\mathcal{I} = (i_1, \ldots, i_\chi)$ and $\mathcal{J} = (j_1, \ldots, j_\chi)$ of pivots, the block rook search algorithm alternates between factorizing $A(\mathbb{I}, \mathcal{J})$ and $A(\mathcal{I}, \mathbb{J})$, updating $\mathcal{I}$ and $\mathcal{J}$ after each factorization. In odd iterations, the block rook search obtains lists $\mathcal{I}' = (i_1', \ldots, i_\chi')$ and $\mathcal{J}' = (j_1', \ldots, j_\chi')$ from a prrLU factorization of $A(\mathbb{I}, \mathcal{J})$. Since the matrix $A(\mathbb{I}, \mathcal{J})$ has more rows than columns, the new column indices $\mathcal{J}'$ are a permutation of the old column indices $\mathcal{J}$, whereas $\mathcal{I}'$ may contain new elements that are not in $\mathcal{I}$. During a prrLU, we denote by $A_r$ the pivot matrix after the inclusion of the first $r$ pivots, i.e. $A_r = A((i_1', \ldots, i_r'), (j_1', \ldots, j_r'))$. These pivots satisfy,

$$(i_r', j_r') = \mathrm{argmax}[A(\mathbb{I}, \mathcal{J})/A_{r-1}]. \tag{A.6}$$

For even iterations, one factorizes $A(\mathcal{I}, \mathbb{J})$, the new pivots satisfy

$$(i_r', j_r') = \mathrm{argmax}[A(\mathcal{I}, \mathbb{J})/A_{r-1}], \tag{A.7}$$

and $\mathcal{I}'$ is a permuation of $\mathcal{I}$. After each prrLU, the pivots lists are updated $\mathcal{I} \leftarrow \mathcal{I}', \mathcal{J} \leftarrow \mathcal{J}'$. The process ends when $\mathcal{I}' = \mathcal{I}$ and $\mathcal{J}' = \mathcal{J}$.

**Definition: Rook conditions.** The pivots generated by sequential rook search (i.e. the standard rook search algorithm known from literature) fulfill the following set of rook conditions:

$$i_r = \mathrm{argmax}([A/A_{r-1}](\mathbb{I}, j_r)), \tag{A.8}$$

$$j_r = \mathrm{argmax}([A/A_{r-1}](i_r, \mathbb{J})), \tag{A.9}$$

where $A_r = A((i_1, \ldots, i_r), (j_1, \ldots, j_r))$.

**Statement 1.** The pivots $(i_1, j_1), \ldots, (i_\chi, j_\chi)$ found by a converged block rook search satisfy the rook conditions (A.8),(A.9). The proof follows from the restriction property of the Schur complement. At convergence, $(i'_r, j'_r) = (i_r, j_r)$ for each $r = 1, \ldots, \chi$. Applying Eq. (26) to the Schur complement of Eq.(A.6), one immediately gets Eq. (A.8). Similarly, one gets Eq. (A.9) from Eq. (A.7).

**Statement 2.** The block rook search must converge in a finite number of steps. The proof can be done iteratively. The search of $(i_1, j_1)$ correspond to looking for the maximum of $A(\mathbb{I}, \mathcal{J})$ (odd iterations) or $A(\mathcal{I}, \mathbb{J})$ (even iterations). For odd iterations $i'_1 = i_1$ unless new columns (that have never been seen by the algorithm) have been introduced in the previous even iteration. Since there are only a finite number of columns, this process must terminate in a finite number of iterations. The same argument works for $j_1$ and the even iterations.

To show that the search for $(i_2, j_2)$ must terminate, one applies the same reasoning to $[A/A_1]$ after $(i_1, j_1)$ has converged. One continues the proof iteratively for all $(i_r, j_r)$. In case the matrix $[A/(1, \ldots, r-1)]$ has multiple entries with the same maximum value, the ambiguity must be lifted to guarantee that the algorithm terminates. A solution is to choose $(i'_r, j'_r) = (i_r, j_r)$ whenever the previously seen pivot $(i_r, j_r)$ is among the maximum elements of that matrix.

## A.3 Nesting properties

Consider a tensor train $\widetilde{F}$ in TCI form (34). If its pivots satisfy nesting conditions, the $T_\ell^\sigma$ and $P_\ell$ matrices have certain useful properties, derived in Ref. [13, App. C] and invoked in the main text. Here, we summarize them and recapitulate their derivations.

For each $\ell$ we define the matrices $A_\ell^\sigma = T_\ell^\sigma P_\ell^{-1}$ and $B_\ell^\sigma = P_{\ell-1}^{-1} T_\ell^\sigma$, with elements

$$[A_\ell^\sigma]_{ii'} = \frac{A_\ell}{i \underset{\sigma}{\downarrow} i'} = \frac{T_\ell \quad P_\ell^{-1}}{i \underset{\sigma}{\square} j \; \diamond \; i'}, \qquad [B_\ell^\sigma]_{j'j} = \frac{B_\ell}{j' \underset{\sigma}{\downarrow} j} = \frac{P_{\ell-1}^{-1} \quad T_\ell}{j' \; \diamond \; i \underset{\sigma}{\square} j}, \qquad \text{(A.10a)}$$

for $i \in \mathcal{I}_{\ell-1}$, $i' \in \mathcal{I}_\ell$, $j' \in \mathcal{J}_\ell$, $j \in \mathcal{J}_{\ell+1}$, $\sigma \in \mathbb{S}_\ell$. If the unprimed indices $i \oplus (\sigma)$ or $(\sigma) \oplus j$ are restricted to $\mathcal{I}_\ell$ or $\mathcal{J}_\ell$, respectively, we obtain Kronecker symbols, in analogy to Eq. (10):

$$[A_\ell^\sigma]_{ii'} = \delta_{i \oplus (\sigma), i'}, \quad \forall\, i \oplus (\sigma) \in \mathcal{I}_\ell, \qquad [B_\ell^\sigma]_{j'j} = \delta_{j', (\sigma) \oplus j}, \quad \forall\, (\sigma) \oplus j \in \mathcal{J}_\ell. \qquad \text{(A.11)}$$

If the pivots are left-nested up to $\ell$, and if $\bar{\imath}_\ell = (\bar{\sigma}_1, \ldots, \bar{\sigma}_\ell)$ is an index from a row pivot list, $\bar{\imath}_\ell \in \mathcal{I}_\ell$, the same is true for any of its subindices, $\bar{\imath}_{\ell'} \in \mathcal{I}_{\ell'}$ for $\ell' < \ell$. Hence, iterative use of Eq. (A.11), starting from $A_1 A_2$, yields a telescope collapse of the following product:

$$\frac{A_1}{1 \underset{\bar{\sigma}_1}{\downarrow}} \cdots \frac{A_\ell}{\underset{\bar{\sigma}_\ell}{\downarrow} i'} = [A_1^{\bar{\sigma}_1} \cdots A_\ell^{\bar{\sigma}_\ell}]_{1i'} = \delta_{\bar{\imath}_\ell, i'}, \quad \forall\, \bar{\imath}_\ell \in \mathcal{I}_\ell \quad \text{if} \quad \mathcal{I}_0 < \mathcal{I}_1 < \cdots < \mathcal{I}_\ell. \qquad \text{(A.12a)}$$

Similarly, if the pivots are right-nested up to $\ell$, and $\bar{\jmath}_\ell = (\bar{\sigma}_\ell, \ldots, \bar{\sigma}_\mathcal{L}) \in \mathcal{J}_\ell$, we obtain

$$\frac{B_\ell}{j' \underset{\bar{\sigma}_\ell}{\downarrow}} \cdots \frac{B_\mathcal{L}}{\underset{\bar{\sigma}_\mathcal{L}}{\downarrow} 1} = [B_\ell^{\bar{\sigma}_\ell} \cdots B_\mathcal{L}^{\bar{\sigma}_\mathcal{L}}]_{j'1} = \delta_{j', \bar{\jmath}_\ell}, \quad \forall\, \bar{\jmath}_\ell \in \mathcal{J}_\ell \quad \text{if} \quad \mathcal{J}_\ell > \cdots > \mathcal{J}_\mathcal{L} > \mathcal{J}_{\mathcal{L}+1}. \qquad \text{(A.12b)}$$

We stress that such collapses do not apply for all configurations, only for pivots from left- or right-nested lists, respectively. Thus, the $A$s and $B$s are not isometries: $\sum_\sigma [A_\ell^{\sigma\dagger} A_\ell^\sigma]_{ii'} \neq \delta_{ii'}$ and $\sum_\sigma [B_\ell^\sigma B_\ell^{\sigma\dagger}]_{j'j} \neq \delta_{j'j}$, because the $\sum_\sigma$ sums involve non-pivot configurations.

The above telescope collapses are invoked in the following arguments:

- *1-Site nesting w.r.t. $T_\ell$:* We say that pivots are *nested w.r.t. $T_\ell$* if they are left-nested up to $\ell - 1$ and right-nested up $\ell + 1$. Then, if $\bar{\sigma}$ is a configuration from the 1d slice on which $T_\ell$ is built, $\bar{\sigma} \in \mathcal{I}_{\ell-1} \times \mathbb{S}_\ell \times \mathcal{J}_{\ell+1}$, the tensor train can be collapsed telescopically using Eqs. (A.12):

$$\widetilde{F}_{\bar{\sigma}} = \left[ A_1^{\bar{\sigma}_1} \cdots A_{\ell-1}^{\bar{\sigma}_{\ell-1}} T_\ell^{\bar{\sigma}_\ell} B_{\ell+1}^{\bar{\sigma}_{\ell+1}} \cdots B_{\mathcal{L}}^{\bar{\sigma}_{\mathcal{L}}} \right]_{11} = \left[ T_\ell^{\bar{\sigma}_\ell} \right]_{\bar{i}_{\ell-1}\bar{j}_{\ell+1}} = F_{\bar{\sigma}}, \qquad \text{(A.13)}$$

$$
\begin{array}{c}
\underset{\substack{1 \\ \bar{\sigma}_1}}{\overset{A_1}{\phantom{|}}} \cdots \underset{\bar{\sigma}_{\ell-1}}{\overset{A_{\ell-1}}{\phantom{|}}} \underset{\bar{\sigma}_\ell}{\overset{T_\ell}{\blacksquare}} \underset{\bar{\sigma}_{\ell+1}}{\overset{B_{\ell+1}}{\phantom{|}}} \cdots \underset{\bar{\sigma}_{\mathcal{L}}}{\overset{B_{\mathcal{L}}}{\phantom{|}}}_{1} = \underset{\bar{i}_{\ell-1} \; \bar{\sigma}_\ell \; \bar{j}_{\ell+1}}{\overset{T_\ell}{\blacksquare}}.
\end{array}
$$

  This proves that if the pivots of $\widetilde{F}$ are nested w.r.t. $T_\ell$, then $\widetilde{F}$ is exact on the slice $T_\ell$. It follows that if the pivots of $\widetilde{F}$ are nested w.r.t. *all* $T_\ell$, i.e. if they are fully nested (cf. Eq. (37)), then $\widetilde{F}$ is exact on all slices $T_\ell$ (and their sublices $P_\ell$), i.e. on all configurations $\bar{\sigma}$ from which it was built. Hence, a fully nested $\widetilde{F}$ is an interpolation of $F$.

- *0-Site nesting w.r.t. $P_\ell$:* We say that the pivots are *nested w.r.t. $P_\ell$* if they are left-nested up to $\ell$ and right-nested up $\ell + 1$. Then, $P_\ell$ is a subslice of both $T_\ell$ (since $\mathcal{I}_{\ell-1} < \mathcal{I}_\ell$) and $T_{\ell+1}$ (since $\mathcal{J}_{\ell+1} > \mathcal{J}_{\ell+2}$), and $\widetilde{F}$ is exact on both (by Eq. (A.13)), hence $\widetilde{F}$ is exact on $P_\ell$. Moreover, if we view $\widetilde{F}_{\sigma}$, with $\sigma = (i_\ell, j_{\ell+1})$, as a matrix with elements $[\widetilde{F}]_{i_\ell j_{\ell+1}}$, then its rank, say $r_\ell$, equals the dimension of $P_\ell$, i.e. $r_\ell = \chi_\ell$. This matrix rank $r_\ell$ is an intrinsic property of $\widetilde{F}$: it will stay fixed under all exact manipulations on $\widetilde{F}$, i.e. ones that leave its values on all configurations unchanged, e.g. exact SVDs or exact TCIs.

- *2-Site nesting w.r.t. $\Pi_\ell$:* We say that pivots are *nested w.r.t. $\Pi_\ell$* if they are left-nested up to $\ell - 1$ and right-nested up to $\ell + 2$. Then, if $\bar{\sigma}$ is a configuration from the 2d slice $\Pi_\ell$, i.e. $\bar{\sigma} \in \mathcal{I}_{\ell-1} \times \mathbb{S}_\ell \times \mathbb{S}_{\ell+1} \times \mathcal{J}_{\ell+2}$ so that $F_{\bar{\sigma}} = [\Pi_\ell]_{\bar{\sigma}}$, the tensor train can be collapsed telescopically to yield $\widetilde{F}_{\bar{\sigma}} = \left[ T_\ell^{\bar{\sigma}_\ell} P_\ell^{-1} T_{\ell+1}^{\bar{\sigma}_{\ell+1}} \right]_{\bar{i}_{\ell-1},\bar{j}_{\ell+2}}$. On this slice the local error, $\left[ \Pi_\ell - T_\ell P_\ell^{-1} T_{\ell+1} \right]_{\bar{\sigma}}$, is therefore equal to the global error, $\left[ F - \widetilde{F} \right]_{\bar{\sigma}}$, of the TCI approximation. A local update reducing the local error will thus also reduce the global error (cf. Eq. (40)).

## A.4  TCI in the continuum

This entire article is based on the cross interpolation of discrete tensors $F_{\sigma}$. In this appendix, we briefly discuss how this concept can be extended to continuum functions $f(\mathbf{x})$, as alluded to in Sec. 2.2.

  Consider the natural TCI representation of a function $f(\mathbf{x})$. Following the notations of Sec. 5.1, we suppose that $f(\mathbf{x})$ has been discretized on a grid $\{\mathbf{x}(\sigma)\}$ and is represented by a tensor $F_{\sigma} = f(\mathbf{x}(\sigma))$. Its TCI approximation $\widetilde{F}_{\sigma}$ is constructed from tensors $T_\ell$ that are slices of $F_{\sigma}$, i.e. with elements $[T_\ell^{\sigma}]_{i_{\ell-1}j_{\ell+1}}$ given by function values of $f(\mathbf{x}(\sigma))$,

$$[T_\ell^{\sigma}]_{i_{\ell-1}j_{\ell+1}} = f\left( x_1(\sigma_1), \ldots, x_{\ell-1}(\sigma_{\ell-1}), x_\ell(\sigma), x_{\ell+1}(\sigma_{\ell+1}), \ldots, x_{\mathcal{L}}(\sigma_{\mathcal{L}}) \right). \qquad \text{(A.14)}$$

In order to extend the associated TCI form to the continuum, we can simply extend $x_\ell(\sigma)$ to new values. In other words, one may perform the TCI on a grid and evaluate the obtained MPS on another, larger, grid. Formally, one simply replaces the matrix $T_\ell^{\sigma}$ by a matrix $T_\ell(x)$ defined as

$$[T_\ell(x)]_{i_{\ell-1}j_{\ell+1}} = f\left( x_1(\sigma_1), \ldots, x_{\ell-1}(\sigma_{\ell-1}), x, x_{\ell+1}(\sigma_{\ell+1}), \ldots, x_{\mathcal{L}}(\sigma_{\mathcal{L}}) \right). \qquad \text{(A.15)}$$

The obtained MPS $\tilde{f}(\mathbf{x}) = T_1(x_1)P_1^{-1}T_2(x_2)P_2^{-1}\cdots T_n(x_n)$ can be evaluated for any $\mathbf{x}$ in the continuum. In practice, it may be convenient to write the matrices $T_\ell(x)$ as an expansion over, say, Chebychev polynomials. This can be readily done if the initial grid is constructed from the corresponding Chebychev roots.

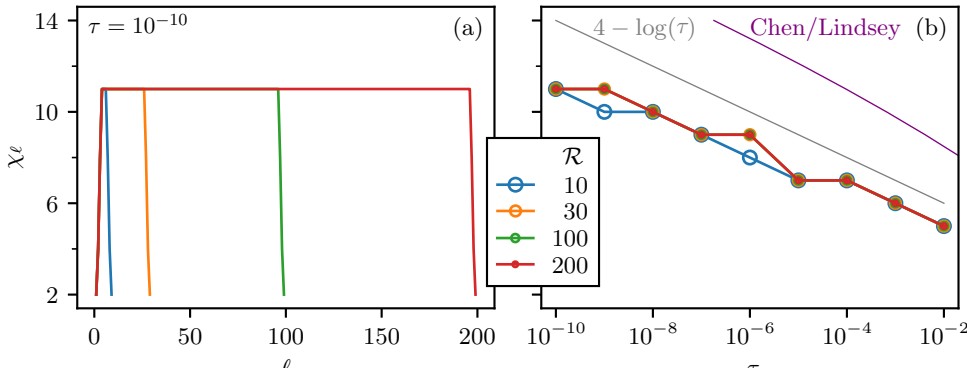

Figure 14: Bond dimensions of the discrete Fourier transform in quantics representation. (a) Bond dimensions $\chi_\ell$ along the MPS. Different colors signify different number of bits $2^{\mathcal{R}}$. Except at both ends of the MPS, the bond dimension $\chi_\ell$ is constant with a value independent of $\mathcal{R}$. (b) Dependence of the maximum bond dimension $\chi$ on the tolerance $\tau$. Bond dimensions increase logarithmically with decreasing tolerance (gray curve), and are independent of $\mathcal{R}$. The error bound from Ref. [66] is shown for comparison (purple curve).

## A.5 Small rank of the quantics Fourier transform

There is an intuitive explanation of the fact that we need to reverse the ordering of the indices of $k$ with respect to those of $m$: large-scale properties in real space (big shifts of $m$, associated with changes of $\sigma_\ell$ with $\ell \approx 1$) correspond to the Fourier transform at small momentum $k$ (i.e. changes of $\sigma'_\ell$ with $\ell \approx \mathcal{R}$), and indices that relate to the same scales should be fused together. More technically, the fact that the scale-reversed encodings (75) yield a tensor $T_{\boldsymbol{\mu}}$ of low rank stems from the factor $2^{\mathcal{R}-\ell'-\ell}$ in its phase. This factor is an integer for $\mathcal{R} - \ell' \geq \ell$ and $\simeq 0$ for $\mathcal{R} - \ell' \ll \ell$, hence $\exp[-i2\pi 2^{\mathcal{R}-\ell'-\ell}\sigma'_{\ell'}\sigma_\ell] = 1$ or $\simeq 1$, respectively, irrespective of the values of $\sigma'_{\ell'}$ and $\sigma_\ell$. Therefore, $T_{\boldsymbol{\sigma}'\boldsymbol{\sigma}}$ has a strong dependence on the index combinations $(\sigma'_{\ell'}, \sigma_\ell)$ only if neither of the above-mentioned inequalities apply, i.e. only if $\mathcal{R} - \ell' + 1$ is equal to or just slightly smaller than $\ell$; in this sense, the dependence of $T_{\boldsymbol{\sigma}'\boldsymbol{\sigma}}$ on $|(\mathcal{R} - \ell' + 1) - \ell|$ is rather short-ranged. This is illustrated in Figure 15 by the color-scale plot of $(2^{\mathcal{R}-\ell'-\ell}) \mod 1$ as a function of $\ell$ and $\mathcal{R} - \ell' + 1$: only a small set of coefficients is not close to an integer, namely those on or slightly below the diagonal, where $\mathcal{R} - \ell' + 1 = \ell$ or $\lesssim \ell$. This is the reason for defining $\mu_\ell$ as $(\sigma'_{\mathcal{R}-\ell+1}, \sigma_\ell)$, not $(\sigma'_\ell, \sigma_\ell)$. Then, tensor train unfoldings $\widetilde{T}_{\boldsymbol{\mu}}$ of $T_{\boldsymbol{\mu}}$ involve, in quantum information parlance, only *short-range entanglement* and have low rank [21, 58].

To show explicitly that TCI is able to find this low-rank representation, numerical experiments are shown in Fig. 14. The resulting tensor train has a rank of $\chi = 11$ for a tolerance of $\tau = 10^{-10}$, independent of $\mathcal{R}$. With decreasing tolerance, we observe that the bond di-

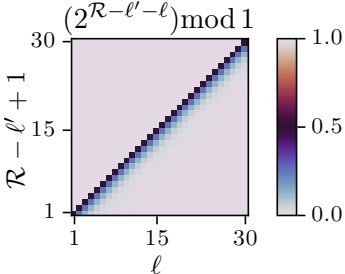

Figure 15: Fractional part of the phase factor of the Fourier transform.

mension increases slightly slower than logarithmically. This is similar to the results found in Refs. [21, 58, 59] for SVD-based truncation, and below the error bound obtained by Chen and Lindsey in Ref. [66].

# B  Code listings of examples discussed in the text

All the examples discussed in the text are associated with a runnable script (in one or more language) that can be found in the supplementary materials. Below, we show the most important parts of these scripts.

## B.1  Python scripts

### B.1.1  Integration of multivariate functions in environment mode

In the environment mode discussed in section 4.3.7, the TCI factorization aims to minimize the error of the integral (whereas with the usual bare mode, it aims to minimize the error on the intergrand). In xfac, the environment mode is switched on when the CTensorCI() class is instanciated with weights $w_\ell(\sigma_\ell)$. Providing these weights actually triggers two things: the activation of the environment mode and the fact that one uses the weighted unfolding Eq. (63). To perform the computation in environment mode, simply replace the call to CTensorCI() in line 19 of code Listing 1 by the lines of code in Listing 6.

```
1 # TCI1 Tensor factorization in "environment mode"
2 par = xfacpy.TensorCIParam()
3 par.weight = [well] * N
4 tci = xfacpy.CTensorCI(f, [xell] * N, par)
```

Listing 6: Python code snippet to perform a TCI factorization in environment mode combined with weighted unfolding Eq. (63) with weights $\{w_\ell(\sigma_\ell)\}$. To activate environment mode, the weights are passed to the CTensorCI() class using the attribute weights of the optional parameter par, which itself is an instance of class xfacpy.TensorCIParam(). The weights well must be provided for each of the N legs of the tensor. In this example we chose the weights to be identical for each leg (Gauss–Kronrod weights) and use the shorthand notation [well] * N, to generate a list of N lists of weights. Note that above code is generic and works similarly with all other TCI classes.

### B.1.2  Quantics for 2-dimensional integration

Listing 7 shows the quantics unfolding of the 2D function defined in Eq. (79) (see Fig. 8 in Sec. 6.3.1). Here, the variables $x$ and $y$ are both discretized onto grids of $M = 2^{\mathcal{R}}$ points each, with $\mathcal{R} = 40$. The corresponding MPS $\widetilde{F}_\sigma$ has $\mathcal{L} = 2\mathcal{R}$ indices, interleaved so that even indices $\sigma_{2\ell}$ encode $x$ and odd indices $\sigma_{2\ell+1}$ encode $y$. Lines 9–18 define conversions between grid indices and interleaved quantics indices. Lines 21–28 then define the function from Eq. (79) and the corresponding tensor in quantics representation, which is then TCI-unfolded using xfac in lines 31–38.

```
1 import xfacpy
2 import numpy as np
3
4 R = 40  # number of bits
```

```
5  M = 2**R  # number of grid points per variable
6  xmin, xmax, ymin, ymax = -5, +5, -5, +5  # domain of function f
7
8
9  def m_to_sigma(m_x, m_y):  # convert grid index (m_x,m_y) to quantics multi-index sigma
10     b1, b2 = np.binary_repr(m_x, width=R), np.binary_repr(m_y, width=R)
11     return np.ravel(list(zip(b1, b2))).astype('int')
12
13
14 def sigma_to_xy(sigma):  # convert quantics multi-index sigma to grid point (x,y)
15     m_x, m_y = int(''.join(map(str, sigma[0::2])), 2), int(
16         ''.join(map(str, sigma[1::2])), 2)
17     x, y = xmin + m_x*(xmax-xmin)/M, ymin + m_y*(ymax-ymin)/M
18     return x, y
19
20
21 def f(x, y):
22     return (np.exp(-0.4*(x**2+y**2))+1+np.sin(x*y)*np.exp(-x**2) +
23             np.cos(3*x*y)*np.exp(-y**2)+np.cos(x+y))
24
25
26 def f_tensor(sigma):  # quantics tensor
27     x, y = sigma_to_xy(sigma)
28     return f(x, y)
29
30
31 # load default parameters for initializing tci object T(1/P)T(1/P)T...
32 p = xfacpy.TensorCI1Param()
33 p.pivot1 = [0 for ind in range(2*R)]  # set first pivot to sigma=(0,0,...0)
34 # use first pivot to initialize tci
35 f_tci = xfacpy.TensorCI1(f_tensor, [2]*2*R, p)
36
37 for sweep in range(40):
38     f_tci.iterate()  # perform half-sweep
39
40 f_tt = f_tci.get_TensorTrain()  # get a TT object MMMM...
41 print("x\t f(x)\t f_tt(x)")
42
43 # evaluate the approximation on some regularly spaced points
44 for m_x in range(0, M, 2**(R-5)):
45     for m_y in range(0, M, 2**(R-5)):
46         sigma = m_to_sigma(m_x, m_y)
47         x, y = xmin + (xmax-xmin)*m_x/M, ymin + (ymax-ymin)*m_y/M
48         print(f"{x}\t{y}\t{f(x,y)}\t{f_tt.eval(sigma)}")
```

Listing 7: Python code, using xfac and TensorCI1 to compute the quantics approximation of the 2D-function $f(x, y)$ of Eq. (79) for $x$ and $y$ between $-5$ and $5$ using $2^{40} \times 2^{40}$ grid points, plotted in Fig. 8.

### B.1.3 Quantics for multi-dimensional integration

```
1  import xfacpy
2  from math import log
3
4  N = 5
5  xmin, xmax = 0.0, 1.0
6  R = 40  # Number of bits
7
8
9  def f(x):  # Integrand function
10     f.neval += 1
11     return 2**N / (1 + 2 * sum(x))
12
13
14 f.neval = 0
15
16 # Exact integral value in 5 dimensions
17 i5 = (- 65205 * log(3) - 6250 * log(5) + 24010 * log(7) + 14641 * log(11)) / 24
```

```
18
19  # Define the multidim quantics grid
20  grid = xfacpy.QuanticsGrid(a=xmin, b=xmax, nBit=R, dim=N, fused=False)
21
22
23  def fq(sigma):  # Integrand function on quatics grid
24      return f(grid.id_to_coord(sigma))
25
26
27  # TCI2 Tensor factorization
28  tci = xfacpy.TensorCI2(fq, [grid.tensorLocDim] * grid.tensorLen)
29
30  # Estimate integral and error
31  for hsweep in range(14):
32      tci.iterate()
33      # calculate the integal over the hypercube
34      itci = tci.tt.sum1()*grid.deltaVolume
35      print("hsweep= {}, neval= {}, I_tci= {:e}, |I_tci - I_exact|= {:e}, in-sample err=
            {:e}"
36            .format(hsweep+1, f.neval, itci, abs(itci - i5), tci.pivotError[-1]))
```

Listing 8: Python code to compute the integral $I^{(\mathcal{N}=5)}$ of Eq. (64) numerically using the multi-dimensional quantics integration from section 6.3.2. The integrand is formally discretized on $2^{40}$ points per variable $x_n$, while the factorization is performed on a multi-dimensional quantics representation using a tensor of $2^{5\times40}$ legs holding 2 sites each. The mapping between the original coordinate space **x** and the quantics representation is performed with the helper class `xfacpy.QuanticsGrid()`. The maximal bond dimension is 30 (default value of `xfacpy.TensorCI2`).

Listing 8 contains the code to compute the multi-dimensional integral Eq. (64) using quantics. The code is very similar to Listing 1, but replaces the Gauss–Kronrod helper functions with corresponding functions for a quantics grid. The helper class `xfacpy.QuanticsGrid()` in line 16 performs the mapping between the original coordinate space $(x_1, \ldots, x_{\mathcal{N}})$ and the quantics representation. `a=0` and `b=1` specify the bounds of the integration interval, the dimension is $\mathcal{N} = 5$ and we have chosen $\mathcal{R} = 40 \equiv$ `nBit`. The last argument `fused=False` indicate that the variable should not be fused, as described above.

The function `fq(sigma)` defined in line 18 and 19 evaluates the integrand function `f(x)` in the quantics representation. The method `QuanticsGrid.id_to_coord(sigma)` provides the mapping from the index position $\boldsymbol{\sigma}$ in the interleaved representation, written in terms of a binary number, onto the corresponding point $(x_1, \ldots, x_{\mathcal{N}})$ in the original argument space of the function `f(x)`. In our implementation $\boldsymbol{\sigma}$ is a Python list consisting of $2^{\mathcal{R}\mathcal{N}}$ binary elements, each either 0 or 1. The first or last element of $\boldsymbol{\sigma}$ represents the left-most or right-most bit, respectively. The tensor is instantiated in line 22. We have chosen TCI2 in this example as opposed to TCI1 in Listing 1, to demonstrate that both implementations of TCI1 and TCI2 are easily interchanged as their interfaces are similar. The overall result will be similar in both cases. The second argument of `TensorCI2`, namely `[grid.tensorLocDim] * grid.tensorLen`, creates a list `[2, 2, 2, ...]` with 200 elements, where each element is equal to 2 (our tensor has $\mathcal{N}\mathcal{R} = 200$ legs, with dimension 2 per leg). The rest of the script is similar to Listing 1, printing the result and the error for each iteration of the TCI algorithm.

### B.1.4 Heat equation using superfast Fourier transforms

```
1  import numpy as np
2  import xfacpy
3
4  # Grid parameters
5  R = 30
6  M = 2**R
```

```python
7    xmin, xmax = 2., 8.

9    # Manipulation of indices and grid
10   def m_to_sigma(m): # convert grid index to quantics multi-index
11       return [int(k) for k in np.binary_repr(m, width=R)]
12
13   def sigma_to_m(sigma): # reversed transform
14       return int(''.join(map(str, sigma)), 2)
15
16   def sigma_to_x(sigma): # convert quantics multi-index to position
17       return xmin + (xmax-xmin) * sigma_to_m(sigma) / 2**R
18
19   # Contraction
20   def contract_tt_MPO_MPS(tt_mpo, tt_mps):
21       mpo = tt_mpo.core
22       mps = tt_mps.core
23       res = xfacpy.TensorTrain_complex(len(mpo))
24       for i in range(len(mpo)):
25           aux = np.reshape( mpo[i], (mpo[i].shape[0], 2, 2, mpo[i].shape[2]))
26           m = np.tensordot(mps[i], aux, axes=([1], [2]))
27           m = np.transpose(m, (0, 2, 3, 1, 4))
28           newshape = (mps[i].shape[0]*mpo[i].shape[0], 2,
29                       mps[i].shape[2]*mpo[i].shape[2])
30           m = np.reshape(m, newshape)
31           res.setCoreAt(i, m)
32       res.compressSVD()
33       return res
34
35   # TCI
36   def build_TCI2_complex(fun, d, pivot1, pivots):
37       p = xfacpy.TensorCI2Param()
38       p.pivot1 = pivot1
39       p.useCachedFunction = True
40       p.fullPiv = True
41       ci = xfacpy.TensorCI2_complex(fun, [d]*R, p)
42       ci.addPivotsAllBonds(pivots)
43
44       nsweep = 3
45       for chi in [4,8,16,32,64]:
46           ci.param.bondDim = chi
47           for i in range(1, nsweep+1):
48               ci.iterate()
49               rank = np.max([x.shape[2] for x in ci.tt.core])
50           if (rank < chi) or (ci.pivotError[-1] < 1e-10):
51               break
52       return ci.tt
53
54
55   # Fourier transform MPOs
56   def qft(mu):
57       m1 = sigma_to_m( [mu[i]%2 for i in range(R)] )
58       m2_swapped = sigma_to_m(reversed( [mu[i]//2 for i in range(R)] ))
59       res = 1/(2**(R/2)) * np.exp(-1j * 2*np.pi * m1 * m2_swapped / 2**R)
60       return res
61   qft_mpo = build_TCI2_complex(qft, 4, pivot1=[3]*R, pivots=[])
62
63   def iqft(mu):
64       m1_swapped = sigma_to_m(reversed( [mu[i]%2 for i in range(R)] ))
65       m2 = sigma_to_m( [mu[i]//2 for i in range(R)] )
66       res = 1/(2**(R/2)) * np.exp(1j * 2*np.pi * m1_swapped * m2 / 2**R)
67       return res
68   iqft_mpo = build_TCI2_complex(iqft, 4, pivot1=[3]*R, pivots=[])
69
70
71   # Initial temperature distribution
72   def u0(x):
73       door = np.where(abs(x-5) <= 1.5, 1, 0)
74       oscillations = (1 + np.cos(120*x) * np.sin(180*x))
75       return door + 0.01 * oscillations
76
77   # Quantics representation of u0
78   def u0_tensor(sigma):
```

```
79        return u0(sigma_to_x(sigma))
80  pivot1= [np.random.randint(2) for i in range(R)]
81  while u0_tensor(pivot1) == 0:
82        pivot1 = [np.random.randint(2) for i in range(R)]
83  pivots = [m_to_sigma(m) for m in [0, M//4, M//2-2, M//2, 3*M//4, M-1]]
84  u0_mps = build_TCI2_complex(u0_tensor, 2, pivot1, pivots)
85
86
87  # Time propagator of the Heat equation in Fourier Space
88  def heat_kernel(sigma,t):
89        k = sigma_to_m(reversed(sigma) ) # work with swapped bits in Fourier space
90        delta = (xmax - xmin) / M
91        g_k = np.exp(- (2/delta * np.sin(np.pi * k / M))**2 * t)
92        return g_k
93
94  # MPO representation of the Heat kernel
95  def build_mpo_heat_kernel(t):
96        # build a Quantics MPS
97        pivot1 = [0]*R
98        heat_kernel_t = lambda sigma : heat_kernel(sigma, t)
99        heat_mps =  build_TCI2_complex(heat_kernel_t, 2, pivot1, pivots)
100       # convert to a diagonal MPO
101       heat_mps = heat_mps.core
102       res = xfacpy.TensorTrain_complex(R)
103       for i in range(R):
104           s = heat_mps[i].shape
105           aux = np.zeros((s[0],4,s[2]),dtype='complex')
106           aux[:,0,:] = heat_mps[i][:,0,:]
107           aux[:,3,:] = heat_mps[i][:,1,:]
108           res.setCoreAt(i,aux)
109       return res
110
111
112
113 # Time evolution
114 ft_u0 = contract_tt_MPO_MPS(qft_mpo,u0_mps)
115 ts = [5e-6, 0.0001, 0.01, 0.25, 1] # times list
116 samples_lists = []
117 m_list = [m for m in range(0,M,2**(R-4))]
118 x_list = [xmin + (xmax-xmin)/M * m for m in m_list]
119 for t in ts:
120       heat_k_mpo = build_mpo_heat_kernel(t)
121       ft_ut = contract_tt_MPO_MPS(heat_k_mpo, ft_u0)
122       ut = contract_tt_MPO_MPS(iqft_mpo, ft_ut)
123       samples_lists.append([np.real(ut.eval(m_to_sigma(m))) for m in m_list])
124
125 # print the evolution of temperature on some regularly spaced points
126 print(*(['x\t'] + [f'u(x,{t})' for t in ts]), sep='\t')
127 for i,x in enumerate(x_list):
128       print(*([f'{x:.3f}'] + [f'\t{samples_lists[j][i]:.3f}' for j in range(5)]),
129             sep='\t')
```

Listing 9: Python code using `TensorCI2` and the quantics representation defined in section 6 to build a superfast Fourier transform and solve the heat equation on a billion points grid, as shown in Fig. 11.

Listing 9 shows the code to solve the heat equation (80) on a $2^{30}$ points grid using quantics and the ultrafast Fourier transform MPO representation, as described in section 6.2.

The `contract_tt_MPO_MPS` defined line 20 performs the contraction of an MPO with an MPS. The `build_TCI2_complex` function defined line 36 calls TCI to build either a MPS when the second argument is $d = 2$ or an MPO when $d = 4$. We define an MPO as a tensor with dimension 4 per leg by fusing the input an output indices $\sigma, \sigma'$ following: $\mu = 2\sigma' + \sigma$.

The Fourier and inverse Fourier transform MPO representations are defined line 56 and 63. The initial temperature distribution (85) is defined line 72 and mapped to a quantics MPS. The `build_mpo_heat_kernel` method line 95 builds the MPO representation of the heat kernel operator (83) to perform time evolution in Fourier space for a given time $t$.

The final temperature distribution is then computed at 5 different times following (84a) and (84b). The code prints a temperature values on some regularly spaced grid points for visualization.

## B.2 C++ code

### B.2.1 Computation of partition functions

Listing 10 shows the C++ code to compute the partition function using TCI2 for classical Ising model with $|\ell - \ell'|^{-2}$ interaction detailed in Eq. (66). We increase the maximum bond dimension by incD (=5) step by step until the error is below the tolerance (=$10^{-10}$).

```cpp
1  #include <iostream>
2  #include <iomanip>
3  #include <vector>
4  #include <cmath>
5  #include "xfac/tensor/tensor_ci_2.h"
6
7  using namespace std;
8  using namespace xfac;
9
10 // function for compute energy for given spin configuration
11 double energy(vector<double> const& spin, vector<double> const& cpln, vector<int> const&
       config){
12   const int len= config.size();
13   vector<double> vecS(len);
14   transform(config.begin(), config.end(), vecS.begin(), [&spin](int i) {return
       spin.at(i);});
15
16   double sum2 = 0;
17   for (int ii=0; ii<len; ii++) {
18     const double si = vecS.at(ii);
19     for (int jj=ii+1; jj<len; jj++) {
20       const double sj = vecS.at(jj);
21       sum2 += -(si*sj) * cpln.at(jj-ii-1);
22     }
23   }
24   return sum2;
25 }
26
27 int main(int argc, char *argv[]){
28   vector<double> spin = {-1,1}; // down:-1; up:+1
29   const double beta = 0.6;      // inverse temperature
30   const double len = 32;        // system size
31
32   // cpln: coupling constant is |i-j|^(-2)
33   vector<double> cpln(len-1);
34   iota(cpln.begin(), cpln.end(), 1);
35   for_each(cpln.begin(), cpln.end(), [] (double& val) {
36     val = pow(val,-2);
37   });
38
39   // TT parameters
40   const int niter = 100; // # of tci sweeps
41   const int minD = 5;    // minimal bond dimension
42   const int incD = 5;    // increment of bond dimension
43   const int dim = spin.size(); //dim of the local space
44
45   // Define partition function
46   long count = 0;
47   auto prob=[=,&spin,&beta,&count](vector<int> const& config) {
48     count++;
49     return exp( -beta * energy(spin, cpln, config) );
50   };
51
52   // Initialize TCI
53   TensorCI2Param pp;
54   pp.bondDim = minD;
55   auto tci = TensorCI2<double>(prob, vector(len,dim),pp);
```

```
56
57   // Initialize PIVOTS
58   auto init1 = vector(len, 1);
59   auto init2 = vector(len, 0);
60   vector<vector<int>> seed = {init1,init2};
61   tci.addPivotsAllBonds(seed);
62
63   // TCI sweep
64   for (int iter=0; iter<niter; iter++){
65     tci.iterate();
66     cout << setw( 6) << fixed << iter << " "
67          << setw( 6) << fixed << tci.param.bondDim << " "
68          << setw(12) << fixed << count << " "
69          << setw(20) << scientific << setprecision(4) <<
70                 tci.pivotError.back()/tci.pivotError.front() << " "
71          << endl;
72     if (tci.pivotError.back() / tci.pivotError.front() <1e-10) {
72       break;
73     }
74     tci.pivotError.clear();
75     tci.param.bondDim += incD;
76   }
77
78   // Measure local moments
79   vector<vector<double>> ones = vector(len, vector(dim,1.0));
80   vector<double> m2(dim);
81   transform(spin.begin(), spin.end(), m2.begin(), [&len](double i) {return pow(i,2);});
82   const double norm = tci.tt.sum(ones);
83   // compute <M>
84   double aM1 = 0;
85   for (int ss=0; ss<len; ss++){
86     auto tmp = ones;
87     tmp.at(ss) = spin;
88     aM1 = aM1 + tci.tt.sum(tmp);
89   }
90   // compute <M^2>
91   double aM2 = 0;
92   for (int s1=0; s1<len; s1++){
93     for (int s2=s1+1; s2<len; s2++){
94       auto tmp = ones;
95       tmp.at(s1) = spin;
96       tmp.at(s2) = spin;
97       aM2 = aM2 + 2*tci.tt.sum(tmp);
98     }
99   }
100  for (int ss=0; ss<len; ss++){
101    auto tmp = ones;
102    tmp.at(ss) = m2;
103    aM2 = aM2 + tci.tt.sum(tmp);
104  }
105
106  // Print resutls
107  const double FE = log(norm)/len; //free energy
108  const double M1 = aM1/norm/len;
109  const double M2 = aM2/norm/len/len;
110  cout << "Beta: "
111       << setw( 6) << fixed << setprecision(2) << beta
112       << " | Free Energy: "
113       << setw(20) << fixed << setprecision(16) << FE
114       << " | M1: "
115       << setw(12) << fixed << setprecision(8)  << M1
116       << " | M2: "
117       << setw(12) << fixed << setprecision(8)  << M2
118       << " | # calls: " << setw(12) << fixed << count
119       << endl;
120  return 0;
121 }
```

Listing 10: C++ code to compute the partition function for classical Ising model with $|\ell - \ell'|^{-2}$ interactions; see Eq. (66).

### B.3 Julia scripts

#### B.3.1 TCI for high-dimensional Gauss–Kronrod quadrature

Listing 11 contains the Julia script for numerical integration of Eq. (64) using TCI for $\mathcal{N} = 5$ using bare error estimate (refer to Sec. 4.3.7).

```julia
 1  import TensorCrossInterpolation as TCI
 2
 3  N = 5                 # Number of dimensions <@$\cN$@>
 4  tolerance = 1e-10    # Tolerance of the internal TCI
 5  GKorder = 15          # Order of the Gauss-Kronrod rule to use
 6
 7  f(x) = 2^N / (1 + 2 * sum(x))   # Integrand
 8  integralvalue = TCI.integrate(Float64, f, fill(0.0, N), fill(1.0, N); tolerance, GKorder)
 9
10  # Exact value of integral for <@$\cN = 5$@>
11  i5 = (-65205 * log(3) - 6250 * log(5) + 24010 * log(7) + 14641 * log(11)) / 24
12  error = abs(integralvalue - i5)
13
14  @info "TCI integration with GK$GKorder: " integralvalue i5 error
```

Listing 11: Julia code to numerically compute the integral $I^{(\mathcal{N}=5)}$ of Eq. (64) using `TCI.jl`. Results are shown in Figs. 4.

#### B.3.2 Quantics TCI for 2-dimensional integration

Listing 12 shows the Julia script for the Julia code to compute a quantics TCI of the 2D-function $f(x, y)$ of Eq. (79) for $x$ and $y$ between $-5$ and $5$ using $\mathcal{R} = 40$. In practice, we use `QuanticsTCI.jl`, which is a thin wrapper around `TCI.jl` that provides a more user-friendly interface for quantics TCI.

```julia
 1  using QuanticsTCI
 2  import QuanticsGrids as QG
 3
 4  R = 40  # Number of bits <@$\cR$@>
 5  xygrid = QG.DiscretizedGrid{2}(R, (-5.0, -5.0), (5.0, 5.0)) # Discretization grid
        <@$\vec{x}(\bsigma)$@>
 6
 7  function f(x, y) # Function of interest <@$f(x)$@>
 8      return exp(-0.4*(x^2 + y^2)) + 1 + sin(x * y) * exp(-x^2) +
 9          cos(3*x*y) * exp(-y ^ 2) + cos(x+y)
10  end
11
12  # Construct and optimize quantics TCI <@$\tF_\bsigma$@>
13  f_tci, ranks, errors = quanticscrossinterpolate(Float64, f, xygrid; tolerance=1e-10)
14
15  # Print a table to compare <@$f(x)$@> and <@$\tF_\bsigma$@> on some regularly spaced
        points
16  println("x\t y\t f(x)\t\t\t f_tt(x)")
17  for index in CartesianIndices((10, 10))
18      m = Tuple(index) .* div(2^R, 10)
19      x, y = QG.grididx_to_origcoord(xygrid, m)
20      println("$x\t$y\t$(f(x, y))\t$(f_tci(m))")
21  end
22
23  println("Value of the integral: $(integral(f_tci))")
```

Listing 12: Julia code to compute a quantics TCI of the 2D-function $f(x, y)$ of Eq. (79) for $x, y \in [-5, 5)$ using $2^{40} \times 2^{40}$ grid points, plotted in Fig. 8.

### B.3.3  Quantics TCI for multi-dimensional integration

Listing 13 shows the Julia script to compute the integral $I^{(\mathcal{N}=5)}$ [Eq. (64)] numerically using the multi-dimensional quantics integration from Sec. 6.3.2. The code is equivalent to the Python script in Listing 8.

```julia
1  import QuanticsGrids as QG
2  import TensorCrossInterpolation as TCI
3
4  N = 5                # Number of dimensions <@$\cN$@>
5  tolerance = 1e-10    # Tolerance of the internal TCI
6  R = 40               # Number of bits <@$\cR$@>
7
8  f(x) = 2^N / (1 + 2 * sum(x))    # Integrand <@$f(\vec{x})$@>
9
10 # Discretization grid with <@$2^{\scN \scR}$@> points
11 grid = QG.DiscretizedGrid{N}(R, Tuple(fill(0.0, N)), Tuple(fill(1.0, N)),
       unfoldingscheme=:interleaved)
12 quanticsf(sigma) = f(QG.quantics_to_origcoord(grid, sigma)) # <@$f(\vec{x}(\bsigma))$@>
13
14 # Obtain the QTCI representation and evaluate the integral via factorized sum
15 tci, ranks, errors = TCI.crossinterpolate2(Float64, quanticsf, QG.localdimensions(grid);
       tolerance)
16
17 # Integral is sum multiplied with discretization volumne
18 integralvalue = TCI.sum(tci) * prod(QG.grid_step(grid))
19
20 # Exact value of integral for <@$\cN = 5$@>
21 i5 = (-65205 * log(3) - 6250 * log(5) + 24010 * log(7) + 14641 * log(11)) / 24
22 error = abs(integralvalue - i5)    # Error for <@$\cN = 5$@>
23
24 @info "Quantics TCI integration with R=$R: " integralvalue i5 error
```

Listing 13: Julia code to compute the integral $I^{(\mathcal{N}=5)}$ of Eq. (64) numerically using the multi-dimensional quantics integration from Sec. 6.3.2. The integrand is formally discretized on $2^{40}$ points per variable $x_n$, the factorization is performed on a multi-dimensional quantics representation of a tensor of $2^{5\times40}$ elements.

### B.3.4  Compressing existing data with TCI

In the example below, we illustrate how to apply (Q)TCI to some existing typical datasets. Let `dataset` be some pre-generated dataset (e.g. read from a file) in the form of an $\mathcal{N}$-dimensional array. Listing 14 shows a test for TCI compressibility. Listing 15 shows a similar test for QTCI compressibility.

```julia
1  import TensorCrossInterpolation as TCI
2
3  # Replace this line with the dataset to be tested for compressibility.
4  grid = range(-pi, pi; length=200)
5  dataset = [cos(x) + cos(y) + cos(z) for x in grid, y in grid, z in grid]
6
7  # Construct TCI
8  tolerance = 1e-5
9  tt, ranks, errors = TCI.crossinterpolate2(
10     Float64, i -> dataset[i...], collect(size(dataset)), tolerance=tolerance)
11
12 # Check error
13 ttdataset = [tt([i, j, k]) for i in axes(grid, 1), j in axes(grid, 1), k in axes(grid, 1)]
14 errors = abs.(ttdataset .- dataset)
15 println(
16     "TCI of the dataset with tolerance $tolerance has link dimensions
          $(TCI.linkdims(tt)), "
17     * "for a max error of $(maximum(errors))."
18 )
```

Listing 14: Julia code to test an existing dataset for TCI compressibility.

```
1  using QuanticsTCI
2  import TensorCrossInterpolation as TCI
3
4  # Number of bits
5  R = 8
6
7  # Replace with your dataset
8  grid = range(-pi, pi; length=2^R+1)[1:end-1] # exclude the end point
9  dataset = [cos(x) + cos(y) + cos(z) for x in grid, y in grid, z in grid]
10
11 # Perform QTCI
12 tolerance = 1e-5
13 qtt, ranks, errors = quanticscrossinterpolate(
14     dataset, tolerance=tolerance, unfoldingscheme=:fused)
15
16 # Check error
17 qttdataset = [qtt([i, j, k]) for i in axes(grid, 1), j in axes(grid, 1), k in axes(grid,
       1)]
18 error = abs.(qttdataset .- dataset)
19 println(
20     "Quantics TCI compression of the dataset with tolerance $tolerance has " *
21     "link dimensions $(TCI.linkdims(qtt.tci)), for a max error of $(maximum(error))."
22 )
```

Listing 15: Julia code to test an existing dataset for QTCI compressibility.

### B.3.5 Adding global pivots

We provide a simple example demonstrating the ergodicity problem discussed in Sec. 4.3.5 and how to fix it by adding a global pivot. We consider a function that takes a finite value at the first and last grid points, but is zero elsewhere (see Fig. 16):

$$f_m = \delta_{m,0} + \delta_{m,M-1}, \tag{B.1}$$

where $m = 0, 1, \cdots, M-1$ and $M = 2^{\mathcal{R}}$. When we interpolate this function using a 2-site TCI in the quantics representation with an initial pivot $\boldsymbol{\sigma} = (0, 0, \cdots, 0)$ ($m = 0$), the interpolation fails to capture the function at the last grid point for $\mathcal{R} \geq 3$ [67]. We can fix this by adding a global pivot at the last grid point. Listing 16 shows the Julia code to demonstrate this.

```
1  import TensorCrossInterpolation as TCI
2  import Random
3  import QuanticsGrids as QD
4  using PythonPlot: pyplot as plt
5  import PythonPlot
6  using LaTeXStrings
7
8  PythonPlot.matplotlib.rcParams["font.size"] = 15
9
10 # Number of bits
11 R = 4
12 tol = 1e-4
13
14 # f(q) = 1 if q = (1, 1, ..., 1) or q = (2, 2, ..., 2), 0 otherwise
15 f(q) = (all(q .== 1) || all(q .== 2)) ? 1.0 : 0.0
16
17 localdims = fill(2, R)
18
19 # Perform TCI with an initial pivot at (1, 1, ..., 1)
20 firstpivot = ones(Int, R)
21 tci, ranks, errors = TCI.crossinterpolate2(
22     Float64,
23     f,
24     localdims,
25     [firstpivot];
```

```
26      tolerance=tol,
27      nsearchglobalpivot=0 # Disable automatic global pivot search
28 )
29
30 # TCI fails to capture the function at (2, 2, ..., 2)
31 globalpivot = fill(2, R)
32 @assert isapprox(TCI.evaluate(tci, globalpivot), 0.0)
33
34 # Add (2, 2, ..., 2) as a global pivot
35 tci_globalpivot = deepcopy(tci)
36 TCI.addglobalpivots2sitesweep!(
37      tci_globalpivot, f, [globalpivot],
38      tolerance=tol
39 )
40 @assert isapprox(TCI.evaluate(tci_globalpivot, globalpivot), 1.0)
41
42 # Plot the function and the TCI reconstructions
43 grid = QD.InherentDiscreteGrid{1}(R)
44 ref = [f(QD.grididx_to_quantics(grid, i)) for i in 1:2^R]
45 reconst_tci = [tci(QD.grididx_to_quantics(grid, i)) for i in 1:2^R]
46 reconst_tci_globalpivot = [tci_globalpivot(QD.grididx_to_quantics(grid, i)) for i in
        1:2^R]
47
48 fig, ax = plt.subplots(figsize=(6.4, 3.0))
49 ax.plot(ref, label="ref", marker="", linestyle="--")
50 ax.plot(reconst_tci, label="TCI without global pivot", marker="x", linestyle="")
51 ax.plot(reconst_tci_globalpivot, label="TCI with global pivot", marker="+", linestyle="")
52 ax.set_xlabel(L"Index $m$")
53 ax.set_ylabel(L"f_m")
54 ax.legend(frameon=false)
55 plt.tight_layout()
56 fig.savefig("global_pivot.pdf")
```

Listing 16: Julia code demonstrating how to add a global pivot. We first construct a TCI object using 2-site TCI with an initial pivot at the first grid index. This fails to interpolate the function at the last grid index due to the local nature of 2-site TCI. This is fixed by adding a global pivot at the last grid index.

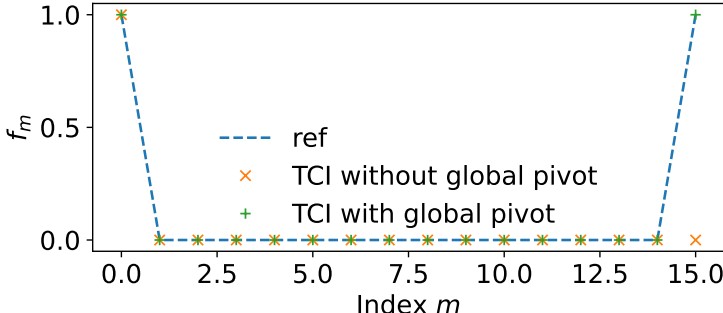

Figure 16: Comparison of the reference function (B.1), the result of TCI without an added global pivot, and the result of TCI with an added global pivot. The global pivot was added at the last grid index.

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
