# Peer review of "Learning tensor networks with tensor cross interpolation: new algorithms and libraries"

_SciPost Physics, doi:SciPost Phys. 18, 104 (2025)_

## Round 2 · Referee Report · Anonymous (Referee 1) · 2024-10-2

Strengths

  1. Detailed and thorough introduction to tensor cross interpolation and its applications.
  2. Conceptually new partial rank-revealing LU decomposition and explanation of relation to Shur's complement.
  3. Examples with source code.

Weaknesses

  1. Some sections, like 4.5.1 CI-canonicalization, are very technical and might be difficult to parse upon first reading.

Report

The paper provides a mathematical introduction and exposition to tensor cross interpolation (building upon previous works of the authors) and refines TCI by introducing the partial rank-revealing LU decomposition. The usefulness and applicability of the method are exemplified by various examples, including source code. A particular highlight is the superfast Fourier transformation. The paper is interesting and relevant for scientific computing in general, since e.g., high-dimensional data interpolation appears in various contexts. The paper is well-written, although somewhat technical in some places. I have listed some minor remarks and suggestions below. In summary, I deem the article meets the journal's acceptance criteria and recommend publication.

Requested changes

Minor remarks and suggestions: 1. In section "3.2.4 Relation with self-energy:" I've found the statement "Eq. (24) can be proven by the trivial identity..." quite hard to follow; isn't the identification with Eq. (15) sufficient - or could you provide one more step for the derivation? 2. You could illustrate the nesting conditions in section 4.2 by an example. 3. Before Eq. (33a): the expression "a zero-dimensional slice" might be unclear. 4. After Eq. (43): briefly explain why the fermionic algebra leads to only two non-zero elements. 5. On page 21 there is a typo: "the pivots fully..." -> "the pivots are fully..." 6. CI-canonicalization: can you motivate why converting an MPS to TCI form is advantageous or worthwhile? (Since TCI is a kind of MPS, I was naively expecting that an MPS representation is already sufficient.) 7. In Table 1, the constant prefactors 3 can probably be omitted from the asymptotic complexity - or is there a specific reason to keep them in the table? 8. In Fig. 5, I'd suggest using different color schemes for the beta and L parameters to avoid confusion (e.g., using the same color but different brightness levels to distinguish the three beta values).

Recommendation

Publish (easily meets expectations and criteria for this Journal; among top 50%)

  • validity: top
  • significance: high
  • originality: high
  • clarity: good
  • formatting: excellent
  • grammar: excellent

Author:  Yuriel Núñez Fernández  on 2025-01-06  [id 5085]

(in reply to Report 1 on 2024-10-02)
Category:
answer to question

We thank the referee for the detailed report. A list of responses to each request follows:

  1. The explanation was indeed not as clear as intended. As the referee correctly noted, it is possible to derive Eq. (24) from Eq.(15) directly through the definition of the Green's function and the self-energy. We have rewritten the explanation given in section 3.2.4 (page 11) to show the derivation of Eq. (24) from Eq. (15) explicitly.

  2. We have added an example for a fully nested configuration of pivot lists, with examples for pivots that would break nesting conditions (pages 17, 18).

  3. A "$k$-dimensional slice" of some tensor $F_{\boldsymbol{\sigma}}$ is a slice with $k$ free indices, i.e. all indices except $k$ of them are constrained. We have added a remark explaining the terminology above Eq. (33) (page 16).

  4. We have added an explaining sentence after Eq. (43) (page 21).

  5. We have corrected the sentence accordingly (page 22).

  6. We thank the referee for pointing out that the motivation for CI-canonicalization may have been unclear. A TCI is indeed an MPS, but an MPS is only a TCI if it is in CI gauge, i.e. comprised of tensors $T_\ell$ and $P_\ell^{-1}$ that are slices of the MPS itself. Some of the algorithms described in our manuscript, such as the TCI optimization algorithms and global pivot insertion, only apply to MPS in CI gauge: they rely on the property that all core tensors of a TCI are slices of the full tensor, or, equivalently, that the TCI can be fully defined through index sets $\mathcal{I_\ell}$ and $\mathcal{J}_\ell$. An arbitrary MPS does not generically fulfill this property, and thus the MPS must be transformed to CI gauge before applying the above-mentioned algorithms. We have added a paragraph detailing this motivation to section 4.5 (page 23). We give explicit examples where the conversion is advantageous.

  7. Indeed, the constant prefactors are inconsistent in the sense that $\mathcal{O}(f(x)) = \mathcal{O}(3 f(x))$. The original intention was to show that achieving full nesting requires exactly 3 sweeps, and the runtime cost is therefore thrice that of a 1-site sweep. Since this comparison only works between CI-canonicalization and 1-site sweeps, we have removed the constant factors of 3 from the asymptotic complexities (now Table 2, page 27).

  8. We now use two different color schemes $\beta$ and $\mathcal{L}$ in Fig. 5, as suggested by the referee.

---

## Round 2 · Referee Report · Anonymous (Referee 2) · 2024-11-27

Strengths

  1. Pedagocical introduction to tensor cross interpolation
  2. Joint presentation of 2 independent packages implementing the algorithms separately in xfac (C++/python) and TensorCrossInterpolation.jl (julia)
  3. New prrLU algorithm as alternative to CI with better numerical stability

Report

The authors provide a pedagogical overview over tensor cross interpolation (CI), a new algorithm that can find a low-rank approximation of an exponentially large tensor in the form of a matrix product state by interpolation from a (polynomially) small subset of its entries. They review the various techniques and tricks introduced in previous works and provide a new variant based on the partially rank revealing LU (prrLU) decomposition, which provides much better numerical stability than the direct CI by avoiding the necessity to take inverses of ill-conditioned matrices. They further showcase possible applications with explicit implementation examples using the xfac and TensorCrossInterpolation.jl packages. As the TCI lifts the MPS toolbox to a plethora of applicatios in other subjects than the quantum many body systems it was traditionally applied to, I agree witht the authors that it provides a synergetic link between different research areas, and recommend a publication.

Requested changes

  1. The manuscript discussed a lot of examples where TCI works - but it leaves open the question where it fails (in the Perspectives section suggesting to just try it out). What are examples where you are aware of a failure? For example, I'm wondering:

  2. what are examples of functions that cannot be represented well in the quantics tensor representation, apart from white noise?

  3. to what extend is the 1D structure of the underlying MPS important? It seems straight forward to extend the examples of the partition function in section 5.3 and heat equation in 6.3 to physically 2D systems, but does the TCI compression still work in this case?

  4. In chapter 7, you argue that the SVD-based compression is unstable. Let me point out that the cited Ref. 64 adresses exactly that instability for typical hamiltonians by acknowledging the jordan-block form of the MPO matrices and separating the idenity part from the local terms.

  5. Typos:

  6. duplicated with in second bullet point of 4.2

  7. Section 5.2, when discussing the listing 1 code example the reference to line 27, tci.sumWeighted([well] * N) from the main text doesn't match the code in the listing.
  8. In Section 6.2 under "Quantics Fourier transform", you first cite Ref 58 for the low rank, but just before equ. 77b, you cite Ref. 21 for the same statement.

Recommendation

Publish (easily meets expectations and criteria for this Journal; among top 50%)

  • validity: top
  • significance: top
  • originality: high
  • clarity: top
  • formatting: perfect
  • grammar: perfect

Author:  Yuriel Núñez Fernández  on 2025-01-06  [id 5087]

(in reply to Report 2 on 2024-11-27)
Category:
answer to question

We thank the referee for the detailed report. A list of responses to each request follows:

  1. We discuss each example below:
    • Any tensor, and thus any function, can be factorized into a quantics tensor train representation. The failure mode for QTT is thus not one where the QTT fails to resolve the function, but one where the costs of using a QTT exceeds those of other representations. As QTT are factorizations in length scales, they are inefficient if the length scales are not separable. An example of such a function is a function $f(\mathbf{x})$ which is equal to $1$ if $\mathbf x$ is inside some contour, and equal to $0$ otherwise. If the contour has non-separable structure, the bond dimension of a QTT that approximates $f$ is close to maximal. Fairly simple curved contours, such as a sphere, have enough non-separable structure to lead to large bond dimensions $\chi \approx 1000$. We have added a paragraph that describes more examples for QTT ranks of simple analytic functions in multiple dimensions to section 6.1 (page 35).
    • Indeed, the examples can trivially be extended to higher dimensions. Whether the convergence with rank will remain fast strongly depends to the problem under consideration. This question has already been intensely debated in the context of the many-body problem. For instance MPS-based methods such as DMRG can be used to study the Hubbard model in 2D but two-dimensional tensor networks (so-called PEPS) are preferable. Nevertheless, since computing with MPS is much faster than PEPS, MPS can remain competitive even in this case. In that respect, the two above examples are very different, and we answer the question separately. Note that in general we can gain expressivity by generalizing the underlying tensor network structure to trees, as shown in arXiv:2410.03572. In examples such as the partition function in section 5.3, the dimensionality of the target function is crucial, basically because we can apply the arguments based on entanglement established in the field of tensor network wavefunction. In other words, the entanglement of the target function is an intrinsic property and TCI is just a way to reveal it. The $W_{\boldsymbol{\sigma}}$ tensor involved in the computation of a 2D partition function has entanglement properties satisfying the area law. Consequently, an exact MPS representation of this tensor will require a rank that increases exponentially with system size (irrespective of whether the MPS obtained using SVD- or CI-based compression). In contrast, we use the quantics representation to solve the heat equation. This reveals the (possibly small) entanglement between different scales, which is not necessarily related to the dimensionality, as was already shown in the literature. Quantics has been applied to partial differential equations in arbitrary dimensions, including quantum chemistry in 3D (see for example arXiv:2308.03508).
  2. Indeed, the comment of the referee clarifies the general scope of the auto-MPO topic. We emphasize that our CI-based compression naturally works without the need to explicitly include specific structure of the operator. We have added a footnote about the approach of Ref. 64 in the main text (page 44).
  3. Typos:
    • We have changed the sentence accordingly (page 17).
    • We have changed the main text accordingly (page 31).
    • We thank the referee for pointing out this inconsistency. The statement that the quantics Fourier transform can be represented as an MPO with low rank can be found in both references, which discovered this representation independent from each other. Therefore, we now cite both references in all places where this statement is made (pages 35, 36, 58).

---

## Round 3 · Referee Report · Christian Mendl (Referee 1) · 2025-1-7

Report

The comments and suggestions from the first review report have been well-addressed in the revised version. I recommend publication.

Recommendation

Publish (surpasses expectations and criteria for this Journal; among top 10%)

---

## Round 3 · Referee Report · Anonymous (Referee 2) · 2025-1-8

Report

The authors have addressed the questions and comments from the previous reports, and I recommend a publication.

Recommendation

Publish (easily meets expectations and criteria for this Journal; among top 50%)

---

## Round 3 · Author Response

We are grateful for the detailed comments of the referees on the manuscript, especially for pointing out where clarity could be improved. We have implemented all changes requested by them. Our paper is further improved by these changes.

---

## Round 3 · List of Changes

Related to Report #1:

  1. We have rewritten the explanation given in section 3.2.4 (page 11) to show the derivation of Eq. (24) from Eq. (15) explicitly.
  2. We have added an example for a fully nested configuration of pivot lists, with examples for pivots that would break nesting conditions (pages 17, 18).
  3. We have added a remark explaining the terminology above Eq. (33) (page 16).
  4. We have added an explaining sentence after Eq. (43) (page 21).
  5. We have corrected the sentence accordingly (page 22).
  6. We have added a paragraph detailing this motivation to section 4.5 (page 23). We give explicit examples where the conversion is advantageous.
  7. We have removed the constant factors of 3 from the asymptotic complexities (now Table 2, page 27).
  8. We now use two different color schemes $\beta$ and $\mathcal{L}$ in Fig.~5, as suggested by the referee.

Related to Report #2:

  1. About failure mode of tci
    • We have added a paragraph that describes more examples for QTT ranks of simple analytic functions in multiple dimensions to section 6.1 (page 35).
  2. We have added a footnote about the approach of Ref. 64 in the main text (page 44).
  3. About typos:
    • We have changed the sentence accordingly (page 17).
    • We have changed the main text accordingly (page 31).
    • We now cite both references in all places where this statement is made (pages 35, 36, 58).

The following entries in the bibliography were changed:

* Sozykin, Chertkov, et al., 2022: removed duplicate arXiv number.
* Jolly, Núñez Fernández, and Waintal, 2023: corrected second author's name (Núñez Fernández) and removed duplicate arXiv number.
* Sakurai, Takahashi, and Miyamoto, 2024: added digital object identifier and removed duplicate arXiv number.
* Takahashi, Sakurai, and Shinaoka, 2024: added digital object identifier and removed duplicate arXiv number.
* Ishida, Okada, Hoshino, and Shinaoka, 2024: added digital object identifier and removed duplicate arXiv number.
* \'Sroda, Inayoshi, Shinaoka, and Werner, 2024: new reference.
*  Gourianov, Lubasch, et al., 2022: removed duplicate arXiv number.
*  Peddinti, Pisoni, et al., 2023: removed duplicate arXiv number.
* Sakaue, Shinaoka, and Sakurai, 2024: added digital object identifier and removed duplicate arXiv number.
* Golub and van Loan, 1996: corrected ISBN.
* Poole and Neal, 2000: added digital object identifier.
* Verstraete and Cirac, 2004: removed duplicate arXiv number.
* Woolfe, Hill, and Hollenberg, 2014: removed duplicate arXiv number.
* Removed entry https://gitlab.com/tensors4fields, as all material there has been moved to tensor4all.org. All references have been updated accordingly.
* Chen and Lindsey, 2024: removed duplicate arXiv number.

---

## Editorial Decision

published